# The glacial systems model (GSM) Version 25G

Lev Tarasov[1], Benoit S. Lecavalier[1,2], Kevin Hank[3], and David Pollard[4]

[1]Department of Physics and Physical Oceanography, Memorial University of Newfoundland and Labrador, St. John's, Canada, A1B 3X7
[2]Defence Research & Development Canada (DRDC), Suffield Research Centre, PO Box 4000, Station Main, Medicine Hat, T1A 8K6, Alberta, Canada
[3]Natural Environment Research Council, British Antarctic Survey, High Cross, Madingley Road, Cambridge, CB3 0ET, United Kingdom
[4]Earth and Environmental Systems Institute, Pennsylvania State University, University Park, PA 16802, USA; and Department of Earth, Geographic, and Climate Sciences, University of Massachusetts, Amherst, MA 01003, USA

**Correspondence:** Lev Tarasov (lev@mun.ca)

**Abstract.** We document the glacial system model (GSM). It is designed for large ensemble ice sheet modelling in glacial cycle contexts. A distinguishing feature is the extent to which it addresses relevant forcing and process uncertainties. The GSM has evolved from three decades of effort to constrain the evolution of each last glacial cycle ice sheet (North American, Greenlandic, Icelandic, Eurasian, Patagonian, and Antarctic, and soon Tibetan). The core ice dynamics uses a hybrid shallow-shelf and shallow-ice approximation with full thermo-mechanical coupling. It also includes one of the largest range of relevant processes for this context of any model to date, ranging from visco-elastic glacial isostatic adjustment with 0-order geoidal deflection to state-of-the-art subglacial sediment production, transport, and deposition. Furthermore, the GSM is to date the only model to have all of the above processes bidirectionally coupled with each other. Other relevant distinguishing features include: permafrost resolving bed-thermodynamics, a fast diagnostic solution of down-slope surface drainage and lake filling, subgrid hypsometric surface mass balance and ice flow, simple thermodynamic lake and sea ice representations, subglacial hydrology with dynamically evolving partitioning between distributed and channelized flow, and surface melt that physically accounts for insolation changes via a novel insolation above freezing scheme.

To address the most challenging part of paleo ice sheet modelling, the GSM includes both a 2D energy balance climate model and variants of traditional input time series weighted interpolation (aka "glacial indexing") of fields from General Circulation Model (GCM) simulations, all under ensemble parametric specification. It also includes options for one and two way scripted coupling with climate models.

We demonstrate the significant errors that can ensue in the glacial cycle simulation of a single ice sheet when three aspects of glacial isostatic adjustment are ignored (as is typical). These are geoidal deformation, global ice load input, and correction of initial topography for present-day isostatic disequilibrium. We also draw attention to the relatively high sensitivity of the GSM (and presumably other ice sheet models) to the specification of the temperature dependence for basal sliding activation.

The associated code archive includes configuration options for all major last glacial cycle ice sheets as well as idealized geometries and validation test setups.

# 1 Introduction

Paleo ice sheet modelling contexts have some features that impose distinct requirements in comparison to models designed for present-day and near future centennial scale modelling. For the latter, certain processes, such as subglacial sediment production, transport, and deposition, are effectively irrelevant (given their much longer characteristic time-scales, *e.g.,* Drew and Tarasov, 2024), and others, such as glacial isostatic adjustment (GIA), can be more simply or more carefully approximated, depending on context and simulation time interval (Whitehouse et al., 2019). Furthermore, given the large uncertainties in climate forcing over a glacial cycle, ice sheet modelling for the purpose of constraining past ice sheet evolution requires large ensembles of simulations with adequate degrees of freedom in the climate forcing. This along with the O(100 kyr) glacial cycle timescale implies that computational costs are a much more critical consideration for paleo ice sheet modelling as compared to present-day modelling contexts.

Other potentially critical processes and feedbacks for glacial cycle contexts that are typically ignored for present-day ice sheet modelling include the following: the evolution of proglacial lakes and their impact on ice sheet mass loss (*e.g.,* Tarasov and Peltier, 2006), the evolution of landfast perennial lake and sea ice into ice shelves and ice tongues (*e.g.,* Bradley and England, 2008), the evolution of geothermal heat flux and permafrost depth and their impact on basal thermal energy balance (*e.g.,* Tarasov and Peltier, 2004), and the impact of changing insolation (due to orbital forcing) on surface melt (*e.g.,* van de Berg et al., 2011).

The Glacial Systems Model (GSM) is a numerical model for simulating ice sheets and their interactions with the rest of the Earth system over glacial cycle time-scales. It features fully coupled components relevant to this context that explicitly model all of the processes and feedbacks listed above. To date, such a complete set of components is not found in any other ice sheet model (various current models used for paleo ice sheet modelling have many, but not all of the GSM features, *e.g.,* Winkelmann et al., 2011; Sato and Greve, 2012; Pollard et al., 2015; Quiquet et al., 2018; Robinson et al., 2020; Berends et al., 2022). For instance, the GSM is the only current glaciological ice sheet model that can resolve englacial sediment transport and subglacial sediment production due to quarrying (with otherwise only Pollard et al., 2015, having even a representation of subglacial sediment transport). Furthermore, no other model has a sediment process model fully coupled to glacio-isostatic adjustment (albeit asynchronously).

A key and distinguishing GSM design consideration is a focus on uncertainty quantification. This entails parameterization of as many significant glacial system uncertainties as is reasonably possible, given the much greater challenge in assessing structural modelling uncertainties. This results in the GSM currently having a minimum of 30 (Patagonia) to a maximum of 53 (North America) ensemble parameters for a single paleo ice sheet. In contrast, all previous paleo ice sheet modelling studies not using the GSM (or its precursor, Tarasov and Peltier, 2004) use fewer than 7 parameters (*e.g.,* Albrecht et al., 2020a, use 4 ensemble parameters for ensemble modelling of the last glacial cycle Antarctic ice sheet). The GSM also has noise insertion options for partial quantification of structural uncertainties (*i.e.* model uncertainties not captured by ensemble parameters, a feature shared by only one other ice sheet model Verjans et al., 2022).

Given the large ensemble requirement for paleo ice sheet modelling, the GSM is highly optimized for serial computation. For instance, a 205 kyr Antarctic simulation at 40 km resolution taking about 10 hours on a (circa 2016) single Intel Xeon E5-2650 (2.3GHz) core. The lack of parallelization limits spatial grid resolution to about 10 km for continental scales, with $0.5^o$ by $0.25^o$ longitude by latitude being the current default. However, to partially compensate for limited spatial resolution, the GSM includes a state-of-the-art subgrid hypsometric surface mass-balance and ice flow model (Le Morzadec et al., 2015).

Another important feature is that the model has been configured for all but one of the last glacial cycle ice sheets (including Antarctic, Greenland, North American, Eurasian, Icelandic, Patagonian, and soon Tibetan), and includes options for one and two way coupling with external climate models. The GSM's internal climate representation enables full Pleistocene simulations of Northern Hemispheric ice sheets in approximate accord with inferences for past sea level from benthic $\delta^{18}$O records with simulations only driven by orbital and greenhouse gas forcing (Drew and Tarasov, 2024).

The GSM has a long history (going back to the purely shallow ice approximation version in Tarasov and Peltier, 1997a), with a significant change being the incorporation of the Pollard et al. (2015) ice dynamical core for inclusion of shallow shelf physics completed in 2017. The current form of the GSM largely matches that used for publications using the GSM from 2020 onwards with a chronology of relevant changes summarized in the supplement. Though the model continues to evolve, it is now at a stage and in a form appropriate for initial public release. Below we document the GSM and provide example test results of the impact of some of its relatively unique features.

## 2   Model description

The GSM includes a number of distinct, fully coupled components (cf Table 1 and Fig. 1). The hybrid shallow-shelf/shallow-ice (SSA/SIA) dynamical core (2.4 and A1) is a modified version of the PSU3D ice sheet model (Pollard and DeConto, 2012; Pollard et al., 2015; Pollard and DeConto, 2020). This dynamical core includes a grounding line flux parameterization (Schoof, 2007; Pollard and DeConto, 2020) and is able to capture marine ice shelf instabilities (Pollard et al., 2015). The main differences from that of Pollard et al. (2015) are conversion to Fortran 90 standards, the addition of the NSPCG generalized numerical solver (Kincaid et al., 1989) for solving the SSA stress-balance, separate basal drag laws for soft and hard beds (2.5), the addition of an alternative grounding line flux parameterization (Tsai et al., 2015), a few minor bug fixes, and changes to the iterative SSA solution to further optimize speed and numerical stability while allowing recovery from iterative convergence failures.

The ice thermo-dynamics (section 2.6) is an energy-conserving finite-volume formulation (Patankar, 1980). The bed thermodynamics (section 2.6) resolves permafrost and includes corrections for seasonal snow cover over ice-free land (Tarasov and Peltier, 2007).

The GSM has an asynchronously coupled global visco-elastic isostatic response GIA solver (Tarasov and Peltier, 1997b) with a linear approximation for the deflection of the geoid (with space-time dependence) from a spherically symmetric eustatic mean sea level anomaly (section 2.15).

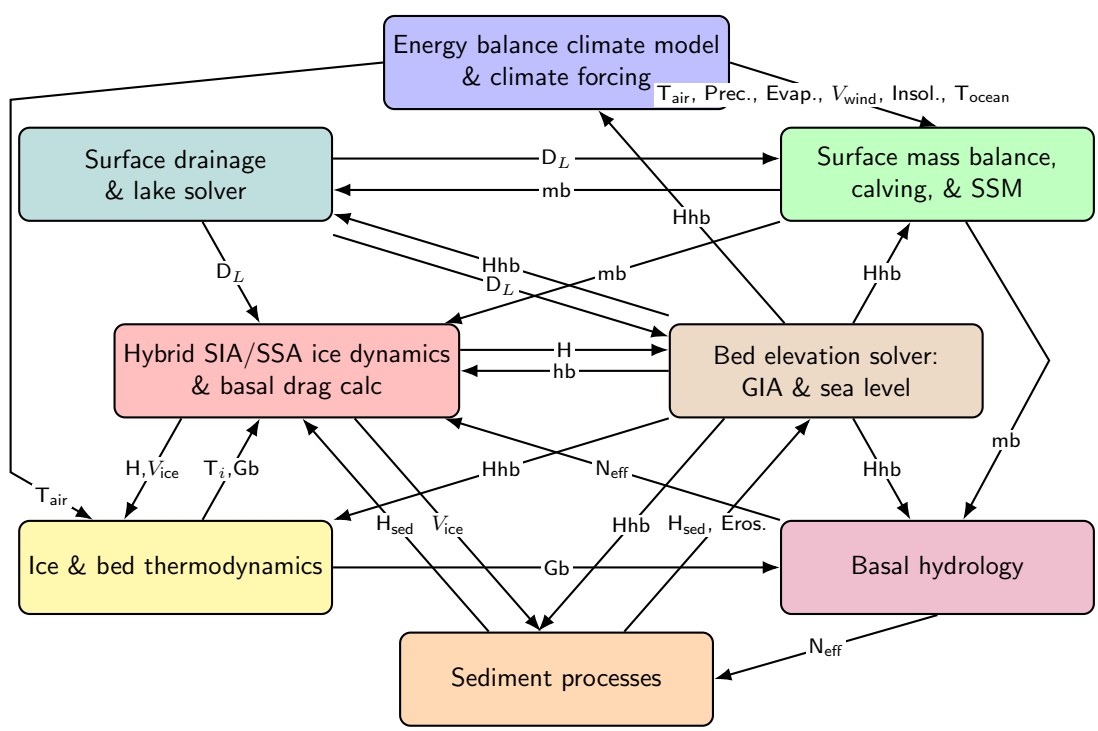

**Figure 1.** GSM key components and process linkages indicating key variables passed. Variable names are as follows. H is ice thickness, hb is bed elevation relative to contemporaneous sea level, Hhb is mix of H, hb, and surface elevation. H is only updated in the ice dynamics module, but to minimize clutter, H is shown as flowing through the GIA solver (the latter also uses H as an input). $D_L$ is lake depth, mb is mass-balance components (melt, accumulation, and calving), SSM is submarine melt, Gb is basal melt, $N_{eff}$ is basal effective pressure, $T_i$ is ice temperature, $V_{ice}$ is the 3D ice velocity field, $H_{sed}$ is subglacial sediment thickness, and 'Eros' is basal erosion. All passed variables are mean annual (or longer) except for mean monthly climatology atmospheric climate inputs for the surface mass balance component: 2 meter air temperature ($T_{air}$), precipitation (Prec), net evaporation (Evap), horizontal wind velocity field ($V_{wind}$) used for orographic downscaling of precipitation, and surface insolation (Insol). To avoid clutter, the subgrid mass balance and ice flow component is not shown, but spans the surface mass balance and hybrid ice dynamics components.

Surface drainage (section 2.8) is diagnostically resolved using a down-slope formulation that fills topographic depressions (lakes) while maintaining mass-conservation (Tarasov and Peltier, 2006). The resolving of pro-glacial lakes permits inclusion of a simplified lake ice parameterization (section 2.9) and a fresh-water calving component limited by available lake heat (section 2.7.5). Surface melt includes a novel positive degree solar insolation component (section 2.7.1). Subshelf melt and freeze-on uses a buoyant plume parameterization (section 2.7.2), while calving parametrically accounts for crack propagation, hydro-fracturing, and strain (section 2.7.3 and 2.7.4) enabling the capture of marine ice cliff instabilities.

**Table 1.** GSM components and relevant section for description.

| Component/process | subsection |
|---|---|
| hybrid ice dynamics | 2.4 and A1 |
| basal drag | 2.5 |
| ice and bed thermodynamics | 2.6 |
| surface melt and refreezing | 2.7.1 |
| submarine melt and refreezing | 2.7.2 |
| ice shelf calving | 2.7.3 |
| tidewater calving | 2.7.4 |
| lacustrine calving | 2.7.5 |
| surface drainage and lake formation | 2.8 |
| lake and sea ice formation | 2.9 |
| glacial indices | 2.10.1 |
| atmospheric climate forcing | 2.10.2 |
| orographic precipitation downscaling | 2.10.3 |
| ocean climate forcing | 2.10.4 |
| subgrid mass balance and ice flow | 2.11 |
| ice margin nudging | 2.12 |
| subglacial hydrology | 2.13 |
| subglacial sediment processes | 2.14 |
| GIA solver | 2.15 |
| noise injection | 2.16 |

For climate forcing, the GSM simultaneously uses glacially-indexed (section 2.10.1) GCM snapshots and an asynchronously-coupled, geographically-resolved, energy balance climate model with non-linear snow and sea ice albedo feedback (section 2.10.2). Precipitation is subject to wind-climatology driven orographic forcing to account for the strong impact of orography (section 2.10.3). The default ocean temperature forcing uses results from a transient deglacial GCM simulation (TRACE Liu et al., 2009), again subject to glacial indexing (section 2.10.4).

Other optional components include the following. The GSM has several fully coupled basal hydrology representations (section 2.13 and Drew and Tarasov, 2023) and a state-of-the-art subglacial sediment process model (section 2.14 and Drew and Tarasov, 2024). There is a subgrid hypsometric surface mass-balance and ice flow model to partly compensate for coarser grid resolution (section 2.11 and Le Morzadec et al., 2015). There is the option of nudging surface mass balance and calving to facilitate consistency of simulated and geologically reconstructed ice margin locations (section 2.12). Finally, the GSM includes a compile flag for activating stochastic noise additions to poorly constrained processes in the model (section 2.16).

Relevant details on each component are provided in the indicated subsections (Table 1). A summary log of key 2023 to 2025 GSM version changes is in section 2 of the supplement.

## 2.1 GSM parameters

Any complex geophysical model will have a host of poorly constrained parameters that significantly impact model simulations. To address this, the GSM has a comparatively large set of ensemble parameters (Tables 2 and 3) that define the parametric

configuration for a given run. This is in contrast to "GSM parameters" denoting parameters set to fixed values based on physics or relative model insensitivity to the parameter (Table 4). The selection of the ensemble parameters has been refined during the course of decades of calibration and sensitivity analysis (e.g. Tarasov and Peltier, 2004; Tarasov et al., 2012).

While each paleo ice sheet will have some specific ensemble parameters, the majority of parameters are common across ice sheets. The majority of ensemble parameters are scaled to give a $0 \rightarrow 1$ input range. However, others that have a somewhat

clearer physical interpretation may have a different scaling. To ensure a reasonably comparable scale, all ensemble parameters are subject to the following scaling rules. First the maximum value must be greater than or equal to 0 and less than 10. Second, the parameter range must be greater than 0.1. Some ensemble parameters (*e.g.,* $h_{\mathrm{wbCrit}}$ in Tables 2) may be scaled exponentially to permit a nominal $0 \rightarrow 1$ range.

## 2.2 GSM grid and structure

The GSM is mostly coded following Fortran 90 conventions and formatting, including the use of modules and implicit none (the latter requires each variable to be explicit declared). A few legacy components, including the coupled energy balance climate model (EBM) have yet to be brought to this standard. Numerous configuration options are under compile flag control.

The GSM has an ice sheet index dimension allowing separate ice sheet domains instead of, for instance, requiring a grid covering the whole globe. There are 3 horizontal grid options: regular dx,dy; regular longitude,latitude; and polar stereographic

projection, with the option of the latter two running concurrently for different ice sheets. As significant changes in the vertical temperature gradient are possible at any relative depth (*e.g.,* Cuffey and Paterson, 2010), the vertical temperature grid is a standard sigma grid for ice temperature. It has default 65 layers (GSM parameter NCZ) with an effectively vertically split basal cell to more accurately compute basal melt. As changes in the vertical gradient of horizontal ice velocities are concentrated near the bed, the GSM uses an irregularly spaced sigma grid for velocities with high resolution near the bed (with default NLEV= 12

layers). Velocities and temperatures are transferred to each other's grid by linear interpolation. As is fairly standard, the ice sheet model uses an Arakawa C-grid, with fluxes and velocities computed on grid cell interfaces.

Given the large variability of the shape of vertical temperature profiles (*e.g.,* Cuffey and Paterson, 2010), the vertical grid for solving ice temperatures is on a uniform sigma grid except for an effectively vertically split basal grid cell to more accurately compute basal melt. This variability also necessitates a significantly larger number of levels than required for the ice velocity

solution.

The bed thermodynamic grid has exponential spacing in accordance with diffusion scaling. It has a default 26 layers (GSM parameter NTBZ) and scaling exponent value of 1.21 (GSM parameter RbedSCALE) for a default 4 km deep bed.

**Table 2.** Ensemble parameters not related to climate forcing. Input parameter ranges are given by the $(a \rightarrow b)$ specification with subsequent scaling/shifting as indicated. The sign in the response column indicates the typical LGM ice volume response to an increased value of the parameter based on sensitivity tests or a priori reasoning when straightforward. It should be noted that opposite responses are possible for some of the parameters depending on the whole parameter vector value. "LGM" is last glacial maximum.

| Definition | Parameter | Code name | Response | Range and scaling |
|---|---|---|---|---|
| Weertman coefficient for soft-bed | $C_{\mathrm{sb}}$ (Eq. 10) | rmu | - | $(0.1 \rightarrow 2.0) \times 3$ km/yr $/(30 kPa)^{m_s}$ |
| sliding coefficient for hard bed | $C_{\mathrm{hb}}$ (Eq. 11) | fnslid | - | $(0.1 \rightarrow 4.0) \times 200$ m/yr $/(100 kPa)^{m_h}$ |
| Glen flow law enhancement | $E_f$ | fnflow | - | subranges of $(1.5 \rightarrow 4.5)$ range depending on grid resolution |
| Coulomb-plastic friction coefficient | $C_{\mathrm{Coul}}$ (Eq. 6) | rCfrict | + | $(3.1 \rightarrow 4.5) - 3.0$ else 0 |
| till cover fraction exponent | | fbedpow | | $(0. \rightarrow 1.0) \times 2 + 0.5$ |
| basal drag soft bed subgrid roughness dependency | $C_{\sigma\mathrm{sb}}$ (Eq. 10) | fSTDtill | + | $0.0 \rightarrow 2.0$ m |
| basal drag hard bed subgrid roughness dependency | $C_{\sigma\mathrm{hb}}$ (Eq. 11) | fSTDslid | + | $0.0 \rightarrow 2.0$ m |
| soft bed Weertman sliding exponent | $m_s$ (Eq. 5) | POWbtill | | $NINT((0.0 \rightarrow 1.0) \times 10)$ |
| effective bed roughness scale | $h_{\mathrm{wbCrit}}$ (Eq. 53) | hwbCrit | + | $0.01 \times 10^{(2(0.0 \rightarrow 1.0))}$ m |
| constant bed drainage rate | | rBedDrainRate | + | $10^{(0.0 \rightarrow 1.0)-3}$ m yr$^{-1}$ |
| effective-pressure factor | $C_{\mathrm{Neff}}$ (Eq. 6) | rNeffFact | - | $2 \times 10^{4-(0.0 \rightarrow 1.0)}$ Pa |
| weight of first input deep geothermal heat flux map | | wGF1 | | $0.0 \rightarrow 1.0$ |
| Alternative till cover map weight | | wtBedTill1 | | $0.0 \rightarrow 1.1$ |
| calving coefficient | $C_{\mathrm{calv}}$ (Eq. 27 and 32) | fcalvin | - | $(0.1 \rightarrow 0.9) \times 10$ km/yr |
| hydro-fracturing coefficient | $C_{\mathrm{hydCrk}}$ (Eq. 31) | pfactdwCrack | | $(0.5 \rightarrow 4.0) \times 100$ |
| calving face melt coefficient | $C_{\mathrm{face}}$ (Eq. 25) | CfaceMelt | - | $(0.5 \rightarrow 4.0) \times 10$ |
| sub shelf melt coefficient | $C_{\mathrm{SSM}}$ (Eq. 23) | fSSMdeep | - | $(0.0 \rightarrow 1.0) \times 1.6 + 0.2$ |
| marine freezing point (effective bias adjustment) | $CT_{\mathrm{ssmCut}}$ (Eq. 24 and 25) | TssmCut | - | $(0.0 \rightarrow 1.0) \times -4^{\circ}C$ |
| lacustrine calving parameter | | flac | - | $0.0 \rightarrow 0.4$ |
| shortwave surface melt coefficient | $C_{\mathrm{RadSMB}}$ (Eq. 17) | fRadSMB | - | $(0.2 \rightarrow 0.5) \times 2$ |
| thickness of the lithosphere | $d_{\mathrm{L}}$ | | | $(46 \rightarrow 146)$ km |
| viscosity of the upper mantle | $\eta_{\mathrm{um}}$ | | | $(0.1 \rightarrow 2.0) \times 10^{21}$ Pa s |
| viscosity of the lower mantle | $\eta_{\mathrm{lm}}$ | | | $(1.0 \rightarrow 50) \times 10^{21}$ Pa s |
| **North America and Eurasia specific** | | | | |
| margin chronology weighting | | wmargw | | $0.0 \rightarrow 1.0$ |
| margin forcing ablation threshold | $F_m$ | margbab | - | $0.0 \rightarrow 1.0$ |
| margin forcing accumulation threshold | $F_a$ | margbac | + | $0.0 \rightarrow 1.0$ |
| margin forcing calving reduction factor | $F_c$ | margcalv | - | $0.0 \rightarrow 1.0$ |

**Table 3.** Climate forcing ensemble parameters. Parameter scalings follow the same rules as described in Table 2.

| Definition | Parameter | Code name | Response | Range |
|---|---|---|---|---|
| weight of annual glacial index from ice core records | | wtIndxYr | | $0.0 \rightarrow 1.0$ |
| weight of energy balance climate model (EBM) for glacial index setting | | rWtEBMindx | - | $0.0 \rightarrow 1.0$ |
| weight of EBM temperature field | $wT_{\mathrm{EBM}}$ (Eq. 45) | fTweightEBM | | $0.0 \rightarrow 1.0$ |
| weight of glacially-indexed input GCM 2 meter temperature field | $wT_{\mathrm{PMIP}}$ (Eq. 45) | fTweightPMIP | | $0.0 \rightarrow 1.0$ |
| scaling of EBM temperature field glacial anomaly | $C_{\mathrm{EBM}}$ (Eq. 41) | fnTEBMscale | + | $0.8 \rightarrow 1.25$ |
| global temperature index scale factor | $C_{\mathrm{IT}}$ (Eq. 36) | fnTdfscale | + | $0.85 \rightarrow 1.2$ |
| temperature glacial index exponent | $\Theta_T$ (Eq. 36) | fnTdexp | - | $0.85 \rightarrow 1.4$ |
| LGM temperature EOF weights | $C_{\mathrm{TEOF}}(i)$ (Eq. 43) | fTEOF[NvTEOF] | | $(0.0 \rightarrow 1.0) - 0.5$ |
| LGM vertical air temperature gradient | $L_{\mathrm{LGM}}$ (Eq. 44) | rlapselgm | - | $(0 \rightarrow 1) \times 4 + 4\,^{o}C/\,\mathrm{km}$ |
| glacial index boost where ice cover | $I_{\mathrm{H+}}$ (Eq. 46) | HboostTndx | | $0. \rightarrow 0.2$ |
| weight of glacially-indexed input GCM precipitation field | | fPREweightPMIP | | $0.0 \rightarrow 1.0$ |
| global precipitation scale factor for PMIP component | $C_{\mathrm{pre}}$ (Eq. 47) | fnpre | + | $0.6 \rightarrow 1.8$ |
| LGM precipitation EOF weighting | $C_{\mathrm{PEOF}}(i)$ | fPEOF[NvPEOF] | | $(0.0 \rightarrow 1.0) - 0.5$ |
| precipitation orographic forcing regularization | $\mu_p$ (Eq. 50 and 51) | pREG | | $0.0 \rightarrow 1.0$ m/s |
| coef. for exponential surface temperature dependence of non-PMIP precip | $C_{\mathrm{Tp}}$ (Eq. 48) | hpre | - | $0.0 \rightarrow 1.0\,/^{o}C$ |
| precipitation glacial index phase exponent | $\Theta_P$ (Eq. 47) | fnPdexp | | $0.4 \rightarrow 2.0$ |
| desert elevation control parameter | $h_{\mathrm{Ides}}$ (Eq. 37) | rtdes | | $0.0 \rightarrow 1.0$ |
| desert-elevation exponent | $C_{\mathrm{des}}$ (Eq. 49) | desFac | - | $0.5 \rightarrow 2.5$ /km |
| default desert-elevation cutoff | $h_{\mathrm{des0}}$ (Eq. 49) | des2 | + | $0.0 \rightarrow 2.0$ km |
| ocean temperature glacial index phase factor | $\Theta_T o$ | rToceanPhase | | $0.5 \rightarrow 2.0$ |
| negative glacial index ocean warming enhancement factor | | rToceanWrm | | $0.0 \rightarrow 1.0$ |
| **North American specific** | | | | |
| South central precipitation enhancement | | fmpreSM | + | $0.0 \rightarrow 1.0$ |
| western desert-elevation cutoff | | desW | + | $0.2 \rightarrow 3.0$ km |
| northwestern desert-elevation cutoff | | desNW | + | $0.0 \rightarrow 2.0$ km |
| north-central desert-elevation cutoff | | desNC | + | $0.0 \rightarrow 1.5$ km |
| central desert-elevation cutoff | | desC | + | $0.0 \rightarrow 2.0$ km |
| Foxe Basin/Baffin desert-elevation cutoff | | desF | + | $0.0 \rightarrow 2.0$ km |
| Quebec/Labrador desert-elevation cutoff | | desQ | + | $0.0 \rightarrow 3.0$ km |
| midsouth-central desert-elevation cutoff | | desScN | + | $0.0 \rightarrow 2.4$ km |
| south-central desert-elevation cutoff | | desSC | + | $0.0 \rightarrow 2.0$ km |
| **Greenland specific** | | | | |
| latitudinal ramp width of added Holocene warming | $\Theta_{\mathrm{wrm}}$ (Eq. 44) | yTagDx | | $42. - 40. \times (0.0 \rightarrow 1.0)\,^{o}$ lat. |
| Holocene warming scale | $C\mathrm{HTM}$ (Eq. 44) | fTag | | $0.0 \rightarrow 1.0$ |
| **Eurasian specific** | | | | |
| added regional summer warming scaling for EBM | | rSumPlusEBM | - | $(0.0 \rightarrow 1.0) + 0.5$ |
| British Isles desert-elevation cutoff | | desBA | + | $0.0 \rightarrow 2.0$ km |
| Fennoscandian desert-elevation cutoff | | desFS | + | $0.0 \rightarrow 2.0$ km |
| Barents-Kara desert-elevation cutoff | | desBK | + | $0.0 \rightarrow 2.0$ km |
| **Antarctic specific** | | | | |
| Regional subshelf ocean temperature shift | | TregSSMCut(0:5) | - | $(0.0 \rightarrow 1.0) \times 4\,^{o}C$ |

**Table 4.** Default GSM (non-ensemble) parameters, symbols, and grid specification.

| Definition | Parameter | Value |
|---|---|---|
| Earth radius | $r_e$ | 6370 km |
| Earth mass | $m_e$ | $5.976 \times 10^{24}$ kg |
| acceleration due to gravity | g | $9.81\ ms^{-2}$ |
| water latent heat of fusion | $L_w$ | $3.35 \times 10^5\ J\ kg^{-1}$ |
| ocean water density | $\rho_s$ | $1028\ kg\ m^{-3}$ |
| ice density | $\rho_i$ | $910\ kg\ m^{-3}$ |
| ice specific heat capacity | $c_i(T)$ | $(152.5 + 7.122 \cdot T)\ J\ kg^{-1}\ K^{-1}$ |
| ice thermal conductivity | $k_i(T)$ | $9.828 \cdot \exp(-0.0057 \cdot T)\ W\ m^{-1}\ K^{-1}$ |
| bedrock density | $\rho_b$ | $3300\ kg\ m^{-3}$ |
| bedrock specific heat capacity | $c_b$ | $1000\ J\ kg^{-1}\ {}^{o}C^{-1}$ |
| bedrock thermal conductivity | $k_b$ | $3\ W\ m^{-1}\ {}^{o}C^{-1}$ |
| number of ice dynamic levels | $nz_i$ | 12 |
| number of ice thermodynamic levels | $nz_{Ti}$ | 65 |
| number of bed thermodynamic levels | $nz_{Tb}$ | 16 |
| Glen flow law coefficient, $T < -10^o$ C | $A_c$ | $8.9836 \times 10^{-6} \mathrm{Pa}^3 \mathrm{yr}^{-1}$ |
| Glen flow law coefficient, $T > -10^o$ C | $A_w$ | $7.43377 \times 10^5 \mathrm{Pa}^3 \mathrm{yr}^{-1}$ |
| creep activation energy of ice, $T < -10^o$ C | $Q_c$ | $6 \times 10^4 \mathrm{Jmol}^{-1}$ |
| creep activation energy of ice, $T > -10^o$ C | $Q_w$ | $1.15 \times 10^5 \mathrm{Jmol}^{-1}$ |
| Glen flow law exponent | $n$ | 3 |
| minimum till friction angle | $\phi_{min}$ | $10^o$ |
| maximum till friction angle | $\phi_{max}$ | $30^o$ |

## 2.3 A caveat on parametrizations in the GSM

Given the breadth of applications the GSM has, or is, being used for (all last glacial cycle ice sheets from Icelandic to Antarctic),

and the dimension of the ensemble parameter space and range of climate inputs used; there is no such thing as an optimal

parameterization. A further complication is the over two decades of continuous development. Optimal fits from earlier GSM

versions may no longer be optimal given changes in input topographies, climate inputs, etc... As such, the approach has been to

combine physical reasoning, parametric forms from the literature, and broaden degrees of freedom across various components

to albeit incompletely convert process uncertainties into ensemble parameter uncertainties.

## 2.4 Ice dynamics

The hybrid SSA/SIA solver was imported from Pollard and DeConto (2012) and thereby uses the identical finite difference

discretization. This discretization naturally imposes the appropriate stress balance boundary condition for a floating ice margin

(cf. eg. Cuffey and Paterson, 2010; Winkelmann et al., 2011). Aside from conversion to F90 standard, the main change after import was the insertion of a sequence of matrix solver options for the SSA equations in case of convergence failure. First, a

150 biconjugate gradient squared solution (BCGS option for the NSPCG solver) is attempted. Upon failure, a generalized minimal residual (GMRES) solution is subsequently attempted. Upon further failure, successive over relaxation (SOR) will be tried. The first two options use a symmetric successive over-relaxation preconditioner (SSOR). If convergence failure persists, the GSM steps back to the beginning of the last dlong interval (default 100 years) and the time-stepping is repeated with half the ice dynamical time-step ($delt$). The short time-step is retained for at least three hundred years and then reverts back to the

155 previous value when permitted by the CFL (Courant–Friedrichs–Lewy) criterion.

The solution of the hybrid SSA/SIA equation involves an outer Picard loop (1:NITERC) for ice thickness and a sequence of two inner loops (1:NITERA) to solve the SSA elliptic equations for the horizontal ice velocities (cf appendix A1), first without the grounding line flux condition, then with it. Default convergence thresholds are $0.5\%$ and 2 m/yr respectively for the maximum grid cell residuals between successive iterations. For the default compile flags, the outer Picard iteration is

160 subject to $10\%$ damping of the first iteration (-DNumDamp) and to the unstable manifold correction of Hindmarsh and Payne (1996) for all but the last iteration (-DHPiterc). Sensitivities to these numerical compiler flag options for example GRIS and AIS glacial cycle simulations are in appendix B.

The default choice for the stopping criteria for the SSA elliptic matrix solution (say of the form $\mathbf{Au} = \mathbf{b}$) is the relative norm (option 10 of the NSPCG package, Kincaid et al., 1989) of the left pre-conditioned residual $\mathbf{z}$ (for preconditioner matrix that

can be split into left and right components $Q = Q_L Q_R$):

$$\left[ \frac{||\mathbf{z}||}{||\mathbf{Q_L^{-1}b}||} \right] < \zeta \tag{1}$$

where $z$ is the current left-preconditioned residual:

$$\mathbf{z} = \mathbf{Q_L^{-1}}(\mathbf{b} - \mathbf{Au}) \tag{2}$$

Convergence thresholds ($\zeta$) are a function of the iteration, with final iteration thresholds of $1 \times 10^{-5}$ or smaller.

In addition to the default grounding line ice flux treatment of Schoof (2007) for Weertman type sliding, we've added a Coulomb-plastic option from Tsai et al. (2015). The grounding-line ice thickness for the flux calculation is determined via a subgrid interpolation as in Pollard and DeConto (2012). The treatment of 2D buttressing effects on grounding line fluxes in the GSM has been revised as per Pollard and DeConto (2020). This addresses limitations of the original (Pollard and DeConto, 2012) approach when compared against the results of an Antarctic ice sheet model with a highly resolved computational mesh

around grounding lines (Reese et al., 2018).

The only other significant ice dynamical differences from Pollard and DeConto (2012) are the specification of basal drag and basal sliding activation as detailed below (section 2.5) and a correction to handle a floating ice margin that subsequently becomes grounded on what was previously ice free terrestrial land (an atypical situation, that was found to occur in Northern Baffin Bay for some GSM glacial cycle simulations).

The GSM uses a default Glen flow law (exponent 3 stress dependence) ice rheology (Glen, 1952; Cuffey and Paterson, 2010). However recent work has favoured an exponent 4 for ice sheet contexts (Fan et al., 2025), though curiously it has

chosen to ignore evidence for grain boundary sliding being the rate-limiting process with exponent 1.8 for typical ice sheet stress regimes (Goldsby and Kohlstedt, 2001; Peltier et al., 2000). Furthermore, for temperate ice with $> 0.6\%$ liquid water content, laboratory experiments indicate a linear viscous rheology dominates due to diffusion creep (Schohn et al., 2025). To address this uncertainty, the GSM flow law exponent is a free parameter with a compile flag option (-DPOWiceEns) to convert it to an ensemble parameter. However, this will entail user coding of an appropriate temperature dependent flow coefficient for the chosen exponent.

The default Glen flow law dependence on ice temperature follows the recommended values of Cuffey and Paterson (2010). An ensemble parameter ($E_f$) provides the default flow enhancement. The basal ice layer in the model is given an extra $50\%$ enhancement to partly capture the observed lower effective viscosity of older basal ice (Cuffey and Paterson, 2010). As well, the enhancement in the upper half of the ice column is reduced by $50\%$ to partly account for the reduced fabric development in younger ice (Cuffey and Paterson, 2010).

Ice flow enhancement in the GSM also partially accounts for anisotropic effects from fabric development in polar ice (Ma et al., 2010). This fabric development tends to stiffen the ice with respect to horizontal strain. As the traditional Glen flow law enhancement factor in good part represents the enhancement in vertical shear due to this fabric development, it stands to reason that for horizontal strain, this factor should take on some sort of inverse relation to the SIA enhancement. Invoking Occam's razor to give a functional form of $E_{shelf} = A/E_{SIA} + B$, along with the requirement that the SSA enhancement $E_{shelf}(E_{SIA} = 1) = 1$ and $E_{shelf}(E_{SIA} = 5.6) = 0.6$ from Ma et al. (2010), the SSA enhancement factor for ice shelves is therefore

$$E_{shelf} = 0.48696/E_f + 0.51304 \tag{3}$$

For ice streams, Ma et al. (2010) recommend a value of 1 at the onset, and the ice shelf value at the grounding line. To avoid the required relative position tracking, an average value of

$$E_{stream} = 0.5 \cdot (E_{shelf} + 1.0) \tag{4}$$

is applied. The above ignores uncertainties in the relation between SIA and SSA flow enhancements, and therefore warrants future investigation. However for now we judge that such uncertainties are swamped by other relevant sources, especially those due to climate forcing controlling surface and basal (for ice shelves) mass-balance.

## 2.5  Basal drag

Given the uncertainties in the appropriate form of the large scale basal drag law for soft bed (*e.g.,* Fowler, 2003), the GSM has both Weertman power law and Coulomb plastic options for soft-bedded basal drag. For the Weertman case, the effective basal sliding law (for both hard or soft beds) is given by (*e.g.,* Weertman, 1957; Cuffey and Paterson, 2010):

$$\boldsymbol{U_b} = C_b |\boldsymbol{\tau_b}|^{m_b - 1} \boldsymbol{\tau_b} \tag{5}$$

with basal drag $\boldsymbol{\tau_b}$ and basal velocity $\boldsymbol{U_b}$. $C_b$ incorporates a basal temperature ramp for sliding activation (*c.f.* section 2.5.2). $C_b$ also accounts for: i) potential subgrid warm based conditions in topographic lows, ii) bed type (soft or hard), and iii) drag

reduction under pinned shelf conditions (as detailed below). The exponent ($m_b$) is generally treated as an ensemble parameter
($m_b = m_s$ in Table 2) when the bed has deforming till cover given the range of inferred values in the literature (*e.g.,* Gillet-Chaulet et al., 2016; Maier et al., 2021).

The -DNeffDRAG compile flag combined with any form of basal hydrology imposes basal effective pressure dependence on the basal drag. When activated, the Weertman basal sliding coefficient is multiplied by the following to give a regularized form of the traditional basal effective pressure dependence (*c.f. e.g.,* page 240 of Cuffey and Paterson, 2010):

$$\min(10., \max(0.2, \frac{C_{\mathrm{Neff}}}{N_{\mathrm{eff}} + N_{\mathrm{reg}}})) \tag{6}$$

where $N_{\mathrm{eff}}$ is the computed effective basal pressure (cf section 2.13), $C_{N_{\mathrm{eff}}}$ is an ensemble parameter scaling coefficient, and the regularization parameter $N_{\mathrm{reg}}$ has value 10 kPa.

Based on the results of a basal drag inversion for Greenland (Maier et al., 2021), the hard bed has a default power law exponent $m_b = 4$ but otherwise has the same form of Weertman type sliding law as for soft-bedded Weertman (eq. 5). This sliding exponent value is in agreement with recent evidence in favour of an exponent 4 stress dependence for ice flow (Fan et al., 2025) if basal sliding is dominated by ice deformation around obstacles. Given the high statistical confidence in the results of the above inversion for the 4 (of 8 total) catchments with mostly strong (hard) beds, we tentatively assume this value is appropriate for all other paleo ice sheets. If there was evidence or judgment to the contrary, turning this exponent into an ensemble parameter would be trivial.

For Coulomb plastic basal drag, a regularized form that accounts for cavitation (similar to that of Schoof, 2005; Joughin et al., 2019) has been found to have better numerical convergence:

$$\boldsymbol{\tau_b} = \max\left(1kPa\ ,\ N_{\mathrm{eff}}\ C_{\mathrm{Coul}}\ \tan(\phi_t)\ \left(\frac{\boldsymbol{U_b}^2}{\boldsymbol{U_b}^2 + U_{\mathrm{sqReg}}}\right)^{1/6}\right) \tag{7}$$

where $C_{\mathrm{Coul}}$ is a drag coefficient, $U_{\mathrm{sqReg}}$ is a regularization velocity term ($(20\,m/yr)^2$) and $\phi_t$ is the elevation dependent friction till angle (as per Maris et al., 2014) to account for the increased prevalence of saturated fine sediment cover in marine sectors:

$$\phi_t = \begin{cases} \phi_{min} = 10^o & h_{\mathrm{bG}} \leq -10^3, \\ \frac{-h_{\mathrm{bG}}}{10^3} \cdot \phi_{min} + (1 + \frac{h_{\mathrm{bG}}}{10^3}) \cdot \phi_{max} & -10^3 < h_{\mathrm{bG}} \leq 0, \\ \phi_{max} = 30^o & h_{\mathrm{bG}} > 0 \end{cases} \tag{8}$$

where $h_{\mathrm{bG}}$ is the bed elevation relative to contemporaneous sea level. This formulation uses an appropriate linearization around the previous value of the basal velocity ($U_b^*$) in the iterative solution of the SSA velocity equation:

$$\boldsymbol{\tau_b} = \boldsymbol{\tau_b^*} + \frac{\partial \boldsymbol{\tau_b^*}}{\partial \boldsymbol{U_b^*}} \cdot (\boldsymbol{U_b} - \boldsymbol{U_b^*}) \tag{9}$$

If the computed Coulomb plastic basal drag is greater than the Weertman basal drag (pre-computed using an SIA approximation for drag law selection only), then the latter basal drag law is used instead. This has both a physical motivation (at

high effective pressure and warm-based conditions, Weertman sliding is plausible, *e.g.,* Tsai et al., 2015), and a numerical motivation (ensuring the basal drag is never larger than the sum of remaining horizontal stresses).

The Coulomb plastic option is more numerically unstable, and as such, a high exponent Weertman law (*e.g.,* exponent 7) is recommended in lieu when computational resources are a limiting factor.

### 2.5.1 Basal drag geological and subgrid topographic controls

The GSM requires a specification of the fractional soft bed cover for each grid cell, either as a constant input (Table 6) or dynamically determined (section 2.14). The determination of soft/hard bed is set according to whether the fractional soft bed cover of the grid cell is above or below GSM parameter SEDCUT (default 0.5). A future improvement will be the inclusion of fractional basal drag from both hard/soft bed components. As the basal drag is computed at grid cell interfaces, the sediment fraction at the interface must be set. This is taken as the square root of the product of adjacent sediment cover fractions in partial accord with a self-consistent treatment for setting diffusion coefficients in a discretized linear diffusion process (the square root operation was chosen to provide an intermediate between an arithmetic mean and the appropriate harmonic mean, cf Patankar, 1980).

One to date unresolved issue is how to deal with fractional and thin till cover as well as the impact of different classes of sediments. The default approach in the GSM is to set the local sediment fraction coefficient (sedF in eqs. 10 and 11) to the minimum of $1.5$ and $2\times$ the input sediment fraction for regions that are presently marine and otherwise to the input value raised to the power of the ensemble parameter fbedpow. This is intended to crudely account for the likely lower drag from marine muds and otherwise provide some ensemble parametric control.

Another unresolved issue for basal drag is the appropriate accounting for the impact of subgrid bed roughness, especially for typical paleo ice sheet model grid resolutions of 10 km or more. Presumably a rougher bed will increase basal drag, with bedrock exposures acting as pinning points. However an opposing mechanism could also be argued with a rough hard bed promoting the trapping of soft subglacial sediment. Past studies of the impact of bed roughness on basal drag typically only consider metre or less bed roughness (*e.g.,* Gagliardini et al., 2007; Wilkens et al., 2015), Though there have been some detailed relationships proposed based on single basin scale analysis (*e.g.,* Li et al., 2010), their validity for continental scale applications are unclear and furthermore their data input requirements (metre scale bed topography) are unlikely to be met for the global ice sheet context in the foreseeable future. To address some of these uncertainties, in addition to ensemble separate parameter sliding coefficients for hard and soft beds ($C_{\mathrm{sb}}$ and $C_{\mathrm{hb}}$), two ensemble parameters ($C_{\sigma\mathrm{sb}}$, $C_{\sigma\mathrm{hb}}$) impose Weertman basal drag dependencies on the subgrid standard deviation of bed elevation ($\sigma_b$ in m). For soft beds, this takes the form of a basal sliding coefficient:

$$C_B = C_{\mathrm{sb}} \cdot \mathrm{sedF} \cdot \min(1, \max(0.2, C_{\sigma\mathrm{sb}}/(0.01\,\sigma_b))) \tag{10}$$

For hard beds, an adhoc term accounting for the subgrid fraction of soft bed cover (sedF) is also included:

$$C_B = C_{\mathrm{hb}} \cdot (1. + 20.\,\mathrm{sedF}) \cdot \min(1.0, \max(0.1, C_{\sigma\mathrm{hb}}/(0.01\,\sigma_b))) \tag{11}$$

The GSM has not been setup for present-day inversion of a basal drag map for existing ice sheets. For paleo contexts, such inversions are problematic given the confounding impacts of changes in basal water pressure and basal sediment thickness. Nor can such inversions provide a value where the bed is currently frozen. There is a need for the development of robust basal drag parametrizations that can be applied to all paleo ice sheets, be it for regions that are presently subglacial, marine, or subaerial.

A key issue for ice shelf modelling is the presence of potential subgrid pinning points under the ice shelf that aren't presently active. This is a significant source of uncertainty given the lack of detailed topographic data for the subshelf environment. To partly address this, the model has a standard option of assuming a Gaussian distribution of subgrid pinning points based on a map of the standard deviation of the subgrid bed elevation. For poorly observed regions, adjacent open marine environments can provide an estimate when creating these maps. The pinning point effect is simply imposed as a fractional coefficient (fpin< 1) that multiplies the basal drag derived as if the shelf was grounded. fpin is set to the cumulative normal distribution (more exactly an analytical approximation thereof) for subgrid elevation above the distance between the bed and ice shelf base. For distances of more than 3 standard deviations of basal roughness, fpin is effectively set to 0 ( an unpinned ice shelf with a very small basal drag (0.001 Pa/(m/yr)) to avoid singularities in the stress balance matrix).

### 2.5.2 Basal sliding activation

A key issue that most ice sheet models do not explicitly address is the appropriate activation function for basal sliding as warm-based conditions are approached. A detailed resolution scaling analysis of this issue has recently been published (Hank et al., 2023), and its recommended activation function is under compile flag choice (-DTrampScN). Briefly this is implemented via an estimated warm-based fraction of a grid cell $F_{\mathrm{warm}}$ (also indirectly accounting for sub-temperate sliding, eg. Fowler, 1986):

$$F_{\mathrm{warm}} = \max\left[0, \min\left(1, \frac{T_{\mathrm{bp,I}} + T_{\mathrm{ramp}}}{T_{\mathrm{ramp}}}\right)\right]^{T_{\mathrm{exp}}}, \tag{12}$$

where $T_{\mathrm{bp,I}}$ is the grid cell interface basal temperature relative to the pressure melting point. $T_{\mathrm{ramp}}$ is the temperature interval for which the grid cell has some warm-based subgrid ice and $T_{\mathrm{exp}}$ is the exponent used for the ramp. On the basis of numerical experiments, resolution dependence is minimized for values of $T_{\mathrm{exp}}$ between 5 and 10 (Hank et al., 2023), with the GSM having a default value of 10. This ramp depends on the subgrid standard deviation of elevation ($\sigma_{hb}$, in metres) given that a higher standard deviation can increase the subgrid fraction at the pressure melting point when the nominal grid cell basal temperature is below the pressure melting point. As such and with explicit dependence on grid cell resolution ($\Delta xy$), $T_{\mathrm{ramp}}$ is given by:

$$T_{\mathrm{ramp}} = \max(1.0 \, , \, 0.02 \, \sigma_b) \cdot \frac{\Delta xy}{50 \text{ km}} \, ^{\circ}\mathrm{C} \tag{13}$$

This choice of resolution dependence (as determined in Hank et al., 2023) leads to a sharper temperature ramp for finer horizontal grid resolutions, as would be expected on physical grounds (since the range of subgrid basal temperatures for a grid cell, when not fully warm-based, will generally be larger for a larger grid cell). The subgrid warm-based fraction $F_{\mathrm{warm}}$ then

enters into the basal drag coefficient $C_b$ (cf eq 5) as following:

$$C_b = \max(F_{\text{warm}} \cdot C_B , C_{\text{froz}}) . \tag{14}$$

$C_{\text{froz}}$ is the fully cold-based sliding coefficient for numerical regularization:

$$C_{\text{froz}} = 2 \cdot 10^{-4} \text{ m yr}^{-1} \left(5 \cdot 10^{-6} \text{ Pa}^{-1}\right)^{m_b} . \tag{15}$$

The detailed analysis of ice sheet model sliding activation specification in Hank et al. (2023) focused on surge cycling in
an idealization of Hudson Bay and Hudson Strait. It did not consider other paleo ice sheets. For an example GRIS simulation, the ice volume response to the width of $T_{\text{ramp}}$ in eq. 12 is very strong (Fig. 2), especially when compared to the minimal response to other numerical compiler flags (Fig. B1). It is also one of the more sensitive numerical flags for an example AIS simulation (Fig. B2). This further underlines the importance of a numerically and physically justified specification of basal sliding activation.

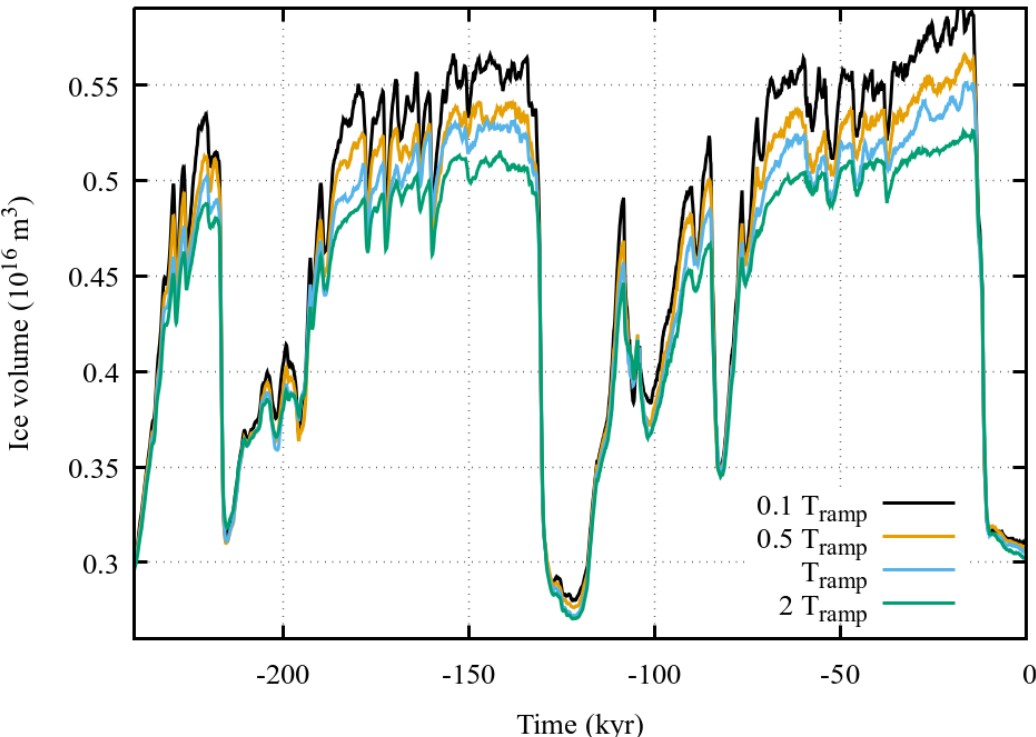

**Figure 2.** Example GRIS ice volume history sensitivity to the width of the basal sliding activation temperature ramp ($T_{\text{ramp}}$) in eq. 12. The simulations use the default $0.5^o$ by $0.25^o$ (longitude, latitude) resolution.

Another non-trivial issue for ice sheet models with continental scale grids is the appropriate determination of the basal interface temperature. The preferred approach in the GSM accounts for the potential warming at the warm-cold interface by refreezing of subglacial meltwater. This is approximated (with -DTbpmGbI) by using half of the latent heat flux embodied in subglacial meltwater generated by the two grid cells bordering the cell interface under question. This latent heat is distributed across the basal ice dynamical layer to convert to a temperature increment and added to the interpolated basal temperature at 320  the interface. Hank et al. (2023) provides a detailed description and comparison of this and alternative treatments for computing the basal temperature at the grid cell interface.

## 2.6   Ice thermodynamics and permafrost resolving bed thermodynamics

The GSM finite volume thermodynamic scheme uses an implicit solution for the vertical and local components and explicit time-stepping solution for the horizontal advection component of the energy conservation equation:

$$325 \quad \rho_i c_i(T(\mathbf{r}))\frac{\partial T(\mathbf{r})}{\partial t} = \frac{\partial}{\partial z}\left(k_i(T(\mathbf{r}))\frac{dT(\mathbf{r})}{dz}\right) - \rho_i c_i(T(\mathbf{r}))\mathbf{V}(\mathbf{r})\cdot\nabla T(\mathbf{r}) + Q_d(\mathbf{r}) \tag{16}$$

where $\mathbf{V}(\mathbf{r})$ is the 3D ice velocity and coefficient values are listed in Table 4. Heat source terms include full SSA and SIA contributions to deformation work ($Q_d$) and the boundary heat flux from basal sliding ($\tau_\mathbf{b}\cdot\mathbf{u_b}$). The coefficients with the $i$ subscript denote ice: density ($\rho_i$), heat capacity ($c_i$), and thermal conductivity ($k_i$).

    Horizontal advection is discretized using a second-order interface consistent 3 pt upwinding. Vertical diffusion and advec- 330  tion is solved implicitly using the power-law finite volume treatment of Patankar (1980). Interface diffusivities are based on geometric means as per ibid.

    The discretization of the energy conservation equation for the basal ice grid cell is non-standard to enable a solution of the temperature at the basal interface as needed for accurately determining thermal activation of basal sliding. To do so, horizontal advection and the time derivative in eq. 16 use a basal grid-cell centre temperature that is determined via linear interpolation 335  between the basal interface temperature and cell centre temperature of the vertically adjacent ice grid cell. On the other hand, solution of the basal interface temperature means that no interpolation is required for vertical diffusion and advection.

    For thin ice ($H < 50$ m), basal temperature is set to that of the highest elevation neighbouring grid cell with ice thickness $> 50$ m, and ice temperature is linearly interpolated in the vertical from the bed to surface. If there is no appropriate neighbour, the whole ice column is set to mean annual surface temperature. For floating ice, the basal temperature is set to the pressure 340  melting point.

    Unlike many older generation ice sheet models, energy is conserved when a grid-cell reaches the pressure-melting point. This is accomplished via an extra iteration within the tri-diagonal solution of the implicit energy conservation solution. The residual heat is then used for basal melt except for floating marine conditions for which a submarine melt and refreezing model is active (section 2.7.2). The vertical implicit solution is over the whole ice and bed grid. Thermodynamic time-stepping is 345  subject to horizontal CFL constraints using time-interpolated horizontal ice velocities. Though the vertical solution is implicit, this does not mean that the solution will have no time-step sensitivity. As such, the solver includes a vertical sub-iteration to restrict the time-step for any single vertical column to a set factor of the CFL stability threshold. This sub-iteration time-step

factor has a default value of 10 (chosen on the basis of sensitivity tests) but is adjusted as needed to impose a maximum of 100 sub-iteration time-steps.

Given the grid cell dimensions for large ice sheet glacial cycle contexts (and generally much lower horizontal temperature gradients relative to vertical temperature gradients), the default bed thermodynamics configuration assumes vertical diffusive heat transport only (as supported by a straight-forward scale analysis). A near unique feature is that the bed thermal model accounts for permafrost via a standard heat capacity approximation (Osterkamp, 1987; Williams and Smith, 1989; Mottaghy and Rath, 2006). It also applies temperature forcing corrections at the top of subaerial frozen ground to partly account for the

effects of seasonal snow cover and surface vegetation (Smith and Riseborough, 2002). The GSM thermal bed has a default depth (GSM parameter BEDTdepth) of 4 km for which the lower flux boundary condition is specified by an input map (section 2.17).

The GSM has the option (-DthreeDbedTdiffusion) of added explicit time-stepping horizontal bed thermal diffusion. This would be more appropriate for grid resolutions of 10 km or finer or for regions where there are large horizontal gradients in the

input deep geothermal heat flux field. However, the available reconstructions for this boundary condition are somewhat vague as to the exact depth they represent, often self-described as being near the bed surface. These reconstructions will already embody horizontal heat diffusion up to their representative depth. If this depth is above the chosen (default 4 km) depth of the bed thermal model, activation of GSM horizontal heat diffusion would effectively result in erroneous doubling of horizontal heat diffusion over the depth of overlap.

The activation of horizontal diffusion is computationally inexpensive (about a $2\%$ increase in run time). For a coarse 40 km grid resolution Antarctic two glacial cycle simulation, its addition can alter root-mean-square-error discrepancies with present day input ice topography and observed marginal ice velocities by approximately 10 m and 45 m/yr respectively.

A comparison of results for an older (SIA only) version of the GSM (but with the same bed thermal model) against North American deep borehole temperature profiles along with a full description of the bed thermal model are in Tarasov and Peltier

(2007).

The default coupling between GSM ice dynamics and thermodynamics is explicit with a minimum one year time-step. However, the GSM includes an option for an iterative implicit coupling solution (-DimplicCoupleDynTherm). The implicit coupling iteration is for each ice dynamical time-step. It is subject to a chosen convergence threshold for both maximum ice thickness and horizontal velocity component differences between successive iterations.

## 2.7   Mass balance processes

Mass balance process representation was chosen based on space-time resolution of required inputs and associated uncertainties.

### 2.7.1   Positive degree day and positive temperature insolation surface melt (PDDsw) and refreezing

The GSM uses a novel extension of the classical positive degree day (PDD) scheme that accounts for the changing short wave (SW) component of the surface energy-balance. PDD schemes (*e.g.,* Cuffey and Paterson, 2010) traditionally use two

constant melt coefficients to crudely account for the changing albedo between ice and snow. However, it is well known that ice

and snow albedos continuously vary. Furthermore, experiments with full surface energy balance models have made clear that orbital changes in short-wave forcing significantly affect surface mass-balance (van de Berg et al., 2011). From a physical point of view, PDD's are effectively a way to account for the long-wave, latent heat, and sensible heat flux components of surface energy balance (as all these fluxes depend on air temperature), but they do not account for variations in net short-wave fluxes beyond the binary choice of snow and ice PDD melt coefficients.

Observationally, fitted PDD melt coefficients vary over a wide range (both spatially and seasonally, *e.g.*, Braithwaite, 1995; Hock, 2003). We ascribe these variations in large part to changing mean net SW inputs and therefore choose a near lowest observationally-inferred value 3.3 mm/PDD (ice equivalent) for a single PDD coefficient (*e.g.*, Braithwaite, 1995; Hock, 2003) to capture the non-SW energy flux components. This value is a bit larger than that which would be inferred on the basis of pure long-wave and sensible energy balance to account for latent heat contributions.

For the shortwave component, a key challenge is that the short-wave input only contributes to surface melt if the surface temperature is at $0^o$ C. This constraint is often accounted for in present-day contexts for which hourly temperature and surface energy flux observations from automatic weather stations are available (*e.g.*, Irvine-Fynn et al., 2014). However, for paleo ice sheet modelling contexts, typically only monthly mean temperature climatologies are available. As such, short of the few coupled ice sheet and climate models able to do full energy balance calculations (*e.g.*, Krapp et al., 2017; Willeit et al., 2022), this constraint has not been applied in paleo ice sheet modelling contexts.

A possible computationally efficient solution to imposing this constraint arises from the similarity of the above temperature threshold to that of the contribution of PDDs to surface melt. Just as PDDs are computed for paleo modelling contexts based on a probabilistic distribution around mean monthly temperatures (*e.g.*, Tarasov and Peltier, 1997a; Wake and Marshall, 2015), a positive temperature time integrated surface insolation flux may also be computed. This requires an assumption relating near surface air temperature to actual snow/ice surface temperature. Though not identical we assume that on a time integrated basis, errors resulting from imposing the $0^o$ C constraint on air temperature (as opposed to surface temperature) are relatively minor compared to other sources of error. The GSM uses a statistical model for the shortwave insolation for 2 meter air temperature above $0^o$ C ($S_{wrm}$) as a function of mean monthly: solar insolation, number of PDDs per day (PDDd), and standard deviation of air temperature $\sigma_{T_{2m}}$. The model was derived from regression of mid to high latitude 4 hourly insolation and 2 meter air temperatures from the PLASIM GCM (Fraedrich, 2012) over a deglacial transient run (Andres and Tarasov, 2019) and takes the form:

$$S_{wrm} = \sqrt{\min(1.0, C_{\mathrm{RadSMB}} \cdot \mathrm{PDDd}/\sigma_{T_{2m}})} \cdot (1 - \mathrm{albedo}) \cdot (\text{mean monthly surface insolation}) \tag{17}$$

This regression captures much of the source GCM data signal (Fig. 3) with residual differences likely dominated by the lack of accounting for variations in cloud cover. The GSM surface insolation solution also accounts for orbital dependence and atmospheric transmissivity dependence on mean monthly solar angle (using the formulation of Irvine-Fynn et al., 2014).

To partially address the sensitivity to unresolved cloud cover, a cloud radiative transmissivity factor (GSM parameter Cloud-Factor) enables ice sheet scale adjustments. This factor is currently set to 0.7 with one exception. Based on initial ensemble

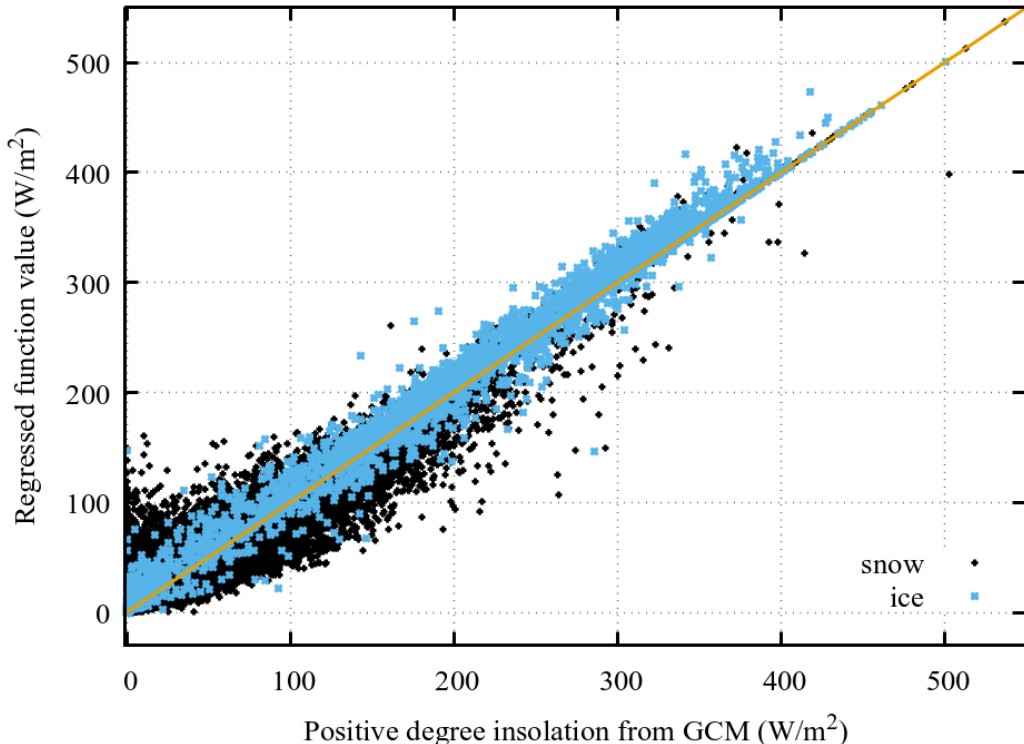

**Figure 3.** Comparison of GCM computed and regressed function used in GSM for monthly mean of positive degree daily surface insolation. Results are disaggregated for snow and ice surfaces for ensemble parameter $C_{\mathrm{RadSMB}}$ in eq. 17 set to its nominally regressed value of 0.345.

modelling and motivated by the high observed frequency of thick cloud cover, the factor for Iceland is set to 0.4. The $C_{\mathrm{RadSMB}}$
ensemble parameter (eq 17) also provides an ice sheet scale ensemble parameter to partly address remaining uncertainties.

Snow surface albedo (using the recommended -DalbT2m compile flag) is a continuous function (Gabbi et al., 2014) of the nominal daily maximum 2 m air temperature ($T_{2\mathrm{max}}$). This is approximated as a function of mean monthly 2 m air temperature ($\overline{T_{2m}}$) and standard deviation ($\sigma_{\mathrm{T2m}}$) thereof:

$$T_{2\mathrm{max}} \quad = \quad \max(1.0, \overline{T_{2m}} + 1.4 \cdot \sigma_{\mathrm{T2m}}) \tag{18}$$

$$\mathrm{albedo} \quad = \quad 0.86 - 0.155 \cdot \log(T_{2\mathrm{max}}) \tag{19}$$

Ice surface coalbedo (1−albedo) is 2.8 times snow surface coalbedo. Firn albedo is set to the average of the snow and ice albedos.

To date, it has been common for paleo ice sheet models to determine PDDs as a function of mean monthly temperature assuming a Gaussian distribution with constant standard deviation (*e.g.,* Tarasov and Peltier, 1997a; Albrecht et al., 2020b).

However, an examination of hourly temperature data from Greenland stations indicates this to be quite inaccurate (Wake and Marshall, 2015). As such, for computing PDDs, the GSM uses a observationally-fitted non-Gaussian distribution as a function of mean monthly temperature that was tested for various sites across Greenland, Norway, and Antarctica (Wake and Marshall, 2015). This distribution has skewness and kurtosis with linear dependence on mean monthly temperature, and quadratic dependence for the standard deviation. When coupled to full climate models, the GSM can instead take the monthly

grid-cell standard deviation from the climate model.

The GSM surface melt model contrasts with insolation-temperature melt models (*e.g.,* Robinson and Goelzer, 2014) implemented with the assumption that snow melt is a linear function of daily mean temperatures and that explicitly ignore the fraction of daily insolation required to bring the ice surface temperature to the melting point. The arguably largest sources of uncertainty for any paleo surface melt model will be errors in accounting for variations in hourly temperature, atmospheric

transmissivity (especially due to cloud cover), and surface albedo.

The GSM uses a surface meltwater refreezing scheme that approximately accounts for firn meltwater retention and available refreezing potential. In detail, the model sets the thickness of annual superimposed (refrozen) ice (supice, with effective sum over repeated melt/refreeze in a year) to

$$H_{\text{act}} \quad = \quad \min( \text{dFRZ}, H + 0.5 \cdot \text{accum}_{\text{year}}) \tag{20}$$

$$\text{supice} \quad = \quad \min(H_{\text{act}} \cdot C_{\text{ice}}/L_{\text{ice}} \cdot \text{NDY}, \text{total snow melt and rain over year}, 1.6 \, \text{accum}_{\text{year}} ) \tag{21}$$

where $NDY$ is the mean number of negative degree years computed in a similar approach to PDDs (or equivalent to $\text{PDD}/365 - T_{\text{2mmeanyear}}$). The maximum thermodynamically active depth dFRZ is set to 3.675 m (ice equivalent) based on loose tuning to present-day RACMO2.3p2 results for Greenland (Noël et al., 2018) and respecting bounds in Reijmer et al. (2012). The first term in eq 21 sets the available freezing potential (as per Huybrechts and de Wolde, 1999), the second

term is the available supply of water for refreezing, and the third term the available pore space for trapping meltwater (set to the maximum modelled value for present-day Greenland for both RACMO2 and MAR Regional Climate Models (RCM) in Reijmer et al., 2012). This parameterization was chosen based on the near best fits after retuning of the Huybrechts and de Wolde (1999) approach in Reijmer et al. (2012) with the added pore trapping condition based on the results shown in this refreezing model comparison. The slight retuning of dFRZ from the value in Reijmer et al. (2012) was necessitated by the use

of the more physical $NDY$ factor as opposed to the mean annual temperature used by Huybrechts and de Wolde (1999). Given the tuning against Greenland RCM modelling, the applicability of this scheme and its current parameters to other ice sheets is unclear. It is likely reasonably applicable for other similar maritime proximal ice sheet ablation zones on the basis of climatic similarity. As such, RCM modelling of continental ablation zones would be a priority for testing/refinement of this scheme. Unfrozen meltwater will also be retained in any ice surface grid-cell scale depressions when the surface hydrology solver is

active in the GSM.

For further partial validation, the combined GSM melt and refreezing scheme (with corresponding default ensemble parameter value $C_{\text{RadSMB}} = 0.33$ in eq. 17) is relatively unbiased over ice in comparison to the results of a regional climate model (MARv3.5.2, Fettweis et al., 2017) for the GRIS (cf Fig. 4). The GSM has an overall negative bias (*i.e.* insufficient melt)

for melt over all surfaces for larger values ($> 1$ m/yr) compared to that of the climate model. It is unclear to what extent this discrepancy as well as the scatter in eq. 17 is attributable to the previously discussed sources of uncertainty that would also apply to all climate models, especially the hard to accurately predict cloud cover.

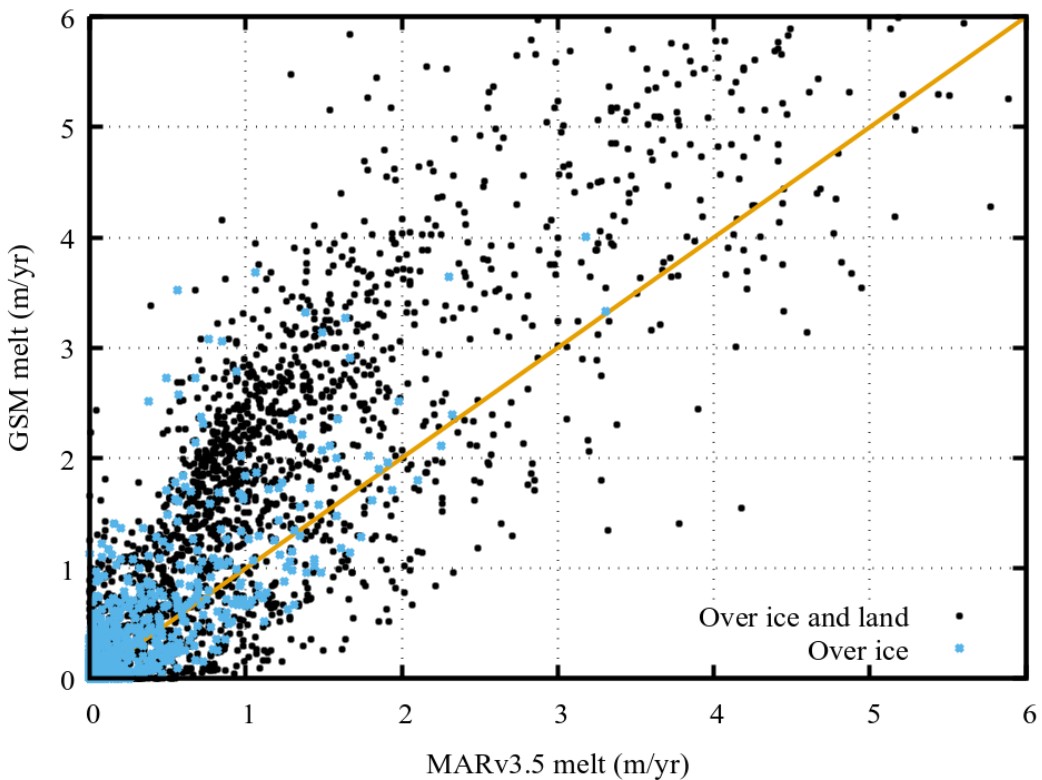

**Figure 4.** Comparison of net melt (total melt - refreezing) between the GSM and the results of 20 km resolution regional climate modelling (1879:1999 mean climatology, MARv3.5.2, Fettweis et al., 2017) for the present-day GRIS. The GSM used the mean monthly precipitation and 2-meter temperature climatologies from the latter.

The determination of monthly mean rain/snow fraction uses the monthly mean positive degree fraction for near surface air temperature. To better reflect that this fraction tends to be physically determined well above the surface, this fraction is computed relative to PDCUT$= 2^oC$. However, if there is evidence for a prevalence of temperature inversions during precipitation, this reference value should be lowered.

To partially account for the reduced variance of hourly temperature during cloudy days, a Gaussian distribution with a reduced effective standard deviation ($\sigma_{PDf}$, as compared to that of Wake and Marshall, 2015, used for the PDD determination) is used for the positive degree fraction. We use the observational fitted value of Seguinot and Rogozhina (2014):

$$\sigma_{PDf} = -0.15 \cdot T_{2m} + 1.66 . \tag{22}$$

Though precipitation, surface melt, PDDs, positive temperature insolation, and NDYs are computed monthly, surface melt-water refreezing is computed yearly. Furthermore, all net snow accumulation (*i.e.* after melt loss) in one year transitions to ice the next year. This invokes the assumption that once surface mass balance is positive in the yearly cycle (on a monthly mean basis), refreezing won't be significant until the start of the next melt season. This avoids issues around tracking snow age and snow amounts between consecutive years at the cost of errors that are overwhelmed by input and parametric uncertainties. For

those doing detailed firn modelling, a more refined (likely sub-diurnal) approach would be required.

### 2.7.2  Submarine melt and refreezing

Though there has been significant progress in submarine melt and refreezing parametrizations (as compared in Asay-Davis et al., 2017; Favier et al., 2019), a confident and computationally tractable representation for submarine melt remains an ongoing challenge. This is especially so for glacial cycle contexts for which the required ocean temperature fields are unlikely

to be available to the requisite accuracy in the foreseeable future.

    The recommended sub ice shelf melt (SSM) representation for the GSM is the (-DSSMslope -DSSMslopeLJGW19) buoyant plume model from Lazeroms et al. (2019). It give the SSM ($ssm$ in m/yr)) as a function of the basal ice angle ($\theta$) , ambient ocean temperature near the grounding line ($T_a$), local ice depth ($z_b$) and a non-dimensional horizontal coordinate $x$:

$$
\begin{aligned}
ssm &= C_{\mathrm{SSM}} \cdot FS(\gamma(\theta)) \cdot (T_a - T_{fz})^2 \cdot M_0(x(\gamma(\theta))) \\
FS(\gamma) &= D_s^{1/2} \cdot \left[ \frac{C_d^{1/2}\Gamma_{TS} \cdot \gamma}{(C_d^{1/2}\Gamma_{TS} + C_T + \gamma))} \right]^{3/2} \left[ \frac{1 - C_{\rho 1}C_d^{1/2}\Gamma_{TS}}{(C_d + \gamma)} \right]^{1/2} \\
M_0(x) &= \frac{1}{2\sqrt{2}}[3(1-x)^{\frac{4}{3}} - 1][1 - (1-x)^{\frac{4}{3}}]^{\frac{1}{2}} \\
x(\gamma) &= \min\left( \lambda_3 \frac{z_b - z_{gl}}{T_a - T_{fz}} \left[ 1 + C_\epsilon \left( \frac{\gamma}{C_d^{1/2}\Gamma_{TS} + C_T + \gamma} \right)^{\frac{3}{4}} \right]^{-1}, 1.0 \right) \\
\gamma(\theta) &= E_0 \cdot \sin(\theta); \; E_0 = 0.036 : \text{Entrainment coefficient} \\
C_d^{1/2}\Gamma_{TS} &= 5.9 \times 10^{-4} : \text{Effective thermal Stanton number} \\
C_d &= 2.5 \times 10^{-3} : \text{Drag coefficient} \\
C_T &= 1.4 \times 10^{-5}; C_\epsilon = 0.6; C_{\rho 1} = 2.0 \times 10^2 \\
D_s &= \frac{\beta_s S_a g}{\lambda_3 (L_w/c_p)^3}; \; \beta_s = 7.86 \times 10^{-4}\mathrm{psu}^{-1}; \; S_a = 34.65\,\mathrm{psu}; \; \lambda_3 = 7.61 \times 10^{-4}\mathrm{K/m}
\end{aligned}
\tag{23}
$$

This plume model also accounts for refreezing via negative SSM values. The depth of the plume source grounding line ($z_{gl}$) and associated location and depth for extraction of $T_a$ is determined via a downslope search. The reference freezing temperature

($T_{fz}$) at the grounding line is depth corrected. There is the option of subjecting $T_{fz}$ to regional or whole grid ice shelf ensemble parameter dependence ($CT_{\mathrm{ssmCut}}$) to partly compensate for limitations in the ocean temperature forcing:

$$
T_{fz} = CT_{\mathrm{ssmCut}} - \lambda_3 \cdot z_{gl} - 2.0^oC
\tag{24}
$$

A related limitation is the plume model is purely buoyancy driven and therefore ignores horizontal advection due to sub ice shelf ocean circulation. The overall SSM ensemble parameter $C_{\text{SSM}}$ adds further parametric degrees of freedom to partly compensate for these error sources.

There are two options for subshelf melt at grounding line grid cells. The default (no special compile flag) option is that the GSM only applies the above subshelf melt parameterization to fully floating grid cells. This is in accordance with resolution convergence tests comparing application of submarine melt to just floating grid cells as well as to fractional inclusion of grid cells crossing the grounding line grid cells in proportion to the subgrid grounded ice fraction (Seroussi and Morlighem, 2018). However Seroussi and Morlighem (2018) only tested subshelf melt parametrizations that do not decrease in magnitude near the grounding line (before application of subgrid relative floating area scaling), unlike that of the recommended Lazeroms et al. (2019) plume parameterization. Furthermore, the experiments only evaluated a 100 year retreat scenario and it remains unclear whether their conclusions hold in the case of a glacial advance and subsequent retreat scenario of more relevance for glacial cycle modelling. As such, the GSM has an option for scaling of subshelf melt by the relative subgrid area that has floating ice (-DGLssm, similar to configuration SEM1 in Seroussi and Morlighem, 2018). This has an added scaling parameter (RfactGLssm) to further reduce grounding line grid cell subshelf melt. Sensitivity tests have found grounding retreat and advance to be more stable for RfactGLssm$= 0.5$ compared to the simulations with subshelf melt only for fully floating grid cells.

Calving face submarine melt is taken from the results of high resolution Massachusetts Institute of Technology general circulation ocean modelling (Rignot et al., 2016). We use their extracted analytical fit for submarine melt of west Greenland outlet glaciers. This has a dependence on the approximated meltwater velocity (q, m/day) and the interpolated (or extrapolated) ocean temperature ($T(x,y,z)$). In detail, the submarine melt ($q_m$ in m/day) of a calving face that extends to depth $d$ is given by:

$$q_m(d) = (A \cdot C_{\text{face}_A} \cdot d \cdot q^a + B \cdot C_{\text{face}_B}) \cdot \max(T_F(x,y) - CT_{\text{ssmCut}} , 0)^{\beta} \tag{25}$$

with $a = 0.39$, $A = 3 \times 10^{-4} \ m^{-a} \ \text{day}^{a-1} \ ^oC^{-\beta}$ , and $\beta = 1.18$ as per Rignot et al. (2016). The freezing point ($CT_{\text{ssmCut}}$) is treated as a ensemble parameter to impose bias corrections for the ocean temperature forcing. $B = 0.15 \ ^oC^{-\beta} \ m \ \text{day}^{-1}$ as per Rignot et al. (2016) assuming an average of 180 melt days. The above equation is rescaled for m/yr quantities and q is approximated by scaling the sum of twice the subglacial melt rate of the grid cell (to allow for some upstream contribution) and surface runoff by the grid cell area to marine face area ratio. To allow for uncertainties in the application of the above formula to the coarser grid scale resolution typical of paleo ice sheet modelling, an ensemble scaling parameter ($C_{\text{face}}$) is added. Until recently this has only been applied to the $B$ coefficient (as $C_{\text{face}_B}$ in eq. 25). However, given the potentially large uncertainties due to submarine circulation (driven in large part by buoyancy forcing from meltwater), the GSM has the option of switching the ensemble parameter to the $A$ coefficient (and thus $C_{\text{face}_A}$ in eq. 25, using the -DCfaceMltFW compile flag).

### 2.7.3 Marine ice shelf calving

For marine floating ice calving, two dynamical controls are assumed. First, a stress-balance crevasse propagation parameterization following Pollard et al. (2015) is used. This is expressed as a horizontal wastage rate ($W_c$) (though numerically applied as an appropriately scaled contribution to the surface mass-balance forcing) subject to ensemble parameter $C_{\text{calv}}$:

$$
\begin{aligned}
W_c &= C_{\text{calv}} \cdot 10\,\text{km/yr} \cdot \max[0, \min[1, (r - r_c)/(1 - r_c)]] \,, && (26) \\
r &= (d_s + d_b + d_a + d_t + d_w)/H_t \,; \; r_c = 0.75 && (27)
\end{aligned}
$$

where each $d_?$ term represents a contribution to crack depth propagation as detailed below. As indicated in eq. 27, calving is activated when the relative total crevasse depth ($r$) reaches the critical relative depth $r_c$, with latter set to the value in Pollard et al. (2015). As calving in the GSM is only allowed for ice marginal grid cells, the sum of the contributions from strain rate divergence to dry-surface ($d_s$) and basal ($d_b$) crevasses is given by that for a free floating unconfined ice face (*e.g.,* Schoof, 2007):

$$ d_s + d_b = H_t/2 \tag{28} $$

Following Pollard et al. (2015), an accumulated strain contribution is included to crudely account for upstream accumulation of fine-scale fracturing from strain divergence:

$$ d_a = H_t \, \max\left[0, \ln(u/800)\right]/ln(1.2) \tag{29} $$

As detailed in Pollard et al. (2015), this derives from a steady flow solution to the time integral of the ice divergence along
the flowlines. To improve GSM fits to present day (PD) observed Antarctic ice shelf extents, the 1600 m/yr value for the denominator (representing the flowline velocity at the beginning of the ice shelf trajectory) in Pollard et al. (2015) was reduced by a factor of 2.

The $d_t$ term is added to prevent floating ice thinner than $\sim 150$ m in accord with present-day Antarctic ice shelves. Our implementation has slight alterations to that imposed in Pollard et al. (2015) to better facilitate ice margin expansion. These are
the use of the maximum of adjacent grid cell ice thickness ($H_{\text{adjmx}}$) instead of the marginal ice thickness ($H_t$) and an increase of the cessation threshold to 200 m from 150 m:

$$
d_t = \begin{cases} H_t \, \max(\, 0., \, \min(\, 1, \, (200 - H_{\text{adjmx}})/50)) & \text{Depth}_{\text{ocean}} > 300\,m, \\ 0 & \text{else} \end{cases} \tag{30}
$$

Recovery of present-day AIS ice shelf extent is also further improved with imposition of a minimum marine depth of 300 m for activation of this component.

The remaining $d_w$ term in eq. 27 is the additional surface crevasse depth due to hydro-fracturing from water infill:

$$ d_w = C_{\text{hydCrk}} \cdot 100 \cdot (\text{GSM surface runoff flux (m/yr)})^2 \tag{31} $$

This matches the corresponding term in Pollard et al. (2015) for $C_{\mathrm{hydCrk}} = 1$ as motivated in that paper.

The default terminal ice thickness ($H_t$) estimate in the GSM is simply $\max(0.95\,H, 200\,m)$ with the floor value set in line with what is mostly observed for the margins of large Antarctic ice shelves. The code includes (as a compile flag - DhedgeActive) the downstream thinning option of Pollard et al. (2015). The activation of this option tends to increase numerical instability with otherwise limited impact on results after accounting for compensation from ensemble parameter variations.

Unlike many other ice sheet models, a second control is the assumption that if summer sea surface temperature forcing (approximated by 2 m air temperature) is too cold to permit sea-ice free conditions (summer $T_{2m} <$ TcalvCut$=-2^oC$), then iceberg production will cease due to back-stress and potentially reduced adjacent marine convection (driving undercutting). This is motivated by both the tendency for seasonal calving in the high Canadian Arctic to initially occur after the loss of land-fast sea ice and the bracketing of the Antarctic ice shelf margin with the $-2^oC$ and $-6^oC$ mean summer sea surface isotherms. The one exception is that complete calving is assumed for ice shelves beyond the continental shelf break. This shelf break location is set by present-day depth equal to GSM parameter rDepthDeepCalv(Ice sheet Index)$= 860$ m, except for Antarctica where a 1700 m depth was found necessary to permit present-day fringing ice shelf margins around the Antarctic Peninsula as observed. This deep sea calving reduces computational cost where an ice shelf would definitely have no confinement and therefore impose no back-stress upstream.

Ice calving is computed for each marine ice margin grid cell interface, even if this entails more than one interface of a grid cell. Given the limited grid resolution, an ice covered grid cell is taken as marine margin if it has an adjacent grid cell with ocean depth greater than 40 m and ice cover less than 5 m thick. The GSM tracks open marine basin connectivity to the ocean and shuts down calving when the connectivity is lost on the assumption that iceberg congestion will adequately increase back-stress on the calving front to terminate calving.

### 2.7.4 Tidewater calving

We use the ice cliff failure enabled tidewater calving scheme of Pollard et al. (2015), with the horizontal calving rate ($W_{ct}$ in m/yr) computed as:

$$W_{ct} = C_{\mathrm{calv}} \cdot 10\,\mathrm{km/yr} \cdot \max[0., \min[1, (h_{\mathrm{sw}}\, F - H_c)/10]] \tag{32}$$

where

$$F = \frac{\theta}{\max[10^{-6}, 2(1 - \theta/2 - d_w/H_{\mathrm{GL}})]} \tag{33}$$

The critical surface height above the water line for ice cliff failure is set to $H_c = 100\,m$. $H_{GL}$ is ice thickness at the grounding line (computed from applying the flotation condition to the horizontally interpolated grounding line depth). $h_{\mathrm{sw}} =$ is height above water at the grounding line, and the contribution from crevasse water depth ($d_w$) is computed as above for ice shelf calving. $\theta$ is related to the back stress on an ice shelf with value 1 for an unbuttressed ice shelf or no floating ice at all. As we restrict calving to the ice marginal grid-cell, the model has a default option of permanently setting $\theta$ to this value. This restriction is distinct from that of Pollard et al. (2015) which allows marine ice cliff failure for interior (non-marginal) ice at the grounding line if there is no significant buttressing by the attached ice shelf.

 **2.7.5   Lake calving**

As lake margins of ice sheets are indicative of relatively warm conditions, the GSM lacustrine calving model assumes that surface melt filling of cracks and associated crack propagation is not a control on lake calving. Instead, it is assumed that the main control is the available heat to melt icebergs. Once all excess heat is used up, the lake is assumed to quickly choke up with icebergs, and thereby block further calving. As such, lacustrine calving is simply implemented as extra melt applied to the grid cell covering the calving margin.

Given the large process uncertainties, the potential iceberg melt is just set to the total computed net potential surface melt of adjacent lake filled grid cells times a GSM parameter (flac). This thereby lacks accounting for extra lake grid cells that are not in contact with the ice-sheet. It also assumes that the effective surface albedo of the lake will be dominated by that of icebergs and ice melange, and thus the melt potential for a unit area is close to that computed (the surface melt calculation doesn't have a separate albedo for lake cover).

Two further somewhat adhoc conditions are imposed to ensure there is sufficient exposure of the grid cell ice to adjacent cell water and sufficient lake depth to enable heat circulation. These requirements are : 1) local water depth of calving grid cell is more than the lesser of SLACMX (set to 50 m) and the local ice thickness, and 2) ice-free adjacent cell water depth is greater than SLACMIN (set to 20 m except for EA = 33 m to enable adequate EA ice sheet expansion in certain regions).

For an example North American ice sheet (NAIS) simulation, inhibition of lacustrine calving (cf Fig. 5) has a significant impact on ice sheet volume during the 80 ka interstadial and 60 ka to 40 ka glacial interval.

## 2.8   Surface drainage solver

The mass-conserving solver simply diagnostically routes water downslope, filling depressions (lakes), until an ocean depth of 200 m or until no water is left. It computes marine drainage summaries for defined drainage basins, for total (including precipitation over ice-free land), ice sourced, and solid-fraction only drainage. For present-day ice free surface topography, the solver uses a modified version of the USGS EROS HYDRO1k hydrologically self-consistent DEM (USGS, 2004). The drainage preserving upscaling of the DEM includes some by hand corrections to capture the controlling sill elevation for the southern drainage of the central LIS (*e.g.,* pro-glacial lake Agassiz). This topography is then dynamically evolved for ice cover and GIA. Details on the solver, drainage topography creation, and validation are in Tarasov and Peltier (2006).

The algorithm is run every dlong years (default is 100 year) and accumulates mean surface runoff and marine ice discharge over the dlong interval. A discharge map is also created for coupling with climate models or other such contexts.

Given the limited subaerial Greenland and Antarctic terrestrial surfaces over which grid-cell scale pro-glacial lakes could form, the surface drainage solver is generally not activated for these ice sheets.

## 2.9   Sea and lake ice formation

The GSM includes a simple thermodynamic sea and lake ice formation module. The inclusion of the former is motivated by evidence for paleocrystic ice (floating ice grown directly from local precipitation and not terrestrially sourced, Bradley and

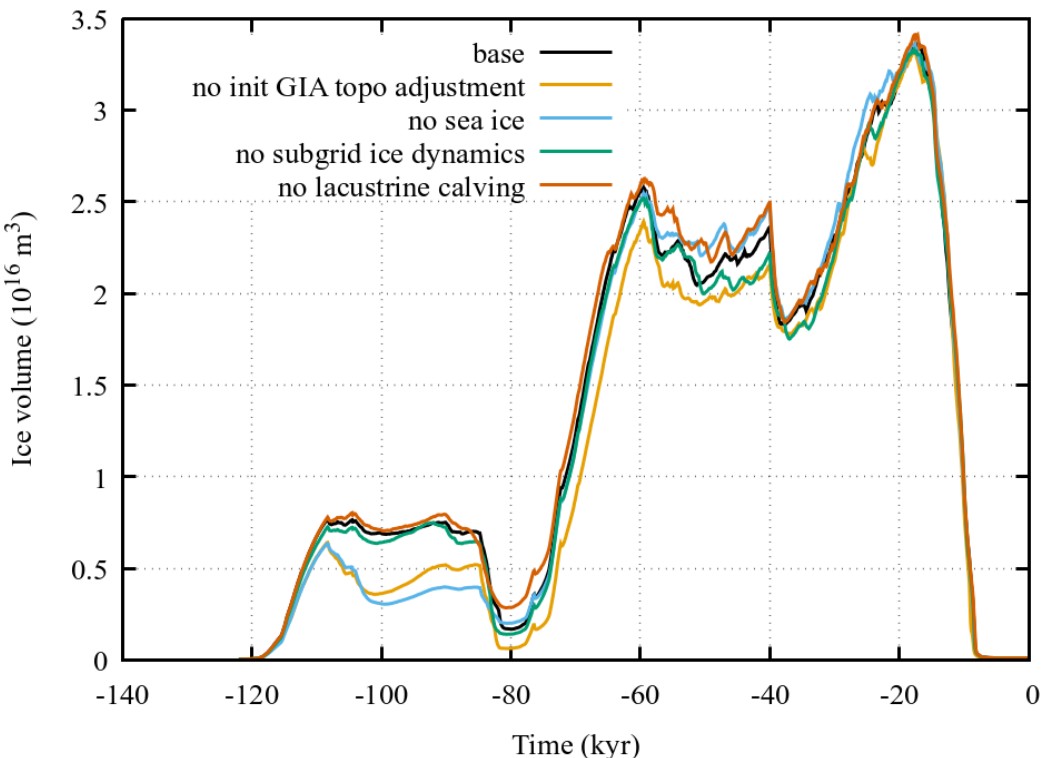

**Figure 5.** Some example GSM sensitivities to one at a time process removal for a NAIS simulation.

England, 2008). It was also found necessary to remove spurious ice holes in the Barents and Kara Seas that occasionally developed under glacial moisture starved conditions. As shown in Fig. 5, sea ice inclusion can play a significant role in increasing NAIS glacial inception ice volume, a long standing challenge for paleo ice sheet models when coupled to full climate models.

The lake and sea ice basal accumulation model assumes a monthly approximately thermodynamic steady state for the floating ice, and thereby a linear temperature profile. After a trivial integration this gives growth in effective sea or lake ice thickness ($H_f$) over time $\Delta t$ as:

$$H_f(t + \Delta t) = \sqrt{H_f(t)^2 + \Delta T(t) \cdot \Delta t \cdot k/(\rho_i \cdot L_w)} \tag{34}$$

where $L_w$ is the latent heat of fusion for water, $k$ is the thermal conductivity of ice, and

$$\Delta T(t) = \max(0, -3^oC - T_{2m}(t)) \tag{35}$$

The change in ice thickness is not directly imposed in the GSM but instead converted to an effective contribution to the surface mass-balance term (otherwise this ice accumulation would break mass conservation in the surface runoff discharge calculation). This lake and sea ice remains subject to all the other mass-balance processes in the GSM.

## 2.10 Climate forcing

The GSM climate forcing generates evolving monthly precipitation, near surface air temperature, and ocean temperature fields. It includes dependence on various glacial indices as detailed below.

### 2.10.1 Glacial indices for climate forcing

The GSM has various time and/or state evolving indices for driving components of the climate forcing (Fig. 6). The indices cover a range from below 0 to beyond 1 with 0 representing the 0 ka (nominally 2000 CE) state and 1 the LGM state. A to date unique feature is the addition of monthly dependence for the glacial indices (Fig. 6). The traditional reliance on mean annual glacial indices (*e.g.,* Marshall et al., 2000; Scherrenberg et al., 2023) hides the significant impact of changes in seasonality over the glacial cycle. For example August $-$ February differences range up to $0.30$ over the last two glacial cycles for the EBM derived monthly glacial index. For a more advanced coupled ice-climate model over the last two glacial cycles, the glacial index differences can exceed $1.0$ (Geng et al., 2025). Furthermore, mean annual glacial indices result in excessive summertime variability as these records average the relatively low climatological summer time variance of air temperature with the much high variance of winter temperatures (Fig. 6 and Geng et al., 2025).

The $I_e$ index is the mean monthly EBM temperature anomaly relative to present-day over the 40N:80N latitudinal band divided by corresponding LGM anomaly. The $I_g$ glacial index uses ensemble parameter rWtEBMindx to weigh $I_e$ with an input glacial index chronology specified in the runscript. The latter glacial index can be in mean annual format from a deep ice core isotopic record. There is also a compile flag option to include a monthly glacial index input from a more advanced coupled ice and climate model.

For computing 2 meter air temperature, the glacial $I_g$ is subject to two ensemble parameters, $C_{\mathrm{IT}}$ and $\Theta_T$, respectively adjusting amplitude and phase:

$$I_c = \mathrm{SIGN}(C_{\mathrm{IT}}, I_g) \cdot |I_g|^{\Theta_T} \tag{36}$$

For controlling the Atlantic meridional overturning circulation impact parameterization (cf next subsection), the annual $I_N$ index uses the average of the scaled pCO2 forcing $(\min(280, pCO2(t))/90)$ and a North Atlantic index for the scaled mean annual EBM temperature anomaly over $40^o$ to $20^o$ W and $40^o$ to $45^o$ N region. $I_N$ ranges from about $-5$ to $0$.

The $I_d$ ice dome index is a function of the maximum elevation of the main ice dome ($h_{\mathrm{Idmx}}$). It is subject to ensemble parameter $h_{\mathrm{Ides}}$ as following

$$I_d = \min((\max(h_{\mathrm{Idmx}} - 1\,\mathrm{km}), 0)/(2\,\mathrm{km}), 1)^{(2 - 1.5\,h_{\mathrm{Ides}})} \tag{37}$$

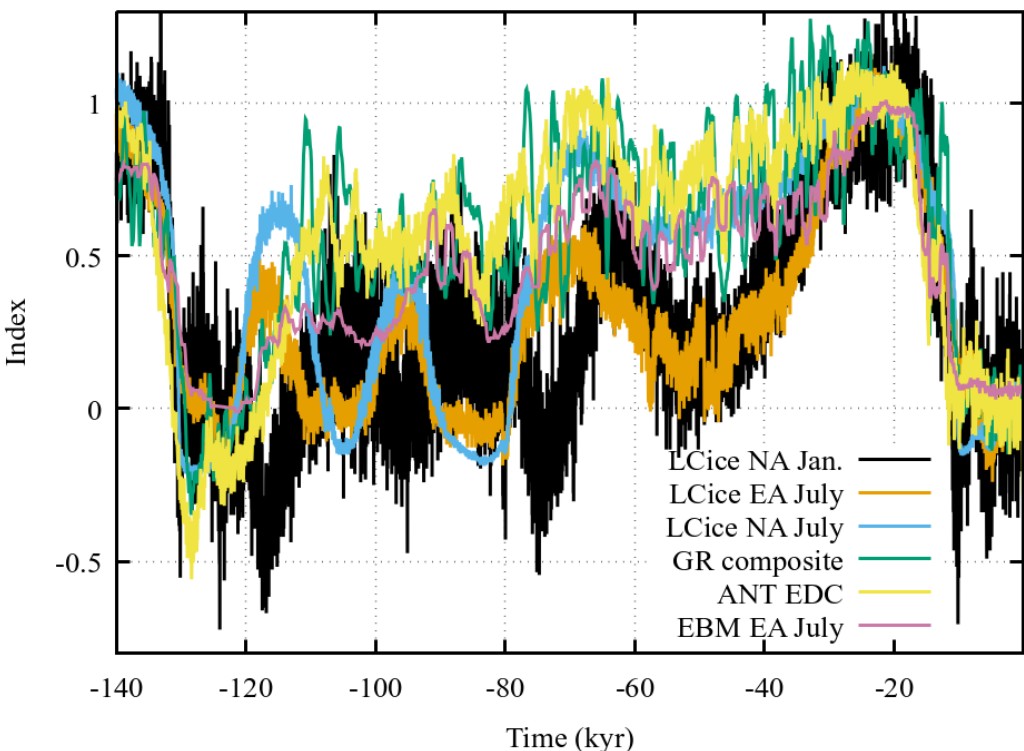

**Figure 6.** Comparison of some GSM glacial index input options. The 'EBM EA July' entry is the computed EBM glacial ($I_e$) index for a sample Eurasian GSM simulation. The LCice indices are derived from the same coupled GSM-LOVECLIM simulation (Geng et al., 2025). The GR composite and ANT EDC (Epica Dome C) indices are derived from ice core records (cf Table 6).

The index is used to partly account for possible large scale circulation response to the changing elevation of the main ice dome as discussed below.

### 2.10.2 Surface climate forcing

The biggest source of uncertainty for glacial cycle ice sheet modelling is the climate forcing. To partly address this, the GSM features different climate forcing options that can be combined under ensemble parameter specified weighting.

**Air temperature:** the first temperature forcing option is an asynchronously coupled 2D energy balance climate model (EBM), running at spherical harmonic truncation T11 with non-linear sea ice and snow albedo feedbacks (Deblonde et al., 1992).

Sea level temperature ($T$) is computed by an approximation for the energy balance of the tropospheric and mixed-layer ocean column that only accounts for vertical radiative fluxes, horizontal diffusive heat transport, and a parameterized North

Atlantic oceanic heat flux contribution (NAHF):

$$C(\mathbf{r})\frac{\partial T(\mathbf{r},t_y,t_m)}{\partial t} = (S_o a(\mathbf{r},t_y,t_m)S(\theta,t_y,t_m)/4 + \text{NAHF}(\mathbf{r},t_y)) + \Delta Rad_{CO2}(t_y) + \Delta Rad_{CH4}(t_y)$$
$$+ \Delta Rad_{\text{otherIce}}(t_y) - (A + B(T(\mathbf{r},t_y,t_m) - \lambda_{\text{ebm}}h(\mathbf{r},t_y)) - \nabla_h[D(\theta)\nabla_h T(\mathbf{r},t_y,t_m)])) \qquad (38)$$

The equation is solved for monthly mean equilibrium solutions on default 100 year increments. Year ($t_y$) and monthly ($t_m$) are explicitly indicated in this section (though at times subsumed into $t$). To avoid clutter, spatial ($x, y$ or $\mathbf{r}$) dependence is usually not shown. The linearized long-wave emission ($A + B(T(\mathbf{r},t_y,t_m) - \lambda_{\text{ebm}}h(\mathbf{r},t_y))$) accounts for reduced emission at higher elevations due to cooler temperatures and is implemented via a constant lapse rate ($\lambda_{ebm}$). The absorbed short-wave radiation is set to the product of the solar constant $S_o$ (1360 $\text{W m}^{-2}$), an effective coalbedo $a(\mathbf{r},t)$, and the solar distribution function

$S(\theta,t_y,t_m)/4$ with dependence on latitude $\theta$. The coalbedo has latitudinal dependence derived from satellite observations (Stephens et al., 1981) and seasonal dependence on snow and sea ice cover. The time-dependent orbital parameters for $S$ are computed as per Berger and Loutre (1991). The heat capacity $C$ has four possible values according to surface type (land, land ice, sea ice, and water). The diffusion coefficient $D$ is tuned to preserve the present-day observed mean latitudinal temperature gradient. The radiative forcing due to changing atmospheric greenhouse gas (GHG) concentrations (currently restricted to CO2

and CH4) is accounted as per Myhre et al. (1998) (though with rounding up of the numerical coefficients to partly compensate for missing feedbacks in the EBM):

$$\Delta Rad_{CO2}(t_y) = 6\ln\left(\frac{pCO2(t_y)}{300\ ppmv}\right) \qquad (39)$$

for CO2 and

$$\Delta Rad_{CH4}(t_y) = 0.04\left(\sqrt{pCH4(t_y)} - \sqrt{1100\ ppbv}\right) \qquad (40)$$

for methane. The chosen reference concentration values are between those corresponding to pre-industrial and a 1980:2000 CE reference climate interval. This is to account for far from complete transient response to present-day GHG changes.

Given the impact of heat transport by the Atlantic meridional overturning circulation and its changes over the past, as well as to improve fits to present-day observed climate, the EBM has an added horizontal surface ocean heat flux ($\text{NAHF}(\mathbf{r},t_y)$). This flux has geographically dispersed weak sinks, and a source concentrated around ($17.5^o$ E, $67.5^o$ N), with the central

position determined by root mean squared error minimization against present-day reanalysis climatology. This represents a displacement by about 15 degrees east from the climatologically observed net ocean surface to air heat fluxes, presumably accounting for eastward advection by mid-latitude westerlies. The heat flux has various choices of dependencies on the current $I_N$ index value and the state history set by a compile flag. The current recommended choice is -DNAHFv3. It is also a function of pCO2 forcing and sea level, configured so as to induce Dansgaard-Oeschger-like oscillations in air temperature. Given the

adhoc nature of the implementation, we leave the documentation to the source code (pGSM.F90) for those interested.

A linear version of the EBM (without seasonal snow/ice albedo feedback) has previously been evaluated against observations and output from an early version of the NCAR CCM general circulation climate model (run at wave number 15 rhomboidal truncation). It was found to capture much of the millennial scale response on this spatial scale especially for the Northern

Hemisphere (Hyde et al., 1989). Given that the EBM lacks atmospheric dynamics and as such won't be able to capture the effects thereof, the model is generally run in anomaly mode, with the EBM providing the climate forcing anomaly relative to a present-day monthly climatology ($T_{\mathrm{rean}}$) and subject to an ensemble scaling parameter $C_{\mathrm{EBM}}$:

$$T_{\mathrm{Eb}}(t_y, t_m) = C_{\mathrm{EBM}} \cdot (T_{\mathrm{EBM}}(t_y, t_m) - T_{\mathrm{EBM}}(0, t_m)) + T_{\mathrm{rean}}(t_m) \qquad (41)$$

This presumes radiative perturbations dominate the climate system response to orbital forcing changes.

Comparison of PMIP II and III simulations along with a dedicated set of CESM 1.2 experiments (Bakker et al., 2020) has identified Siberia as the region having the highest LGM summer temperature sensitivity to climate model choice and configuration. Lofverstrom and Liakka (2018) have also shown that strong grid resolution dependence for Northern Eurasian June/July/August (JJA) surface temperatures for the NCAR CAM3 atmospheric GCM when run below T85. Given the low T11 resolution of the EBM and its lack of atmospheric dynamics, an added parameterization is used to correct excessive glacial summertime cooling as evidenced by Siberian ice growth in simulations contrary to the geological record. The additive correction field is approximately derived from differences between EBM and mean PMIP LGM JJA sea-level temperatures. On the assumption that relevant circulation changes are driven by topographic changes, this warming is scaled by product of the Fennoscandian ice dome elevation index $I_d$ and the GSM ensemble parameter rSumPlusEBM.

When run in single ice sheet mode, the EBM will under predict glacial cooling as a result of the missing radiative impact of ice sheets that are not modelled. As such, the GSM has an option (-DdRadIndx) to implement a scalar decrease in shortwave input to compensate for missing ice sheets. Concretely, this is implemented as

$$\Delta Rad_{\mathrm{otherIce}}(t_y) = -\mathrm{dradSea} \cdot \max(-\mathrm{globalMeanSealevel}(t_y)/125\, m, 0)^{1.5} \qquad (42)$$

with parameter dradSea set in the run script (generally ranging from 1 to 7 $W/m^2$, determined by comparison of EBM results with single and global ice sheet configurations). This implementation assumes a $-125\, m$ mean glacial maximum sea level so that dradSea is the corresponding radiative forcing. The exponent value was chosen to approximately account for the limited radiative impact of the major Northern hemispheric ice sheets until their southern extent reaches regions not typically covered by snow or sea ice for the majority of the year. The exponent value will at some point be refined by modelling experiments with the EBM.

The second temperature forcing option is a glacial climate indexed ($I_c$) interpolation between a present-day reanalysis climatology and a full-glacial (LGM) climatology for mean monthly sea level temperature:

$$\begin{aligned} T_{\mathrm{PMIP}}(t_y, t_m) \;=\; & I_c(t_y, t_m) \cdot \left( T_{\mathrm{PMIP}}(LGM, t_m) + \sum_i C_{\mathrm{TEOF}}(i) \cdot \mathrm{TEOF}_i(LGM, t_m) \right) \\ & + (1. - I_c(t_y, t_m)) \cdot T_{\mathrm{rean}}(t_m) \end{aligned} \qquad (43)$$

The glacial climatology ($T_{\mathrm{PMIP}}(LGM, t_m, x, y)$ and $P_{\mathrm{PMIP}}(LGM, t_m, x, y)$) is derived from the highest resolution three to four climate model simulations in past PMIP experiments. It is the sum of the mean of the simulations and the top one to three inter-model Empirical Orthogonal Functions (EOFs, a mathematical tool to capture orthogonal modes of maximum variance). The addition of each EOF component for precipitation and two meter air temperature is subject to individual ensemble parameter weighting $C_{\mathrm{PEOF}}(i)$ and $C_{\mathrm{TEOF}}(i)$) to account for the significant inter-model differences in the PMIP simulations.

For North America, the orographic forcing from a large Keewatin dome has been shown to significantly perturb atmospheric circulation and therefore North American climate (Kutzbach and Wright Jr, 1985; Andres and Tarasov, 2019). To partly address this dependency on changing dome size over a glacial cycle, the GSM takes advantage of the difference in LGM boundary conditions between PMIP I and PMIP II with the former have no Keewatin ice dome (ICE4G Peltier, 1994) and the latter having an excessively high Keewatin ice dome (ICE5G Peltier, 2004). The Keewatin dome elevation index $I_d$ is used to weight mean LGM PMIP I and PMIP II temperature and precipitation fields.

Greenland has a further temperature component parameterized as functions of latitude, longitude, surface elevation, month, and glacial index, based on those derived from linear regression of present-day climatologies (Fausto et al., 2009). Greenland also includes an added Holocene latitude-dependent warming ($T_{\text{agy}}$) to capture one of the main regional forcings that had been previously found to help address misfits in ensemble fitting of deglacial Greenland ice sheet simulations to paleo and geophysical constraints (Lecavalier et al., 2014). This strong high latitude warming also has support from analysis of the isotopic record of the Agassiz ice cap (Ellesmere Island, Canada, Lecavalier et al., 2017). The added warming component takes the form:

$$T_{\text{agy}}(t_y) = C_{\text{HTM}} \cdot T_{\text{ag}}(t_y) \cdot (\max(0.0, \min(1.0, (\theta - 60.0)/\Theta_{\text{wrm}})))^2 \tag{44}$$

with explicit dependence on latitude ($\theta$) and ensemble parameters $C_{\text{HTM}}$ and $\Theta_{\text{wrm}}$ respectively controlling amplitude and latitudinal range of the Northward linear ramp-up. $T_{\text{ag}} = 9^{\circ}\text{C}$ for the early Holocene then linearly ramped down to 0 over the 10.15 ka to 4.7 ka interval. As this extra warming is beyond that inferred from GRIP and NGRIP, the forcing is linearly ramped down to 0 over the 0 to 2000 masl surface elevation interval.

A third temperature forcing component option for Antarctica is simply a scalar glacial index forcing plus $10^{\circ}\text{C}$ forcing per pCO2 doubling and lapse rate vertical temperature adjustment (as in Pollard and DeConto, 2012) applied to the present-day reanalysis climatology.

The above sea level temperature forcings are then subject to a weighted sum (controlled by ensemble parameters $wT_{\text{EBM}}$ and $wT_{\text{PMIP}}$) :

$$\begin{aligned} T_{2m} = \ & wT_{\text{EBM}} \cdot T_{\text{Eb}} + (1. - wT_{\text{EBM}}) \cdot (wT_{\text{PMIP}} \cdot T_{\text{PMIP}} \\ & + (1. - wT_{\text{PMIP}}) \cdot T_{\text{Thirdoption}}) + (I_c \cdot L_{\text{LGM}} + (1. - I_c) \cdot L_0) \cdot h \end{aligned} \tag{45}$$

The final effective near surface air temperature ($T_{2\text{m}}$) is obtain with the addition of a glacial index interpolated lapse rate (between 0 ka value $L_0$ and LGM value $L_{\text{LGM}}$) applied to the contemporaneous surface elevation ($h$).

A major problem with glacial indexed interpolation of input GCM fields is that these fields have a very strong imprint of their ice sheet boundary condition. The implicit migration of the ice sheet margin between GCM time-slices and the impact thereof is unlikely to be captured by the imposed linear interpolation for sea level temperature. As such, an optional parameterization (-DHboostTindx), imposes an increase of the glacial index (limited by the LGM value of 1) whenever the GSM grid cell is ice covered. This is linearly imposed for thin ice as follows:

$$I_c \leftarrow \min(1 , I_c + \min(1 , H(x,y)/500) \cdot I_{\text{H}+}) \tag{46}$$

with $I_{H+}$ being an ensemble parameter (nominal range 0:0.2). In the future, this may be made more accurate by adding a blurred version of the GCM glacial state ice mask, so that changes are only imposed where there is a discrepancy between the GCM ice mask and the GSM grid cell ice cover. This would also enable the clean addition of local glacial index reduction if the GSM has no ice cover in a grid cell for which the GCM LGM ice mask has ice.

**Precipitation and evaporation:** the first precipitation forcing option is relative interpolation between the present-day observed climatology $P(0, x, y)$ (Hersbach et al., 2023) and the LGM field from the PMIP ensemble $P(LGM, x, y)$ using the following function of the glacial index $I_g(t)$:

$$P(t, x, y) = P(0, x, y) \cdot \left( \frac{C_{\text{pre}} \cdot P_{\text{PMIP}}(LGM, x, y)}{P_{\text{PMIP}}(0, x, y)} \right)^{I_g(t)^{\Theta_P}} . \tag{47}$$

The "ensemble phase factor" ($\Theta_P$) parameterize some of the uncertainty associated with the transition from interglacial to glacial atmospheric states. $C_{\text{pre}}$ is a global ensemble scale parameter.

The second precipitation forcing option is a generalization of the Clausius-Clapeyron relation for the saturation vapour pressure dependency on temperature. This precipitation component ($P_T$) is also subject to ensemble parameter $C_{\text{Tp}}$ as follows:

$$P_T = \exp(C_{\text{Tp}} \cdot (T_{2m} - T_{2m_0})) P_0 \tag{48}$$

where the 0 subscripted components are from present-day reanalysis.

Precipitation is further controlled by a range of regional parameters. Most take the form of a desert-elevation (Budd and Smith, 1981) threshold control over a specified geographic region. Ensemble parameters set the regional threshold in an array (rdes, listed as "desert-elevation cutoffs" in Table 3). Computed precipitation is then subject to the factor:

$$\exp(C_{\text{des}} \cdot \max(h_{\text{dk}} - (I_{\text{dk}} \cdot \text{rdes}(x, y) + (1. - I_{\text{dk}}) \cdot h_{\text{des0}} + \text{fmindeselcut}), 0)) \tag{49}$$

where $h_{\text{dk}}$ is 75% of the difference in elevation in kms between the GSM and the $I_{\text{dk}}$ index interpolated orography for the climate model fields. The $I_{\text{dk}}$ index is given by the average of the climate index ($I_c$) and dome elevation index $I_d$ (cf section 2.10.1) to crudely insert global and regional scale dependencies of atmospheric circulation. fMINdeselcut has a default value of $-0.5$ km to allow for potentially excessively high input orographies. These controls are perhaps best interpreted as regional smooth limits on maximum ice elevation facilitating fit to deglacial constraints operating within the large uncertainties for paleo precipitation.

Evaporation, sublimation, and deposition are three further climate forcing components that affect surface mass-balance. As these processes are highly dependent on near surface vapour pressure gradients which in turn depend on boundary layer turbulent mixing, they are unlikely to have any simple large scale orographic dependencies as imposed on precipitation in the GSM. Lacking a better alternative, the GSM simply applies the same glacial index geometric interpolation between present-day and LGM fields as used for precipitation in eq. 47 to the net of deposition less evaporation and sublimation. The resultant field is then added to precipitation as the final step in determining monthly mean precipitation within the GSM (*i.e.* after all other adjustments described in this and the next section).

### 2.10.3 Orographic precipitation down-scaling

Paleo ice sheet modellers have traditionally relied on a simple exponential function of surface elevation or temperature (as per the form of eq. 48) to downscale precipitation fields from lower resolution climate model output (typically about 1000 km for climate models of intermediate complexity to at best 100 km for global general circulation climate models run for paleo contexts in semi-equilibrium time slice mode) that poorly resolves the orography and/or to otherwise add parametric dependence on air temperature (*e.g.,* Tarasov and Peltier, 1997a; Albrecht et al., 2020b). However, though this approximately captures the Clausius-Clapeyron dependence on temperature, it does not account for the orographic forcing of precipitation that can drive higher precipitation at higher elevations and wind-shadowing on leeward sides (*e.g.,* Roe, 2005, for a review). To account for these effects, the GSM uses an orographic down-scaling approach that assumes precipitation corrections for orography on the windward side are proportional to the ratio of mean vertical wind velocities between high resolution and low resolution orographies as diagnosed by the scalar product of the horizontal wind velocities and surface slopes. In detail the windward orographic correction factor (fPorog) is

$$\text{fPorog} = \sum \min(\max((\mathbf{U}_{\text{GCM}} \cdot \mathbf{slope}_{\text{GSM}} + \mu_p)/(\mathbf{U}_{\text{GCM}} \cdot \mathbf{slope}_{\text{GCM}} + \mu_p), \text{FPorogMN}), \text{FPorogMX}) \cdot \text{Uweight} \quad (50)$$

The factor is applied in the downscaling of coarse-gridded input precipitation climatology. The model uses both mean monthly wind velocities as well as standard deviations thereof to account for intra-monthly variability. This involves a summation of mean and mean $\pm 1\sigma$ wind velocities (($\mathbf{U}_{\text{GCM}}$) with weighting (Uweight) as per the corresponding Gaussian distribution. $\mu_p$ is an ensemble parameter that regularizes the orographic forcing. For the leeward side, the orographic forcing factor is set to the regularized difference of vertical velocities:

$$\text{fPorog} = \sum \min(\max((\mathbf{U}_{\text{GCM}} \cdot \mathbf{slope}_{\text{GSM}} - \mathbf{U}_{\text{GCM}} \cdot \mathbf{slope}_{\text{GCM}} + \mu_p)/\mu_p, \text{FPorogMN}), \text{FPorogMX}) \cdot \text{Uweight} \quad (51)$$

The above value of fPorog is further scaled for each GSM grid cell to ensure that precipitation is conserved at the input GCM grid cell scale. As such, the latter is kept purposely coarse (on the order 8 by 4 degrees in longitude and latitude respectively). An analysis of the strong impact of this downscaling approach for fully coupled GSM and climate model simulations is given in Bahadory and Tarasov (2018). This orographic correction is only applied to the components of the precipitation from climate model output.

### 2.10.4 Ocean climate forcing

Ocean temperature forcing is required for both marine calving and submarine melt. For the former, ocean surface temperature is set to the mean sea level summer temperature from the atmospheric forcing with the condition that ocean surface temperature cannot go below the freezing point ($-2^{\circ}$C ). For the latter, either the ocean basal (default) or the average water column (-DToceanDepthAvg) temperature from a low resolution input chronology is horizontally interpolated. For our source chronology, we use the ocean temperature field from the Transient Climate Evolution (TRACE Liu et al., 2009) deglacial simulation carried out with the CESM Earth system model for which only mean decadal annual average ocean temperature fields were

available. The chronology is time interpolated for the last deglacial interval covered by the simulation and otherwise computed by glacial index ($I_g$) weighted interpolation between the full glacial (LGM) and present-day time slices for any other time. To partly address uncertainties in the relationship between the glacial index and the ocean state, the index is subject to an ensemble parameter exponential phase factor ($\Theta_{To}$). As such, the applied glacial index is $I_g^{\Theta_{To}}$.

Given the present-day discrepancies in the TRACE fields, we impose a correction for present-day bias using the ECCO ocean state estimate (Fukumori et al., 2018). This bias correction is subject to an ocean state dependent weighting given by ensemble parameter rToceanBiasCor. The latter specifies the bias correction for glacial index value 1 (LGM). The weighting increases to full bias correction at 0 ka. The factor for the added present-day ocean bias correction is rToceanBiasCor + rw $(1.-\text{rToceanBiasCor})$, with rw given by the square root of the fractional time from LGM to present-day of the TRACE time slice. As the ECCO dataset provides monthly means, we use the average of summer and mean annual ECCO ocean temperatures to partly capture summer season warmth, while retaining some partial consistency with the mean annual temperature fields from TRACE.

After an initial set of history matching waves (Tarasov and Goldstein, 2023; Lecavalier and Tarasov, 2025), it was found that the simulated Antarctic contribution to the Eemian sea level high-stand (generally less than 2 m eustatic equivalent) was inadequate to cover the inferred possible range (*e.g.,* Kopp et al., 2009) even after accounting for potential contributions from Greenland (*e.g.,* Tarasov and Peltier, 2003). As the largest component of relevant climate forcing uncertainty is the subshelf ocean temperature, this inadequacy was assigned to this uncertainty, especially since this required glacial index based extrapolation of TRACE ocean fields. As such, the GSM has an option (-Doceanwarm) to simply add a fraction (given by ensemble parameter rToceanWrm) of the inferred warming at the EPICA ice core site to the 0 ka ocean temperature when the glacial index indicates warmer than 0 ka conditions (index $I_g < 0$).

For Antarctica, an imposed controlling sill depth of 500 m for the Ronne-Filchner sector limits the depth from which the ocean temperature is taken even if the depth of the grid-cell with floating ice is below this.

If the model is fully coupled to an Earth systems model, the GSM can use the ocean temperature field from the ocean model. As detailed in Bahadory and Tarasov (2018), to minimize regridding overhead for complicated ocean model grids, the GSM has a default option of just taking ocean temperature profile chronologies for a number of index sites. The chronologies are then applied to specified downstream sectors of the ocean.

## 2.11 Subgrid ice flow and surface mass balance

The subgrid ice flow and surface mass balance component (inclusion via -DSGhyps and make paleonSG) reduces the subgrid topography for each GSM grid cell with thin or no ice cover to a set of hypsometric curves upon which a fast 1D SIA ice flow and sliding calculation is carried out. Surface mass balance for each hypsometric curve uses the same solver as for the full GSM grid. A unique feature is that this module accounts for subgrid ice flow between adjacent full grid cells. Details on the module design, impact, and validation are in Le Morzadec et al. (2015).

## 2.12 Ice margin nudging

For North America and Eurasia (Hughes et al., 2016; Dalton et al., 2020, 2022), there exist geologically reconstructed ice margin chronologies that include maximum and minimum isochrones for each timeslice. For such a context, the GSM has an option of automatically adjusting the surface mass-balance and calving to favourably nudge the computed ice margin when outside of time interpolated input maximum and minimum isochrones. The number of such grid-cell adjustments is summed for each time-step as a cost function that can be used for model calibration contexts.

The input nudging chronology is a sequence of time slice raster maps on the GSM grid, with value 0 for regions that are definitely ice free, 1 for regions that are likely ice free or ablation zones, 2 for the likely ice margin location, 3 for accumulation zones, and 4 for regions that likely had thick ice well inside of the accumulation zone. With time interpolation between time-slices, these maps provide a nudging field $I_m(x, y, t)$. The nudging is subject to three ensemble parameters ($F_m$, $F_a$, $F_c$, cf Table 2) that specify onset thresholds as well as strength of nudging. Nudging perturbations to the calculated surface mass balance (SMB) are imposed as follows:

$$
\text{SMB} = \begin{cases}
\min(0., \text{SMB} - \text{fmgm} \cdot (2\,F_m - I_m)) & I_m < 2\,F_m \ \wedge \ H > 0. \\
\text{accumulation} & I_m > 4 - 2\,F_a \ \wedge \ \text{grounded} \ \wedge \ \max(H_{adj}) > \text{HmgMx} \ \wedge \ V_b < 100 \text{ m/yr} \\
F_c \cdot \text{SMB} & I_m > 4 - 2\,F_a \ \wedge \ \text{floating} \ \wedge \ \text{active calving with effective SMB} < -5 \text{ m/yr}
\end{cases}
\tag{52}
$$

where HmgMx$= 300$ m. The nudging ablation factor fmgm can be either specified directly in the run script (typically $\leq 10$ m/yr) or more physically specified (compile flag -DnudgMelt) as the product of a constant (typically 2) and the computed gross surface melt. The condition on maximum adjacent ice thickness ($\max(H_{adj}) > \text{HmgMx}$) for accumulation nudging is imposed to avoid the occurrence of "pancake ice" when extended regions in the nudging chronology switch from, for example, neutral zone value 2 to accumulation zone 4. The condition on the magnitude of the basal velocity ($V_b < 100$) inhibits accumulation nudging for active ice streams for which the associated lower surface elevations may physically allow some surface melt.

## 2.13 Subglacial hydrology

As fully detailed and tested in Drew and Tarasov (2023), the GSM has various options for subglacial hydrology. It includes both linked cavity and poro-elastic options for distributed drainage as well as a diagnostic down-pressure-gradient subglacial tunnel solver that thereby avoids CFL constraints which would be prohibitive for glacial cycle modelling.

The GSM also has a much computationally cheaper local 0D hydrology option (enabled with -DNeff0) with a constant drainage rate (given by ensemble parameter rBedDrainRate) leaky bed, and with effective basal pressure ($N_{\text{eff}}$) a non-linear function of basal water thickness as per the poro-elastic version:

$$
N_{\text{eff}} = g\rho_{\text{ice}} H \cdot \left( 1 - \min\left[ \frac{h_{\text{wb}}}{h_{\text{wbCrit}}}, 1.0 \right]^{3.5} \right),
\tag{53}
$$

where $g = 9.81\,\mathrm{m\,s^{-2}}$ is the acceleration due to gravity, $\rho_{\mathrm{ice}} = 910\,\mathrm{kg\,m^{-3}}$ the ice density, $H$ the ice thickness, $h_{\mathrm{wb}}$ the basal water thickness, and $h_{\mathrm{wbCrit}}$ an ensemble parameter for effective bed roughness scale (cf Drew and Tarasov, 2023, for motivation and validation).

Though lacking explicit englacial hydrology, the GSM has a compile flag option (-DZWALLY) to impose local grid cell surface runoff penetration (via assumed moulins) into the local subglacial hydrology system. This assumes that ice is thin enough and crevassed enough for all regions (*i.e.* grid cells) with significant surface runoff to have such englacial hydrological connectivity to the base.

## 2.14    Subglacial sediment production, transport, and deposition

The optional fully coupled subglacial sediment model includes two choices for abrasion representation (Boulton, 1979; Bernard, 1979), quarrying, both subglacial and englacial transport, and deposition. The englacial transport resolves both horizontal and vertical advection within the ice. The model requires basal water pressure from an activated basal hydrology component (section 2.13). The sediment model is fully described and validated in Drew and Tarasov (2024) building on the early version of Melanson et al. (2013). The model can be run in both 1-way (diagnostic) as well as full 2-way coupling with the rest of 905    the GSM. In the latter case, the basal till thickness map for determining the choice between hard and soft basal drag laws dynamically evolves. The changes in surface sediment and bedrock load as well as bed surface elevation can also be fed into the coupled GIA solver.

## 2.15    GIA and sea level

Isostatic adjustment of the bed in response to changes in surface load is computed as per a linear visco-elastic field theory 910    for a spherically symmetric Maxwell model of the Earth (Peltier, 1974, 1976). The bedrock displacement $R(\theta, \psi, t)$ (relative to the Earth's center of mass) is given by a space-time convolution of the surface load per unit area $L(\theta, \Psi, t)$ with a radial displacement Greens function $\Gamma(\gamma, t - t')$ (Peltier, 1974):

$$R(\theta, \psi, t) = \int_{-\infty}^{t} \int \int_{\Omega} L(\theta', \psi', t') \Gamma(\gamma, t - t') d\Omega' dt' \tag{54}$$

Here $\gamma$ is the angular separation between a source point $(\theta', \Psi')$ and field point $(\theta, \Psi)$. The integral is evaluated pseudo-915    spectrally as per (Mitrovica and Peltier, 1991) with triangular truncation at degree and order 256 or 512. This convolution necessitates storage of the discretized load change history.

Bed response is computed every dlong years (default is 100 years). It accounts for all direct changes in surface load, including ice, lake water, and seawater within the ice sheet grid. The GIA calculation requires global surface load change inputs. Therefore, outside of regions covered by the simulation, surface ice load changes follow an input global chronology (cur-920    rently GLAC2A) for the last glacial cycle and a sea level weighted interpolation between input PD and LGM states for prior time intervals. Not accounting for global ice load changes in Greenland simulations (as used to be typical for paleo ice sheet modelling, *e.g.,* Tarasov and Peltier, 2002; Lecavalier et al., 2014), can have significant impacts (cf Figs. 7 and 8).

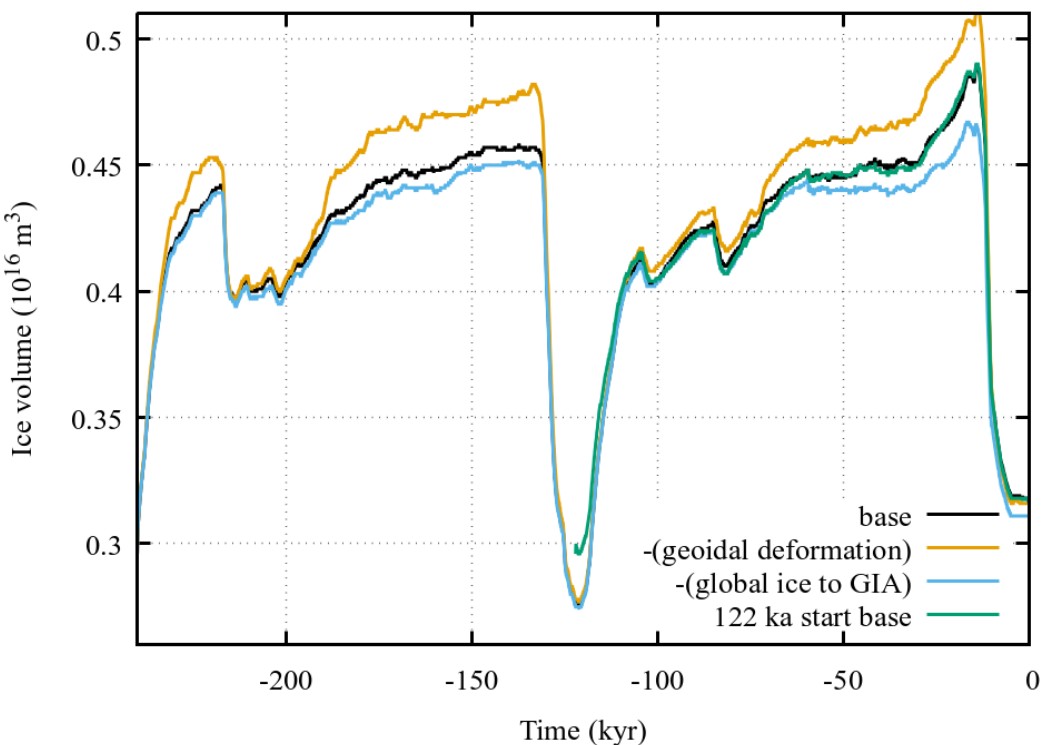

**Figure 7.** Example process removal sensitivities as given by mean response for a 10 member GRIS ensemble. The listed time series are identified by the GSM process that has been removed.

Surface load and elevation changes due to subglacial erosion and sediment transport can also be activated with the -DdynSed compile flag (for details, c.f. Drew and Tarasov, 2024).

The load history must be stored as spherical harmonic coefficients and thereby represents a major memory load. To limit the required memory, the GSM only retains a specified past interval of load history at 100 year resolution (default 30 kyr) and then a subsequent second interval at either 500 year or 1 kyr resolution (default 210 kyr). Ice load changes prior to this second long time-step interval are continually summed and imposed as a step load. This approach requires the load intervals to be stored in first in first out (FIFO) memory stacks as well as the tracking of the variable time interval between the current time-step and

the load steps in the long time-step stack.

     This load history treatment has been verified with simple two step instant GRIS unloading tests (with half of the ice removed after 100 years, and the rest removed after 2.5 kyr). After 60 kyr, the maximum bed elevation difference between the default (100 year step for 30 kyr, then 1 kyr steps) versus a continuous 100 year load step configuration is $< 2$ cm for a relatively soft earth rheology ($2 \times 10^{20}$ and $3 \times 10^{21}$ Pa s respective upper and lower mantle viscosities and $< 41$ cm for an extremely stiff

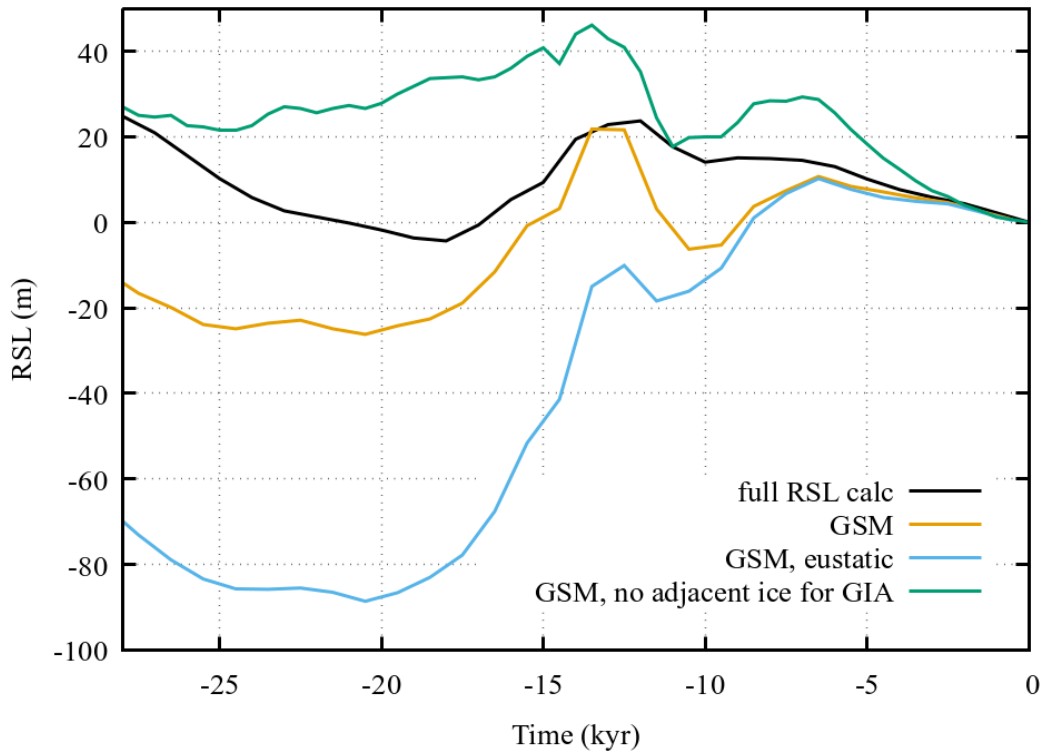

**Figure 8.** Comparison of computed RSL for Franz Joseph Fjord Greenland ($27.42^o$ west, $73.02^o$ N). Shown are: the gravitationally-self consistent solution from post-processing ("full RSL calc"), the GSM internal solution using linear geoidal deflection ("GSM"), the solution when geoidal deflection is typically neglected ("GSM, eustatic"), and the GSM solution when ice outside of the regional grid is not accounted for in the visco-elastic bedrock response calculation ("GSM, no adjacent ice for GIA").

earth rheology (respectively $5 \times 10^{21}$ and $30 \times 10^{21}$ Pa s). However, this is not the case for a transient fully coupled simulation of the GRIS. In this case, high sensitivity of the coupled system can be evident depending on the ensemble parameters. As shown for an example parameter vector (chosen for higher sensitivity) in Fig. 9, the simulated LGM ice volume anomaly relative to 0 ka can vary up to $\approx 10\%$ depending on the choice of intervals for the 100 year and either 500 or 1000 year load storage steps. Furthermore, the response to the number of time-steps is not monotonic, with eg 10 kyr coverage of the most recent load

changes at 100 year time-steps performing worse for this case (relative to the maximum 38 kyr coverage simulation) than the simulation with 1 kyr coverage (NTIM1=10 in Fig. 9). These results show higher sensitivity to the GIA load history time-step than that of a previous study that also implemented variable load history time-steps Han et al. (2022). However the latter only used 200 years as their shortest GIA load history time-step, kept their short time-step interval to a fixed 5 kyr and based their whole analysis on a single North American ice sheet configuration.

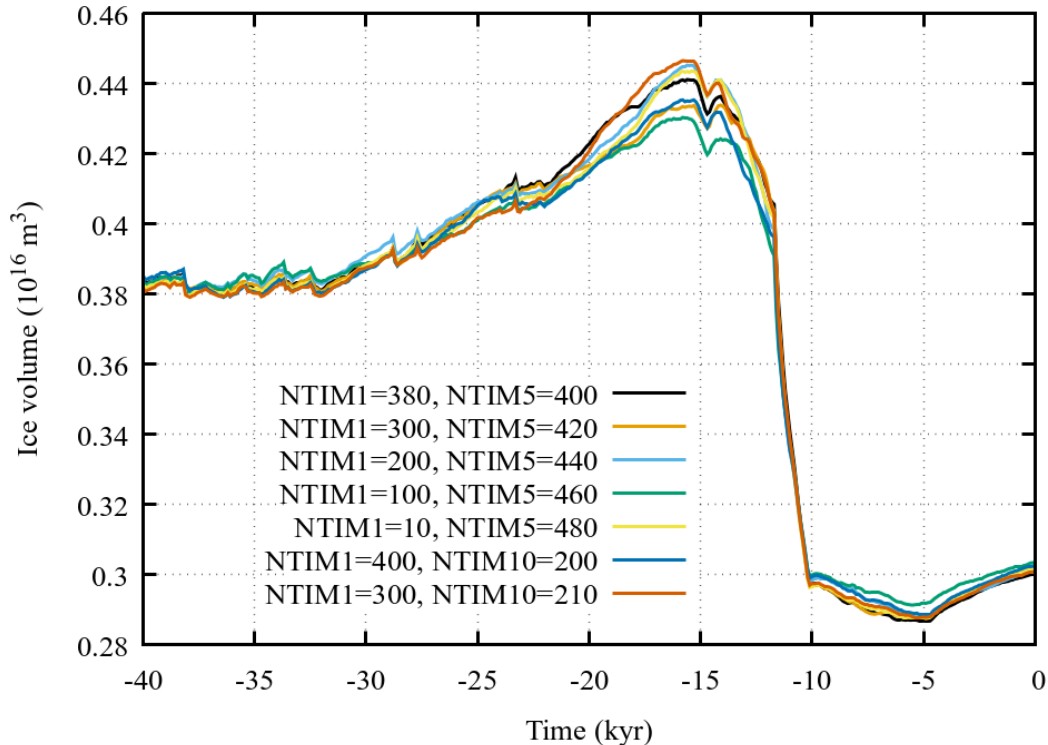

**Figure 9.** Example GRIS ice volume sensitivity to the number of 100 year (NTIM1) and either 500 year (NTIM5) or 1000 year (NTIM10) load history steps for the GIA calculation in the GSM. These 240 kyr simulations used a moderately soft earth rheology with $3 \times 10^{20}$ and $2 \times 10^{21}$ Pa s respective upper and lower mantle viscosities.

The GSM includes a small archive of Earth model Greens functions (in the form of Love number sets, from Love et al., 2016) nominally specified by 3 ensemble parameters for the lithospheric thickness and upper and lower mantle viscosities. The Earth model Love numbers were computed using the mixed collocation method as per Mitrovica and Peltier (1992). The radial elastic structure for this set is that determined from seismic constraints (PREM Dziewonski and Anderson, 1981).

     The GSM has an option to add a GIA correction to the input present-day topography ($h_{\mathrm{bo}}$) for run initialization to partially 950   correct for discrepancies between $h_{\mathrm{bo}}$ and the resultant 0 ka topography from a transient simulation due to present-day isostatic disequilibrium. The corrected initial bed topography ($h_{\mathrm{boc}}$) is implemented as

$$h_{\mathrm{boc}}^n = (1+n) \cdot h_{\mathrm{bo}} - \sum_{i=1}^{i=n} h_{\mathrm{bf}}^i \tag{55}$$

where $n$ is the number of iterated full glacial cycle simulations applied to create the present-day discrepancy correction, $h_{\text{bf}}^n$ is the 0 ka final bed topography from a transient simulation using the previous iteration of the corrected initial topography $h_{\text{boc}}^{n-1}$.

This approach can be justified inductively, starting from $h_{\text{boc}}^1 = h_{\text{bo}} - (h_{\text{bof}}^1 - h_{\text{bo}})$.

For ice sheets with no significant present-day ice cover, these corrections are implemented as the average of two different not-ruled-out-yet simulations from previous history matching iterations (for an introduction to history matching, cf Tarasov and Goldstein, 2023). These correction fields need to be regenerated for different Earth model viscosity profiles (at least for those that give more than 2 kyr relaxation times). Depending on the Earth viscosity, one to two iterations are generally adequate in that improvements. For instance, for an NROY test set of 10 GSM parameter vectors, the root-mean-squared-error in 0 ka (0 ka RMSE) surface topography relative to the input present day topography using the order 1 topography correction was on average 3.1 m with a maximum value of 5.3 m (upper mantle viscosities were $< 1 \times 10^{21}$ Pa s, lower ranged from $1 \times 10^{21}$ to $30 \times 10^{21}$ Pa s. This correction to Eemian topography can have a significant impact on simulated NAIS evolution (cf Fig. 5).

For ice sheets with extensive present-day ice cover, the sensitivity of the correction to differences in simulated 0 ka ice thickness are too strong for the use of common correction fields ($h_{\text{boc}}^n$) for a given earth rheology. For this case, the correction fields have to be extracted for each GSM parameter vector (as is also done in van Calcar et al., 2023). For the case of the GRIS, initial exploratory experiments with a 10 member NROY parameter vector set indicate one correction iteration can be marginally adequate, with *e.g.,* 0 ka RMSE differences between gridded observed and modelled (after a 240 kyr transient simulation) subglacial bed topography of less than 16 m. Two iterations brought this down to $< 3$ m for the majority of simulations. However some simulations diverged when 3 or more iterations were applied. Damping of the corrections fields (up to 25%) was tested but did no improve convergence.

The error for not accounting for this can be significant, with one test parameter vector in last two glacial cycle simulations giving differences in GRIS 0 ka ice surface elevation RMSE of 54 m and a reduction in the Eemian high-stand contribution by a factor of 2 (from 1.4 to 0.6 mESL).

The GSM computes spatial variations in the geoid in response to changing (ice/water/earth) mass distribution with a linear approximation. The model modifies the mean volumetric (eustatic) sea level change with a spatially varying geoidal deflection computed as linear contributions from each of the 4 major ice sheets. For ice sheets not modelled, default chronologies for the deflection contributions are read in (currently based on gravitationally-self-consistent post-processing of interim GLAC3 ice sheet chronologies). For the actively modelled ice sheet (referenced by ice sheet index $k_i$), the deflection ($G_d$) is simply a relative volume anomaly scaling of the reference input time-interpolated deflection ($G_{dInterp}(x,y,t,k_i,k_s)$) contribution from each major ice sheet (referenced by ice sheet index $k_s$):

$$
\begin{aligned}
G_d(x,y,t,k_i) \quad = \quad & \sum_{k_s} [\, G_{dInterp}(x,y,t,k_i,k_s) \cdot (\mu_G + \max(vol(t,k_s) - vol_{Ref}(0ka,k_s)\,,\,0.))/ \\
& (\mu_G + \max(vol_{Ref}(t,k_s) - vol_{Ref}(0ka,k_s)\,,\,0.))\,]
\end{aligned}
\tag{56}
$$

where $\mu_G$ is a small regularization parameter (with value dependent on the ice sheet). Prior to the last glacial cycle, the geoidal deflection contribution from each ice sheet is a volume anomaly (relative to input present-day ice volume) weighted

**Table 5.** GSM variables and inputs subject to noise insertion with -DIDassess.

| process | max amplitude | dependency |
| --- | --- | --- |
| deep geothermal heat flux boundary condition | $\pm 5\%$ | input(x,y) |
| initial ice temperature for thermodynamic spinup | $\pm 1^{o}C$ | input(x,y) |
| surface insolation for snow/ice melt | $\pm 5\%$ | x,y,t |
| precipitation | $\pm 20\%$ | x,y,t |
| annual glacial temperature index (Tdiffin) | $\pm 1^{o}C$ | input(t) |
| monthly glacial index (vTdiffin) | $\pm 10\%$ | input(t) |
| glacial index for pre-LGM phasing of ocean temperature | $\pm 10\%$ | t |
| sea level chronology | $\pm 3\%$ | input(t before 6.5ka) |
| effective pressure for basal drag | $\pm 10\%$ | x,y,t |
| basal roughness map used for basal drag | $\pm 20\% + \pm 10m$ | input(x,y) |
| Subgrid sliding coefficient | $\pm 20\%$ | x,y,z,t |

interpolation between reference geoidal stadial and interstadial time slices. These reference time slices are chosen by matching reconstructed sea level low and high-stands to that of the last glacial cycle.

The geoidal deflection in the GSM is a 0 order approximation. It is better than the typical purely eustatic assumption as shown in Figs. 7 and 8. However, for comparison to RSL data, post-processing of the simulation output with a gravitationally-self consistent solver is necessary. The upgrade of the GSM to a gravitationally-self consistent coupled solution is technically not a major challenge and will likely be added in a future version.

The mean sea level in the GSM can be an input or determined from ice volume changes in the GSM with scalings for missing ice sheets.

The GSM also has a simple local relaxation option for GIA. This is useful for GSM testing as well as for running on non-geographic grids (such as for idealized model inter-comparison experiments).

### 2.16 Noise injection for internal discrepancy assessment or direct noise sensitivity analysis

The GSM has a compile flag (-DIDassess) for activation of noise injection into various poorly constrained component processes and inputs (as listed in Table 5). This can be used for internal discrepancy assessment (Tarasov and Goldstein, 2023) to quantify associated structural uncertainties of the GSM or for sensitivity experiments directly analyzing unresolved process noise impacts. The noise is generated as a sign preserving square of a uniform sampling (-range:range) to ensure substantial noise density near amplitude bounds while concentrating the distribution around 0. The choice of noise distribution and amplitude was based on informed author judgment but should be reconsidered by any user based on context and confidence in relevant inputs. An example of an auto-regressive Gaussian noise approach applied to a narrower range of variables is provided by Verjans et al. (2022).

## 2.17 GSM input and core internal fields

Table 6 provides a brief summary of the input data sets used in the GSM. When the GSM is fully coupled to an EMIC or more advanced climate model (*e.g.,* Bahadory and Tarasov, 2018), all of the climate related inputs will by dynamically taken from the climate model. When run in a global configuration, no input sea level chronology is required.

The key internal fields of the GSM that are passed between GSM components are listed in Table 7. Not listed are all the atmospheric climate and solar forcing related fields (precipitation, evaporation, insolation, climatological winds for precipitation downscaling). Nor are most fields that are derived from the listed fields shown. For instance, the flotation mask (maskwater(x,y)) is derived from $H$ and $h_{bG}$.

## 2.18 GSM initialization

The appropriate initialization of the temperature field in an ice sheet model is, to date, unsettled (*e.g.,* Goelzer et al., 2018; Seroussi et al., 2019). After extensive testing, the following approach was chosen. The initial ice temperature for the Greenland and Antarctic ice sheets is set to analytical approximations of the respective GRIP and EDC ice core borehole temperature profiles scaled from the applied surface temperature to a basal temperature of -6°C for Greenland and -4°C for Antarctica. The choice of the profile locations is motivated by ice dome centres having the slowest velocities and therefore the longest relaxation times to approach a self-consistent vertical temperature profile. This initial temperature profile will become more inaccurate farther away from ice dome centres but this will be compensated by faster evolution to a more self-consistent vertical temperature profile given the higher ice velocities. Setting the whole ice sheet initial basal temperature below the pressure melting point stabilizes the initial ice dynamical solution. The ice sheet velocities are then computed with an SIA solution, and the ice/bed thermodynamics is brought towards partial thermal equilibration (over 1 and 1.5 kyr intervals for Greenland and Antarctica respectively). This is facilitated by temporarily reducing the bed heat capacity by a factor of 1000. The fully coupled hybrid ice dynamics and thermodynamics is then stepped over the asynchronous coupling time-step (dlong years, cf section A2 for general GSM code structure), after which thermal partial-equilibration is advanced with the updated ice velocity field (for respectively 3 and 6 kyr intervals). Reduced spinup time intervals are used for other ice sheets and ice caps.

For a set of history matched not-ruled-out-yet Antarctic and Greenland simulations examined, this spinup approach tends to bring the basal warm-based ice fraction to within half of the glacial cycle maximum value. For Antarctica, this spinup approach works especially well for Eemian cold starts. For at least the example run shown in Fig. 10, the grounded ice volume history for a 122 ka to 0 ka simulation is nearly the same between the cold-started run and a run started with the terminal restart file from a 205 ka to 0 ka prior simulation (using the same parameter vector). The approach is sensitive to the initial climate forcing as is evident in the difference between simulations with 206 and 204 ka cold starts (Fig. 10). The O(100kyr) memory in these results is in accordance with the thermodynamical time scale of the Antarctic ice sheet. Given the order of magnitude larger precipitation for Greenland (which reduces the thermal equilibration time scale), this spinup approach results in minimal ensemble mean differences for post 110 ka ice volumes for cold start simulations starting at 240 ka and 122 ka (Fig. 7).

**Table 6.** GSM input data sets. All climate fields are in the form of monthly climatologies.

| component | data source |
|---|---|
| present day mean temperature | reanalysis or RCM output |
| LGM mean temperature | PMIP I to III ensembles (Braconnot et al., 2007, 2012) |
| present day mean precipitation | reanalysis or RCM output |
| LGM mean precipitation | PMIP I to III ensembles |
| present day mean evaporation | reanalysis or RCM output |
| LGM mean evaporation | PMIP I to III ensembles |
| present-day and LGM wind field horizontal components: | |
| monthly means and standard deviations | PMIP III |
| orographic surface slopes from precip input source | PMIP III |
| 4D ocean temperature chronology | TRACE (Liu et al., 2009) |
| present-day ocean temperature field: | |
| present-day ocean temperature field: Antarctica | Whole Antarctica Ocean Model (Richter et al., 2022; Boeira Dias et al., 2023) |
| present-day ocean temperature field: rest | ECCO (Forget et al., 2015) |
| Northern Hemisphere climate index | inversion of ice core isotopic records (NGRIP dating group, 2006) |
| | (North Greenland Ice Core Project members, 2004; Barker et al., 2011) |
| Southern Hemisphere climate index | inversion of EDC deuterium excess on AICC2012 chronology |
| | (Jouzel et al., 2007; Bazin et al., 2013) |
| bed and surface topography | user choice |
| hydrologically corrected bed topography | modified HYDRO1K DEM (USGS, 2004; Tarasov and Peltier, 2006) |
| bed subgrid standard deviation | user choice |
| sediment thickness map | (Laske and Masters, 1997) |
| fractional subgrid sediment cover map for till deformation | derived from sediment map (Laske and Masters, 1997) |
| | and surficial geology map (Fulton, 1995, for NA) |
| deep geothermal heat flux map(s) | global map of Davies (2013) and/or regional options |
| Earth model love numbers for GIA calculation | (Love et al., 2016) |
| Earth radial elastic structure | PREM model (Dziewonski and Anderson, 1981) |
| eustatic sea level chronology | Fairbanks (1989); Waelbroeck et al. (2002); Lisiecki and Raymo (2005); |
| | Peltier and Fairbanks (2006); Lambeck et al. (2014) |

**Table 7.** Key (mostly) primary GSM dynamical fields. All fields evolve with year. Most surface mass-balance components are computed monthly, but the resultant smb field is passed to the ice dynamic solver as a mean yearly rate.

| definition | field | units |
|---|---|---|
| monthly mean 2 m air temperature | $T_{2m}(x, y, t_m)$ | °C |
| ice temperature | $\text{Ti}(x, y, z)$ | °C |
| basal ice temperature with respect to the pressure melting point | $T_{\text{bmp}}(x, y)$ | °C |
| basal melt (not including submarine melt) | $\text{Gb}(x, y)$ | m/yr |
| ice velocity | $\mathbf{V} = (u, v, w)(x, y, z)$ | m/yr |
| basal stress | $\tau_b(x, y)$ | Pa |
| surface mass balance | $\text{smb}(x, y)$ | m/yr |
| net submarine melt and refreezing | $\text{ssm}(x, y)$ | m/yr |
| calving rate | $\text{cmb}(x, y)$ | m/yr |
| ice thickness | $\text{H}(x, y)$ | m |
| bed elevation relative to present-day sea level | $h_b(x, y)$ | m |
| sea surface geoid | $\text{sealev}(x, y)$ | m |
| bed elevation relative to contemporaneous sea level | $h_{\text{bG}} = h_b - \text{ sea level}$ | m |
| surface elevation relative to contemporaneous sea level | $h$ | m |
| lake depth | $D_L$ | m |
| effective basal pressure | $N_{\text{eff}}$ | Pa |
| subglacial sediment thickness | $H_{\text{sed}}$ | m |
| cumulative basal erosion | $H_{\text{Eros}}$ | m |

## 3    Tests and sensitivities

As the core hybrid SIA/SSA ice dynamics solver is from the PSU3D model (Pollard et al., 2015). It has already been exten-
sively tested in ice sheet model intercomparisons (Pattyn et al., 2012, 2013; Cornford et al., 2020). A repeat of some of the
experimental tests in the latter two (MISMIP3D and MISMIP+) intercomparisons with the revised 2D grounding line buttress-
ing treatment in the solver has results within the envelope of the higher-order higher-resolution models in the intercomparisons
(Pollard and DeConto, 2020).

For partial verification of the coupled ice dynamics and thermodynamics, we compare GSM results against a few of the EIS-
MINT (Payne et al., 2000) and HEINO (Calov et al., 2010) SIA model intercomparison results. For the EISMINT experiment
G with basal sliding activated everywhere, the GSM simulations preserve the x and y axis symmetry of the forcing (not shown)
with most statistics in Table 8 close to the mean value of the intercomparison results in Payne et al. (2000). The one exception
is the areal fraction of warm-based ice being just above the minimum value in the intercomparison (note the outlier EISMINT
model "U" was removed when calculating means and ranges). This one partial discrepancy is likely attributed to basal sliding
given that the GSM result for the EISMINT A experiment with no basal sliding gives all statistics very close to that of the

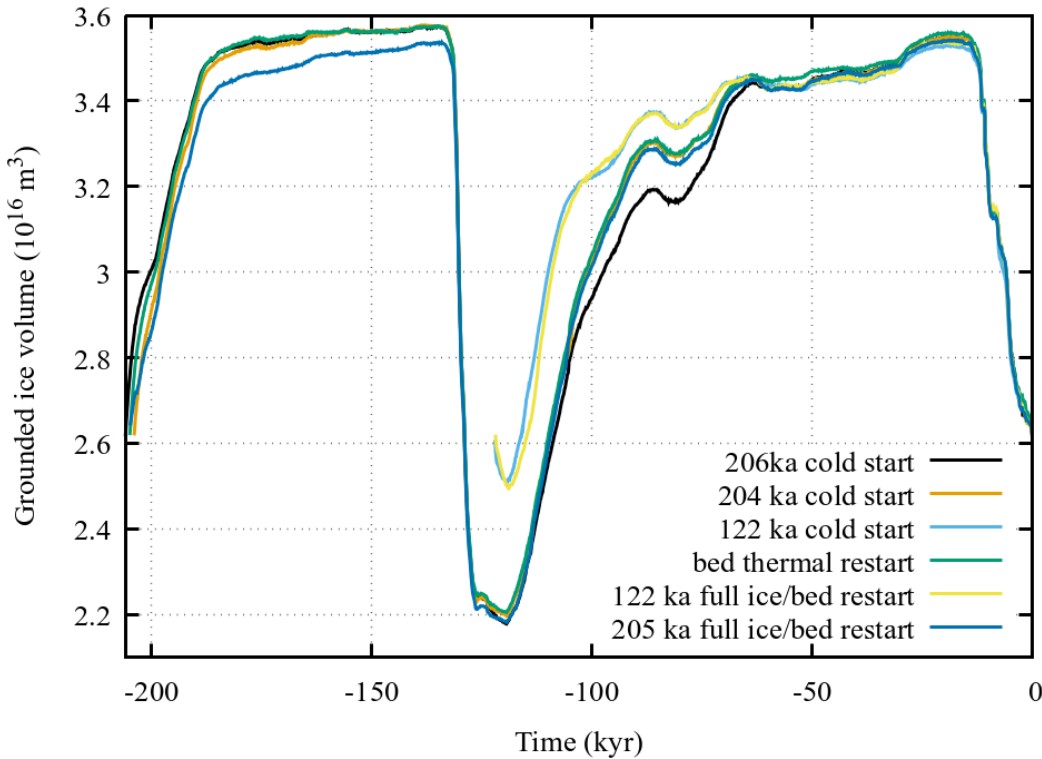

**Figure 10.** Example AIS ice volume history sensitivity to initialization. As specified in the legend, the last three chronologies use simulations that were initialized from the indicated restart output at the end of full 205 kyr simulations.

intercomparison ensemble mean (not shown). Imposition of full SSA ice dynamics everywhere for this configuration induces a slight (3%) decrease in ice volume and even slighter (1.5%) increase in ice area (Table 8).

For EISMINT experiment H with thermal activation of basal sliding, all EISMINT and GSM simulations do not equilibrate. GSM maximum and minimum statistics from the last 20 kyr of simulation are within the range of reported EISMINT end of simulation results. However the GSM warm-based fraction is $0.04$ below the minimum EISMINT value (only the final time-step values are available from Payne et al., 2000). Similar to the majority of EISMINT models, the GSM experiment H lacks full x and y axis symmetry (not shown).

To further verify the current coupled ice dynamics/thermodynamics, the standard ISMIP HEINO (Calov et al., 2010) experiment was repeated with various modifications. The climate forcing and boundary conditions for this experiment have full y-axis symmetry but the basal boundary condition is x-axis asymmetric. When run in base hybrid (SIA/SSA) configuration, the simulation has near complete y-axis symmetry (Fig. 11). Symmetry can be further enhanced if basal sliding is activated every-

**Table 8.** Comparison of GSM results (after 200 kyr simulations) against comparable results for SIA EISMINT G and H experiments (Payne et al., 2000). Listed "EISMINT" results have outlier model U removed. For the GSM SIA experiment H only, the listed minimum and maximum are from the last 20 kyr of simulation (since experiment H never reaches steady state for any model, and EISMINT only provided the last time-step results). Both experiments have the same rotationally symmetric climate forcing (and flat bed). The only difference is basal sliding is activated everywhere for experiment G while experiment H uses standard basal temperature dependent sliding activation.

| | Volume $10^{15} m^3$ | Area $10^{12} m^2$ | Warm-based fraction | Divide thickness m | Divide basal temperature $^o$C |
|---|---|---|---|---|---|
| **experiment G** | | | | | |
| GSM SIA | 1.532 | 1.016 | 0.261 | 2223.65 | -24.74 |
| GSM SSA | 1.487 | 1.031 | 0.235 | 2199.30 | -24.64 |
| mean EISMINT | 1.520 | 1.026 | 0.301 | 2233.2 | -24.65 |
| min EISMINT | 1.503 | 1.016 | 0.250 | 2212.6 | -25.45 |
| max EISMINT | 1.533 | 1.032 | 0.351 | 2228.3 | -23.65 |
| **experiment H** | | | | | |
| GSM SIA min | 2.016 | 1.0094 | 0.313 | 3614.8 | -17.42 |
| GSM SIA max | 2.035 | 1.0288 | 0.406 | 3646.6 | -17.41 |
| min EISMINT | 1.744 | 1.020 | 0.351 | 3433.1 | -19.22 |
| max EISMINT | 2.034 | 1.032 | 0.622 | 3645.3 | -16.78 |

where (-DwarmBaseEverywhere, Fig. 12). Complete symmetry is obtained when all grid cells are treated as SSA (-DSSAall, not shown).

As for other GSM components, the GSM subgrid hypsometric ice dynamics component verification was carried out in the source reference (Le Morzadec et al., 2015). The surface drainage solver verification against present-day drainage basins is in Tarasov and Peltier (2006). Numerical sensitivity analysis for surge cycling is extensively examined in Hank et al. (2023). For partial verification of the whole ice sheet system, Lecavalier and Tarasov (2025) provides a recent application of the GSM to history matching of the last glacial cycle Antarctic ice sheet that includes comparisons to present-day observations. For convenience, a comparison of 0 ka ice thickness from a sample 205 kyr transient simulation against observations (as inferred BedMachine V2 Morlighem et al., 2019) is presented in section 2 of the supplement.

Given the nonlinearities in the system, ensemble parameter sensitivities are assessed by automatic relevance determination (Neal, 1996) during the history matching iterations for each paleo ice sheet. For the purposes herein, a simple one at a time parameter sensitivity analysis is provided in the supplement for the Greenland (GRIS), North American (NAIS), and Antarctic (AIS) ice sheets. Collectively, these demonstrate that each ensemble parameter has significant impact for at least one metric component.

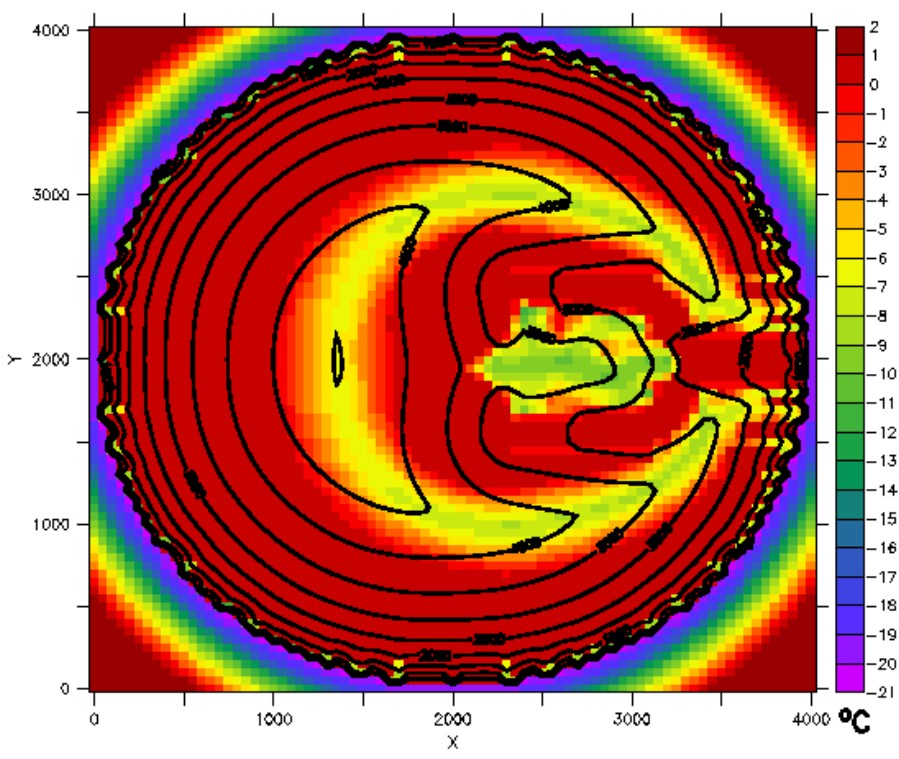

Basal temperature relative to pressure melting point

**Figure 11.** GSM basal temperature relative to the pressure melting point and ice thickness contours for the HEINO experiment with hybrid SIA/SSA.

## 4   Summary discussion and conclusions

The GSM has specific relative strengths and weaknesses compared to other available ice sheet models. The GSM is specifically designed for paleo contexts; where forcing uncertainties necessitate relatively large ensembles of runs along with appropriate methodologies to assess uncertainties and infer parameter vectors consistent with available proxy and observational constraints. As such, it hasn't been parallelized. This can put a strain on available memory resources depending on cluster configuration. For modelling continental-scale ice sheet response to ongoing and projected climate change where high grid resolution (5 km or higher) is much more important than computational cost as well as for coupling to parallelized Earth system models, parallelized ice sheet models such as ISSM (Larour et al., 2012), CISM (Lipscomb et al., 2019), , or BISICLES (Cornford et al.,

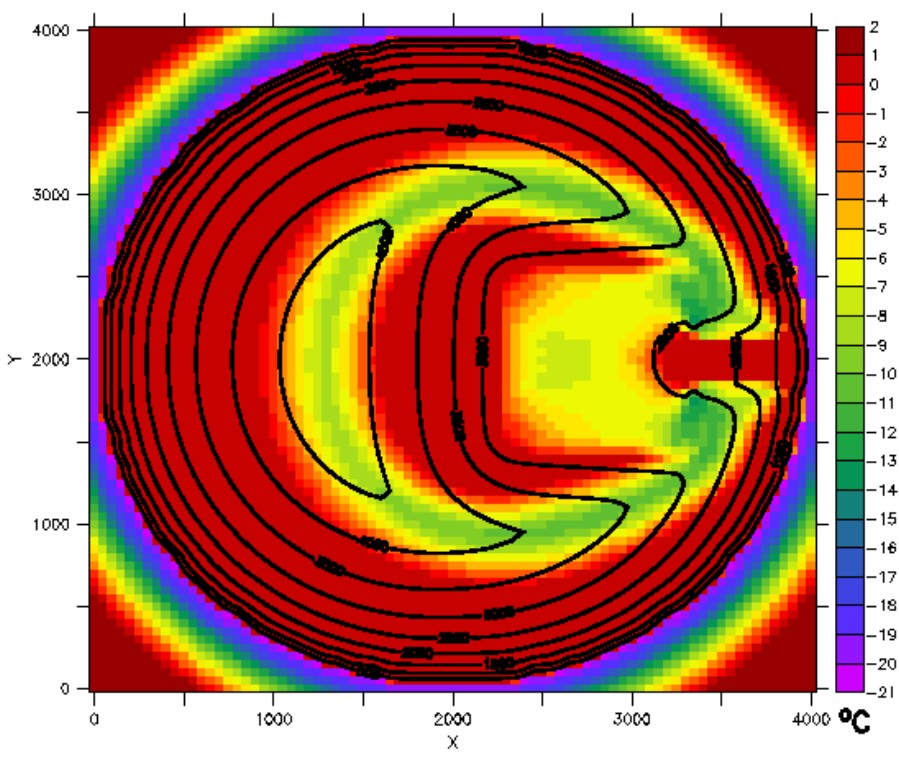

**Figure 12.** GSM basal temperature relative to the pressure melting point and ice thickness contours for the HEINO experiment with hybrid SIA/SSA and full activation of basal sliding everywhere.

2013) that include the higher order ice dynamical solutions that such high resolutions require would be advised. The BISICLES
model is especially noteworthy given its dynamically adaptive grid. Though lacking higher order physics, UFEMISM (Berends et al., 2021) may be the computationally most efficient adaptive grid model enabling 4 km grid resolution around grounding lines and outlet glaciers at relatively moderate computational cost (88 core hours for an example glacial cycle Antarctic simulation Berends et al., 2021). Over the long-term, it would be advantageous (albeit challenging) for the ice sheet modelling community to move towards plug-and-play compatibility between different components.
The GSM's key strengths for the paleo modelling context are the breadth of relevant incorporated processes, relatively large degrees of freedom in the climate forcing components, and relatively low computational cost (with *e.g.,* an approximately 10

hour wall clock time for a 122 kyr Antarctic simulation at 40 km resolution on a single circa 2016 Intel core including global visco-elastic GIA). The first two features both helps minimize structural uncertainties as well as enable simulation comparison against a wide range of paleo proxies. The GSM's computational speed facilitates large ensemble modelling. The GSM's design focus on addressing structural uncertainties for glacial cycle modelling context is also reflected in the inclusion of process noise injection for internal discrepancy assessment.

A novel feature of the GSM is the PDDsw surface melt scheme that explicitly imposes the physical constraint that shortwave insolation only contributes to surface melt when the surface temperature is at $0^o$ C. This thereby enables explicit accounting of changes in spatio-temporal insolation on surface melt. It also enables future inclusion of the surface melt impact of surface dust accumulation via changes in surface albedo.

We have demonstrated the importance of three features of the GSM GIA component that are often ignored for glacial cycle contexts. The geoidal deformation feature will in the future be upgraded to a full pseudo-spectral calculation. The second feature, input of global ice load outside of the ice sheet grid, necessitates some confidence in the global chronology or a move to global ice sheet modelling. The third feature, correcting for present-day isostatic disequilibrium when specifying the initial (Eemian or earlier) topography, is relatively simple and cheap to implement for ice sheets that are presently absent given its sole dependence on the earth rheology model. For presently existing ice sheets, this topographic correction is much more expensive as it requires a recursive solution of the fully glacial simulation for each different ensemble parameter.

The GSM is to date only the second coupled visco-elastic GIA and ice sheet model with variable load history time resolution (the first being Han et al., 2022) and the only to date that uses a memory stack to minimize the expensive memory requirements for load histories. For coupling to much more computationally expensive 3D GIA models, the iterative approach of van Calcar et al. (2023) reduces the number of GIA model step at the cost of repeat iteration of the the ice sheet dynamics The implementation in the GSM currently only allows one user specified interval of 100 year storage resolution, and one (farther back in time) with either 500 or 1000 year resolution. An explicit examples shows that even a 30 kyr 100 year load step coverage can give 6% discrepancies in the GRIS LGM ice volume anomaly relative to a simulation with 38 kyr coverage (with both using 500 year load steps for the remainder). Given that the above time resolutions are finer than any other coupled ice sheet and visco-elastic GIA models for glacial cycle contexts, this sensitivity of the coupled system points to a modelling challenge for the community.

We have also demonstrated the significant impact of changes in the specification of the basal sliding activation function for Antarctic (Fig. B2) and more so for Greenland (Fig. 2) ice sheet simulations (also cf section 2.5.2). As such, we suggest the appropriate specification of this function (cf Hank et al., 2023) needs more attention within the ice sheet modelling community.

GSM development is ongoing to better ensure that such uncertainties are more confidently bracketed within the range of ensemble parameters and associated process representations with the model. Aside from the over-riding challenge of appropriately representing climate (both atmospheric and oceanic) over glacial cycles, the other least confident components of the GSM (as well as other paleo ice sheet models, if even addressed) are the following. 1) Basal drag as a function of basal roughness, bed geology, and mean sediment thickness (and perhaps class of sediment: clay, till, ...). 2) Subshelf and fjord water temperature, circulation, and salinity and how these fields drive subshelf melt. 3) Lacustrine melt and calving. Aside from climate

inputs, the deep geothermal heat flux is a poorly constrained input field for all ice sheets (with potentially significant impact on processes such as Hudson ice stream surge cycling, as shown in Hank and Tarasov, 2024). For regions with present-day ice cover, bed elevation and subgrid bed roughness still have much room for improvement though have already benefited from ongoing efforts (*e.g.,* Morlighem et al., 2017).

The next priority addition to the GSM will be efficient ice age tracing (Rieckh et al., 2024), to enable comparison of simulations against isochronal depth inversions (MacGregor et al., 2015). The other outstanding addition is a fully coupled glaciogenic dust production and deposition on ice (*e.g.,* as in Ganopolski et al., 2010).

The GSM is currently only available as a tarball with updates available on the lead author's website as per the code availability statement below . Depending on community interest and involvement, a GitHub for the model will likely be created in the near future.

*Code and data availability.* A code and input data archive for the GSM is available on Zenodo (https://doi.org/10.5281/zenodo.14599678, Tarasov et al, 2025). Code updates will be available from the first author's website https://www.physics.mun.ca/~lev/software.html . A tarball of model output for the figures in the text is attached as an asset for this submission.

## Appendix A: more technical GSM details

### A1 Hybrid SIA/SSA ice dynamics

The heuristic combination of the depth-integrated Shallow Ice Approximation (SIA, vertical shearing) and Shallow Shelf Approximation (SSA, horizontal longitudinal stretching) ice dynamics equations follows (Pollard and DeConto, 2012). The two sets of equations can be linked to each other in three ways:

1. inclusion of shear softening terms when calculating the effective viscosity

2. distinction between the depth-averaged internal-shear and basal velocity in the SSA basal stress term

3. reduction of the SIA driving stress by horizontal shear and longitudinal stress gradient terms from the SSA equations

Each of the three SSA/SIA couplings can be turned on/off individually using compile flags. -DNOSOFTCROSS and -DNOLHSCROSS turn off coupling options 1 and 3, respectively. -DUIACROSS turns on option 2. When using all three coupling options, the SIA-like internal shear equation in x-direction is calculated according to

$$\frac{\partial u_i}{\partial z} = 2A \left[ \sigma_{xz}^2 + \sigma_{yz}^2 + \sigma_{xx}^2 + \sigma_{yy}^2 + \sigma_{xy}^2 + \sigma_{xx}\sigma_{yy} \right]^{\frac{n-1}{2}} \sigma_{xz} \tag{A1}$$

where $\sigma_{ij}$ are the deviatoric stresses. The SSA-like horizontal stretching equation in x-direction is

$$\frac{\partial}{\partial x} \left[ \frac{2\mu H}{\bar{A}^{1/n}} \left( 2\frac{\partial \bar{u}}{\partial x} + \frac{\partial \bar{v}}{\partial y} \right) \right] + \frac{\partial}{\partial y} \left[ \frac{\mu H}{\bar{A}^{1/n}} \left( \frac{\partial \bar{u}}{\partial y} + \frac{\partial \bar{v}}{\partial x} \right) \right] + \tau_{bx} = \rho_i g H \frac{\partial h_s}{\partial x} \tag{A2}$$

where $u_b = \bar{u} - \bar{u}_i$ with bars indicating vertical averages. A similar expression can be found in y-direction and a list of symbols is provided in Table A1. The SSA stress balance boundary condition at a floating ice margin is implemented effectively as in Winkelmann et al. (2011) with appropriate densities for ocean or lake. For tidewater ice margins, the subgrid ice margin is assumed to be at flotation, and the grounding line flux condition (Schoof, 2007; Tsai et al., 2015) provides the effective boundary condition.

The reduction of the SIA vertical driving stress follows

$$\sigma_{xz} = -\left(\rho_i g H \frac{\partial h_s}{\partial x} - \text{LHS}_x\right)\left(\frac{h_s - z}{H}\right) \tag{A3}$$

where $\text{LHS}_x$ is the left hand side in Eq. A2. A similar expression can be found for $\sigma_{yz}$. The effective viscosity $\mu$ (including the shear softening term) in Eq. A2 is determined by

$$\mu = \frac{1}{2}\left(\dot{\epsilon}^2\right)^{\frac{1-n}{2n}} \tag{A4}$$

with

$$\dot{\epsilon}^2 \approx \left(\frac{\partial \bar{u}}{\partial x}\right)^2 + \left(\frac{\partial \bar{v}}{\partial y}\right)^2 + \frac{\partial \bar{u}}{\partial x}\frac{\partial \bar{v}}{\partial y} + \frac{1}{4}\left(\frac{\partial \bar{u}}{\partial x}\frac{\partial \bar{v}}{\partial y}\right)^2 + \frac{1}{4}\left(\overline{\frac{\partial u_i}{\partial z}}\right)^2 + \frac{1}{4}\left(\overline{\frac{\partial v_i}{\partial z}}\right)^2 \tag{A5}$$

The above equation can be expressed in terms of $\sigma^2$, the purely horizontal components of which are given as

$$\sigma_{xx}^2 + \sigma_{yy}^2 + \sigma_{xy}^2 + \sigma_{xx}\sigma_{yy} = \left(\frac{2\mu}{\bar{A}^{1/n}}\right)^2\left[\left(\frac{\partial \bar{u}}{\partial x}\right)^2 + \left(\frac{\partial \bar{v}}{\partial y}\right)^2 + \frac{\partial \bar{u}}{\partial x}\frac{\partial \bar{v}}{\partial y} + \frac{1}{4}\left(\frac{\partial \bar{u}}{\partial x} + \frac{\partial \bar{v}}{\partial y}\right)^2\right] \tag{A6}$$

This expression for $\sigma^2$ is then used in Eq. A1.

**Table A1.** Model symbols. Bars indicate vertical averages.

| Symbol | Description | Value | Unit |
|---|---|---|---|
| $x, y, z$ | Cartesian coordinates, $z$ increasing upwards | - | m |
| $u, v$ | horizontal ice velocities | - | $\frac{\text{m}}{\text{yr}}$ |
| $u_i, v_i$ | internal shearing ice velocities | - | $\frac{\text{m}}{\text{yr}}$ |
| $u_b, v_b$ | basal ice velocities | - | $\frac{\text{m}}{\text{yr}}$ |
| $H$ | ice thickness | - | m |
| $h_s$ | ice surface elevation | - | m |
| $\bar{A}$ | $\int A\frac{dz}{h}$ | - | $\frac{1}{\text{yrPa}^3}$ |
| $\mu$ | effective viscosity | - | Pa s |
| $m_b$ | basal sliding exponent | - | - |

## A2   GSM code structure, time-stepping, and recovery from convergence failure

The top-level main GSM routine (Fig. A1) reads in required inputs, initializes components, and then loops with the asynchronous coupling time-step (dlong). Nested flow charts are as follows. Relevant GSM subroutine or variable names are shown within parentheses and relevant source files are enclosed with square brackets.

To optimize speed, the model computes the ice dynamics time-step delt (of value $1/(2^N)$ years, N a whole number) according to CFL constraints based on previous maximum horizontal ice velocities. This is embedded in a larger (default dlong$= 100$

1175   years) time-step for GIA, surface drainage, and EBM coupling. If there is complete convergence failure in the ice dynamics, delt is halved, and the calculation is restarted from the beginning of the dlong time interval with recovered ice fields.

## Appendix B:  GSM numerical sensitivities

Numerical sensitivities are shown for example base parameter vectors with approximately mean parameter values. Mean (10 member) ensemble based sensitivity plots display even less sensitivity to numerical flags (not shown).

For the example GSM Greenland simulation shown in Fig. B1, the only visibly significant ice volume sensitivity of the displayed GSM numerical flags is for the -DTHETANEWA compiler flag enabling the 2D buttressing correction of Pollard and DeConto (2020).

Grounded ice volume sensitivity to GSM numerical flags for an example 122 kyr Antarctic simulation is only significant during the 100 ka to 80 ka interval (Fig. B2). In this case, all compiler flags have a visibly discernible impact, though for some

(such as the -DNumDamp compiler flag enabling damping of the iterative solution for the ice thickness) this difference is never more than 2%.

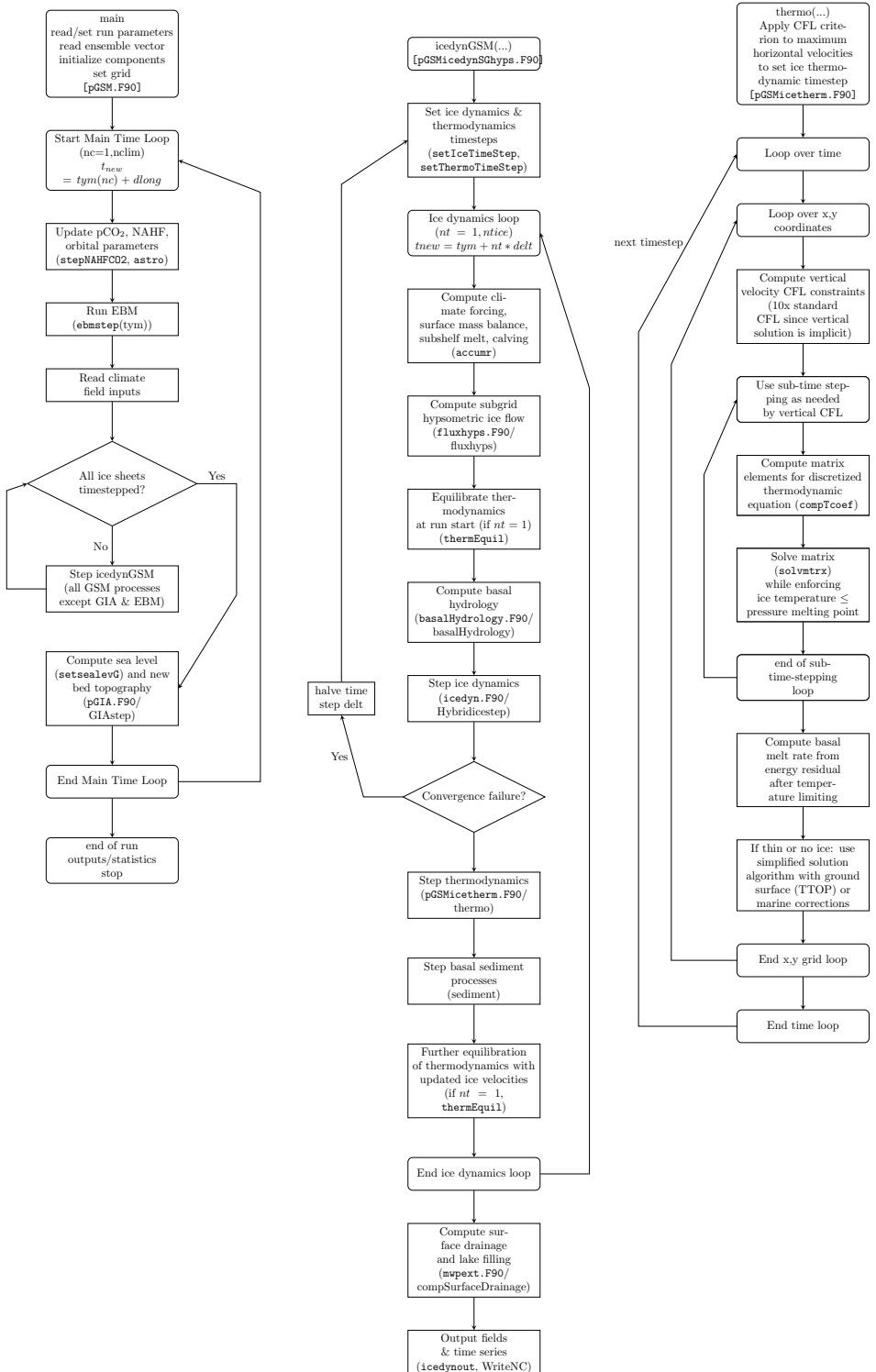

**Figure A1.** GSM flow chart.

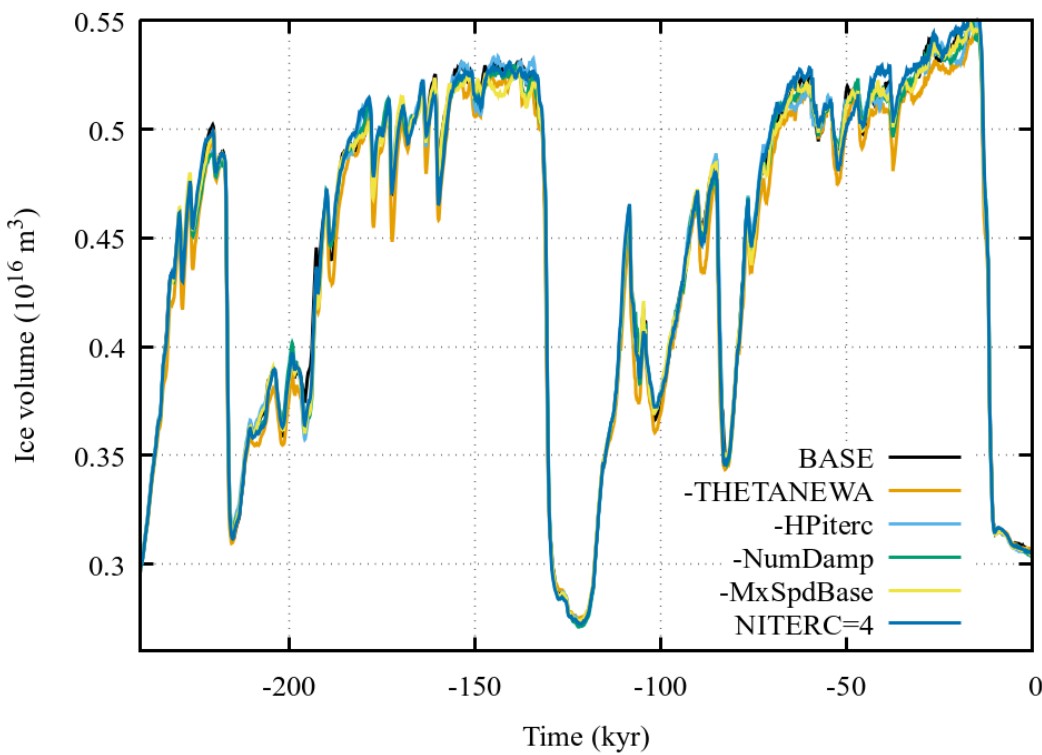

**Figure B1.** Example GRIS ice volume history sensitivity to numerical compiler flags. The simulations use the default $0.5^o$ by $0.25^o$ (longitude, latitude) resolution. Aside from "BASE", simulation key names show the flag that was removed from the recommended default configuration. Removal of MxSpdBase increases the maximum allowed SSA velocity component from 25 km/yr to 30 km/yr. THETANEWA is the revised grounding line flux treatment to address 2D buttressing effects (Pollard and DeConto, 2020). The response to an increase in the maximum number of Picard iterations for the ice thickness solution is keyed by NITERC= 4 (default is 3).

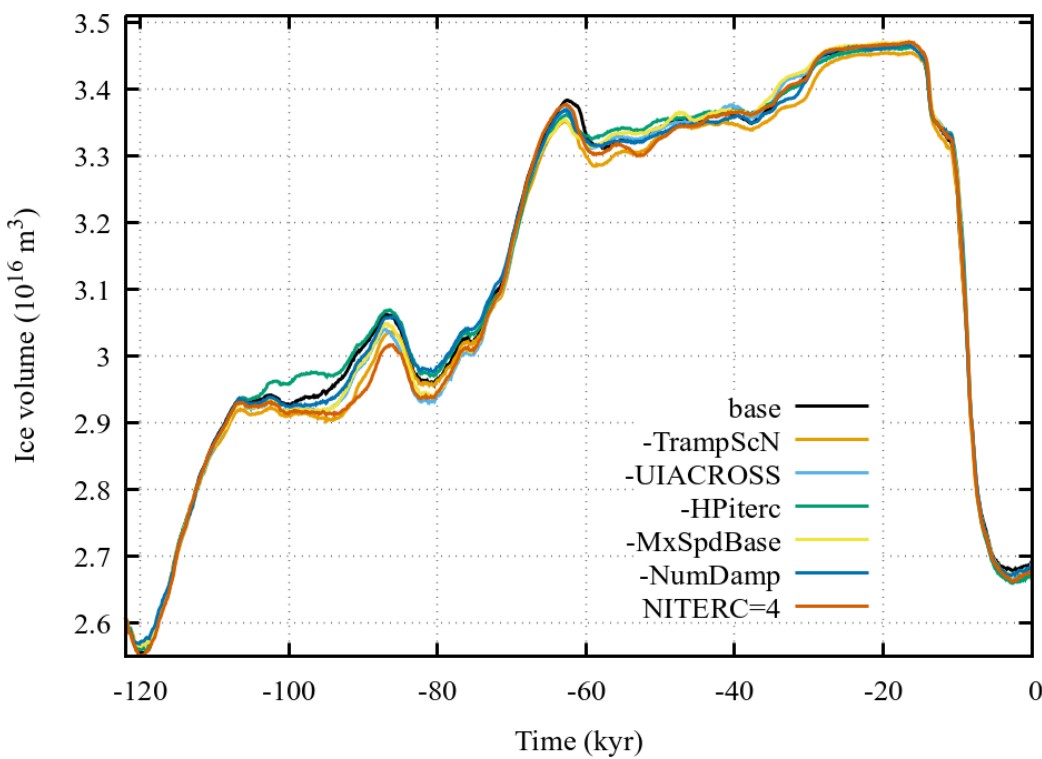

**Figure B2.** Example AIS ice volume history sensitivity to numerical compiler flags as per previous for GRIS (Fig. B1). TrampScN imposes grid resolution dependence on the basal sliding activation temperature ramp ($T_{\mathrm{ramp}}$) in eq. 13 and eq. 12. The removal of this flag results in $T_{\mathrm{ramp}}$ in eq. 12 being nearly twice as large.

*Author contributions.* LT wrote this paper with editorial contributions from BL and KH. DP developed the original SIA/SSA hybrid ice dynamics solver as well as the revised grounding line flux treatment to address 2D buttressing effects. BL coupled the SIA/SSA hybrid ice dynamic core, added the Tsai et al. (2015) grounding line flux scheme as well as dual basal drag law capability, and carried out some of the associated validation and verification. BL also acquired and processed the majority of the Antarctic initial and boundary conditions. KH developed and tested the activation function for sub-temperate basal sliding and tested the various interpolation schemes for basal temperature at the grid cell interface. Unless otherwise specified, LT has developed all the remaining components, optimized the SIA/SSA solver, and continues to maintain the GSM.

*Competing interests.* The authors have no competing interests.

*Acknowledgements.* GSM development has been partly supported by an NSERC Discovery Grant (number RGPIN-2018-06658) and the Canadian Foundation for Innovation.

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
