# Peer review of "The glacial systems model (GSM) Version 24G"

_Geoscientific Model Development, 2024_

## Author Response (AR1)

(NOTE, the response to both reviewers is in this document, so the editor/reader doesn't need to download 2 separate documents)

**1    General comment to both reviewers**

I would like to thank both reviewers for their very thorough reviews, which have helped to make this submission clearer and better motivated (even if I have pushed back in places where I have disagreements on target audience, wording, and such).

**2    General note to both reviewers and editor**

I have spent much effort to address each reviewer comment, which has resulted in a crazy long response. This is a large model, with numerous configurations, and I've had to set what I think are clear boundaries on what should be covered in this submission, to keep the size even half reasonable. Splitting the paper makes no sense to me and I see no logical criteria for such a split. Furthermore, GMD has no stated page length limitation that I could find. As well, APC charges and my limited grant funds means that any split would result in one paper being in a closed access journal.

The initial user manual is set to get a new user started, but it will take ongoing effort to fully document all possible options and configurations. User manual expansion and refinement of model configuration and I/O for user ease will have to be a user community endeavour. If that community doesn't develop, so be it and the manual expansion will be limited. The point of this submission is to reasonably document a well used model with a collective unique set of features for a specific (paleo ice sheet systems) context. I hope that this will lay initial groundwork for a user community, but I don't see that as a necessary condition for acceptance for publication.

**3    General comments: Referee 1**

[**Referee**] Summary

Tarasov and coauthors describe the glacial system(s) model GSM, intended for simulating ice sheet evolution on the glacial-interglacial timescale. It covers a broad range of modules (climate, ice sheet flow, hydrology, solid Earth deformation, sediment transport, ... ) reflecting the complexity of relevant Earth system processes. I appreciate the effort of the authors to document this model and generally support publication of the manuscript while considering the comments below.

General comments

1. The manuscript obviously struggles with the difficulty to document 30+ years of model development in one paper. Since many aspects are interlinked and grown together, it seems difficult to suggest a meaningful subdivision on

the component level. Nevertheless, the idea to split the paper in two or more smaller parts seems obvious and should be considered. In particular in view of my comments below, which if properly addressed, would make the paper even longer. To give a concrete suggestion: some parts of the manuscript already tend to have the characteristics of a reference manual. One could envision to fully focus this manuscript as part 1 on that more technical description, and have a second part that fully explores model capabilities in a more applied sense.

[**Lead author response**] The exploration of "model capabilities in a more applied sense" is more properly done in cited and upcoming articles applying the GSM to specific paleo ice sheets and other relevant contexts. Covering GSM capabilities for every last glacial cycle ice sheet would also result in a monster paper. I also don't see how such a split would significantly reduce the size of this submission since the current submission largely corresponds to the part 1 paper suggested by the referee.

The only aspect that I see as having the "characteristics of a reference manual" is the parenthetical inclusion of relevant compile flags for various options. I favour this approach as it would aid potential users and it takes minimal space in the text. But if clear reasons can be provided for purely relegating this to an added user manual, I'm fine to do so. If there are other aspects that the reviewer has clear reasons for relegating to a user manual or supplement, please elucidate.

I have also searched the GMD guide for authors, and can't find a limit on page count. If there is such a limit, please point me to it.

Another unfortunate factor is that not all Universities financially support open access publication. I am now in that boat and am sadly having to significantly curtail the number of future submissions to EGU journals, so any split would result in one paper going to a different journal.

2. It is difficult to suggest more content in a paper that is already 55 pages long, but while I find the descriptions both detailed and transparent, several parts of the model description are reduced to mere technical level. Generally, there isn't much opportunity to get a grip on what the model choices imply for actual simulations. I am thinking in particular about the different mass balance processes (2.7) and climate forcing options (2.10). The selection of what receives more detail with some examples and figures seems arbitrary. To give a better idea of the model's capabilities, all processes that are not already documented in separate publications should be fully described and illustrated.

A summary impact of the climate forcing options are provided in the supplementary ensemble parameter sensitivity analyses (since all climate forcings are under ensemble parameter control) and this will be made clear in the revised submission. The orographic precipitation downscaling forcing is fully described with sensitivity tests in the cited Bahadory and Tarasov (2018) paper. The EBM climate model is also fully described and validated in the given citations. The use of glacial indexed interpolation of GCM climate fields is over two decades old (*e.g.,* Marshall et al., 2000). The use of monthly glacial indices is novel, but this is just basic orbital physics, and the submission already clearly states the errors that can ensue from reliance on annual indices as has been the norm

until now. To better document some of the various glacial indices, a time-series comparison has been added to the main text (new figure 6.).

The marine and calving components in 2.7 have cited source studies where they are fully motivated, derived, and tested. The lake calving impact is summarized in fig 3. The surface melt refreezing is largely based on previously published models, and the cited reference (Reijmer et al., 2012) is an intercomparison of these models. The revised text now fully motivates the refreezing model with the addition of:

This parametrization was chosen based on the near best fits after retuning of the Huybrechts and de Wolde (1999) approach in Reijmer et al. (2012) with the added pore trapping condition based on the results shown in this refreezing model comparison. The slight retuning of dFRZ from the value in Reijmer et al. (2012) was necessitated by the use of the more physical $NDY$ factor as opposed to the mean annual temperature used by Huybrechts and de Wolde (1999). Unfrozen meltwater will also be retained in any ice surface grid-cell scale depressions when the surface hydrology solver is active in the GSM.

The key novel aspect in section 2.7 is the positive temperature insolation surface melt scheme. This has a strong physical justification, with the main aspect not derivable from first principles (positive degree insolation as a function of mean monthly insolation, temperature and associated standard deviation) justified by figure 2. To provide some further partial evaluation/validation, I've now added a comparison of computed net surface melt by the GSM against results from high resolution regional climate modelling for present-day Greenland (cf new figure 4).

3. The introduction is rather short (effectively 1 paragraph, before going into model specifics) and doesn't give a good introduction to the science GSM is meant to address and the parts of the Earth system it tries to model. After reading the introduction, the reader should have gotten a basic idea of what GSM tries to model, how the different Earth system components interact and hang together, and what other approaches exist to do so. A schematic flow diagram or similar would be useful to support this part.

Yes the intro is short (though now there is a second paragraph to provide more context about often ignored processes), but that is because the context is clear, ie paleo ice sheet modelling. I do not see the necessity of describing the scientific purpose of paleo ice sheet modelling in this model description submission. To make the design purpose of the GSM even more explicit, I've added the following:

The Glacial Systems Model (GSM) is a numerical model for simulating ice sheets and their interactions with the rest of the Earth system over glacial cycle time-scales.

Furthermore, I would remind the reviewer about the third paragraph sic: "A key and distinguishing GSM design consideration is a focus on uncertainty quantification....". This along with the large number of ensemble parameters, diverse climate forcing, and noise injection together provide a distinguishing set of characteristics of the GSM.

I also do not see it appropriate to contextualize each component process of

the GSM in the intro as to how they compare to other approaches. That is more appropriately partially done in the relevant subsections (otherwise there would be excessive process description repetition). Furthermore, how many ice sheet model description papers to date that are at least partially targeted for paleo contexts justify not including pro-glacial lakes, direct radiative impacts on surface melt, monthly resolved orbital forcing,.., or even critically exam the chosen parameterizations in the context of all available science? These one could argue are larger failures of adequate context provision.

To partly address the reviewers desire for more clarity on how "the parts of the Earth system it tries to model" as well as show "how the different Earth system components interact and hang together" I've added a process diagram to the main text (new figure 1).

4. I miss a better view on the approaches and processes that are celebrated in the introduction to be more complex, complete or otherwise superior to other models (englacial sediment transport, noise insertion). I feel these would require a more thorough analysis and comparison to show their relevance and justify these claims. Otherwise they should probably be toned down.

I'm not clear what more thorough analysis is required unless the referee is expecting justification as to why inclusion of englacial transport needs to be provided. This comes down to a question of who is the targeted reader for this submission. I presume two classes of readers: a) those who are not ice sheet modellers and want just a quick summary of the GSM in more detail than provided in whatever GSM application paper they are reading and b) ice sheet modellers who understand the physics of the glacial system. For the a) class readers, the intro and introductory part of section 2 (Model description) along with the conclusions section should be adequate, especially given Table 1 that lists the relevant components. Anyone interested in subglacial sediment processes should know that englacial transport is a significant component of total sediment transport or at least read the cited paper and preprint (Melanson et al., 2013; Drew and Tarasov, 2024) to get a basic understanding of it. If the referee expects justification for the inclusion of every component, then I would rebut that every other ice sheet model description paper should justify their exclusion of relevant components, which is clearly not done.

If the referee disagrees with my target classes of readers, I'd be interested in whom they think I might be missing and why they warrant attention. If a good case can be presented, I would endeavour to expand to that target as well.

The noise insertion is also clearly motivated in the intro (" for partial quantification of structural uncertainties (i.e. model uncertainties not captured by ensemble parameters)").

5. Large parts of the Conclusions section actually read like a classical discussion section. Consider changing the title to Discussion and Conclusions. Also, the discussion should be extended to give a better view on model advantages, caveats and shortcomings compared to other ice sheet models used in the paleo context (resolution, grid refinement, approximations, ...) or why these are not relevant in the given context.

Resolution is already addressed in the conclusions as is the order of the ice

dynamical approximation. Adaptive grids should also have been included and that is now rectified. I don't see the contents as "classical discussion" but more accurately perhaps as "Summary and Conclusions" which I've now implemented as the section title.

Specific comments

Title. How does this release relate to earlier (probably unpublished) version of a model with the same name, e.g. https://doi.org/10.5194/tc-7-1949-2013? This should be made clear in abstract and introduction.

I don't see it appropriate for the abstract, but it is now briefly spelled out in the introduction:

The GSM has a long history (going back to the purely shallow ice approximation version in Tarasov and Peltier, 1997), with the most significant change being the incorporation of the Pollard et al. (2015) ice dynamical core for inclusion of shallow shelf physics completed in 2017. The current form of the GSM largely matches that used for publications using the GSM from 2020 onwards with a chronology of relevant changes summarized on the supplement.

l2. Say what other components it includes to make it a glacial system model, rather than an ice sheet model?

That is already done starting a few lines farther down.

l2. could remove "glaciological" before "ice sheet model".

nope, given "models" like ICE-5G and PaleoMIST that are not glaciological (though the latter explicitly claims "glaciologically plausible", Gowan et al, Nature Comm. 2021) However, the abstract revisions have now removed this whole phrase.

l29. Text from here already reads like part of the model description (Sec. 2).

As described above, I have expanded the earlier part of the intro. And the rest of the intro is too thin for a model description. It is a summary of key (and often distinguishing) features. I make the assumption that many readers (especially non-modellers primarily interested in the history matching results of the GSM for different paleo ice sheets) will just read the intro to get a brief sense of what the GSM is and how it compares to other ISMs for paleo contexts.

l29. "Glacial Systems Model" like in the title or "Glacial System Model" like in the abstract? good catch for which I've been inconsistent. Will change everything to Glacial System Model.

l30. "... not found as a set in any other ice sheet model"

I think it would be more useful to continue in the spirit of the first paragraph and discuss what this specific context requires as features, before stating how GSM addresses those, and only finally how that is an improvement over other models.

I've added the following sentence:

Other potentially critical processes and feedbacks for glacial cycle contexts that are typically ignored for present-day ice sheet modelling include the following: the evolution of proglacial lakes and their impact on ice sheet mass loss (*e.g.,* Tarasov and Peltier, 2006), the evolution of landfast perennial lake and sea ice into ice shelves and ice tongues (*e.g.,* Bradley and England, 2008), the

evolution of geothermal heat flux and permafrost depth and their impact on basal thermal energy balance (*e.g.,* Tarasov and Peltier, 2004), and the impact of changing insolation (due to orbital forcing) on surface melt (*e.g.,* van de Berg et al., 2011).

l34. replace 'nor' by 'or' or 'and'?

done

l38. "GSM currently having 30 (Patagonia) to 53 (North America) ensemble parameters"

What about the other regions (Antarctica, Eurasia, ...)? Generally, the numbers by themselves are not so meaningful for the uninitiated.

The point was to show the range of parameter vector dimensions, so the above was re-written to make this clearer: having a minimum of 30 (Patagonia) to a maximum of 53 (North America) ensemble parameters for a single paleo ice sheet

l41. "(uniquely to date) also has noise insertion options"

How is that different from https://doi.org/10.5194/gmd-15-8269-2022? Include a reference and explain.

was not aware of article. It is now cited and the statement corrected to :

The GSM also has noise insertion options for partial quantification of structural uncertainties (*i.e.* model uncertainties not captured by ensemble parameters, a feature shared by only one other ice sheet model Verjans et al., 2022).

It is also now cited in the noise insertion subsection:

An example of an auto-regressive Gaussian noise approach applied to a narrower range of variables is provided by Verjans et al. (2022).

Table 1. Include headings for 2.7 and 2.10

done

l56. "SSA/SIA"

Include reference to 2.4 and A1

done

l58. "appropriate" to do what? Remove "appropriate" or explain.

removed and yup, "appropriate" has been used too much in the submitted draft...

l61. "basal drag laws for soft and hard beds"

Include reference to 2.5

done, though wondering if duplicate of table 1 contents makes sense

l67. Why not continue following the order of Table 1?

The subsection are organized based on topical cohesion. Here, the organization is partly based on significance, and the visco-elastic GIA solver is an important (and long-standing) feature.

l68. "geoidal deflection"

I get very few hits for this term on online searches. Is there a more common term for this, E.g. drawn from https://doi.org/10.1007/s10712-019-09525-z?

To make this clearer, I've rewritten to:

with a linear approximation for the deflection of the geoid (with space-time dependence) from a spherically symmetric eustatic mean sea level anomaly

l75. "glacially-indexed GCM snapshots"

Is this treated in 2.10.1?

glacial indices are on 2.10.1 but the GCM snapshots are in the subsection cited at the end of the sentence (2.10.2)

added 2.10.1 ref after "glacially-indexed"

l78. Include description of 2.10.4?

done, but this seems like overkill given table 1

l79. Sec 2.12 is called "Ice margin nudging" in the main text but "mass balance nudging" in Table 1. Make consistent. Include a short description and reference to 2.12 here.

fixed and added

l82. Include short description of 2.16 here.

done

l82. In the end, refer again to Table 1.

done.

l85. Remove one "(".

done

Table 2 caption "Non-climate forcing ensemble parameters"

A bit difficult to parse. Maybe "Ensemble parameters (not related to climate forcing)"

changed to

Ensemble parameters not related to climate forcing

Table 3

Greenland specific

latitudinal ramp width of added Holocene warming "$42. - 40 \times (0.0 - > 1.0)$"

Missing dot after 40?

fixed

l106. Sec 2.3 may be better placed as part of or directly after 2.1?

I disagree. The large ensemble parameter dimension is a core distinguishing feature that I want to emphasize by early placement.

l114. Does the solver translate without adaptation to regular and lat-lon grids?

not sure what you mean by "translate without adaptation". The solver uses a generalized grid, that can be specified (with compile flag settings) to be any of the stated options given in section 2.2. In that section, I have now appended: "with the option of the latter two running concurrently for different ice sheets" to "There are 3 horizontal grid options: regular dx,dy; regular longitude,latitude; and polar stereographic projection, "

l144. "Glenn" $- >$ "Glen", also in l150.

done

l144. Several recent studies have suggested the Glen flow law exponent could be closer to 4. It seems like an obvious candidate to sample in the ensemble design. Is it easy in GSM to change n? Is that planned?

The appropriate rheology for ice sheet flow is not just a recent issue and the exponent will depend on the stress regime and flow history with evidence for a $N = 1.8$ regime as well (cf. eg. Peltier et al., 2000, and references therein). A further complication is that the true rheology is evidently anisotropic as already

partly discussed on line 149. The Glen flow law exponent in the GSM is currently a Fortran parameter (ie constant) that is easy to change, but I've now added compile flag options to facilitate this.

To further address the above, I've appended the following to the subsection:

The GSM uses a default Glen flow law (exponent 3 stress dependence) ice rheology (Glen, 1952; Cuffey and Paterson, 2010). However recent work has favoured an exponent 4 for ice sheet contexts (Fan et al., 2025), though curiously it has chosen to ignore evidence for grain boundary sliding being the rate-limiting process with exponent 1.8 for typical ice sheet stress regimes (Goldsby and Kohlstedt, 2001; Peltier et al., 2000). Furthermore, for temperate ice with $> 0.6\%$ liquid water content, laboratory experiments indicate a linear viscous rheology dominates due to diffusion creep (Schohn et al., 2025). To address this uncertainty, the GSM flow law exponent is a free parameter with a compile flag option (-DPOWiceEns) to convert it to an ensemble parameter. However, this will entail user coding of an appropriate temperature dependent flow coefficient for the chosen exponent.

l165. "for a detail examination" $- >$ "for a detailed examination of the"

that larger phrase was removed as it's better dealt with in the appropriate subsection: 2.5.2 Basal sliding activation

l166. Consider using i), ii) , iii) after "accounts for:"

done

l170. "With the -DNeffDRAG compile flag"

Can this be motivated (physically)? What is the aim of this change?

Sentence before the formula has been replaced with:

The -DNeffDRAG compile flag combined with any form of basal hydrology imposes basal effective pressure dependence on the basal drag. When activated, the Weertman basal sliding coefficient is multiplied by the following to give a regularized form of traditional basal effective pressure dependence (*c.f. e.g.,* page 240 of Cuffey and Paterson, 2010):

l175 "exponent mb = 4"

Clarify is this is only for the Greenland domain or generally the case.

the following was added

"Given the high statistical confidence in the results of the above inversion for the 4 (of 8 total) catchments with mostly strong (hard) beds, we tentatively assume this value is appropriate for all other paleo ice sheets. If there was evidence or judgment to the contrary, turning this exponent into an ensemble parameter would be trivial."

l183. "contemporaneous sea level"

In which vertical reference frame is the model operating?

That question is not relevant to this section (basal drag). If your question is in regards to GIA, current GIA models (including that of the GSM) are relative to Earth's centre of mass. GIA effects are applied as displacements to the present-day bed topography and associated Geoid topographic reference frame as given by the choice of topographic input (BedMachine, GEBCO, or what not). To make this clear, I've appended "(relative to the Earth's center of mass)" after

the declaration of the bedrock radial displacement function $R(\theta, \psi, t)$ in the first paragraph of the GIA subsection.

l193. "fractional soft bed cover of the grid cell"

Where does this information come from? Is this from a dataset (which one) or dynamically computed based on sediment transport in the model?

I've prepended the following:

The GSM requires a specification of the fractional soft bed cover for each grid cell, either as a constant input (Table 6) or dynamically determined (section 2.14).

l197. "in partial accord with a numerically self-consistent treatment for the setting of cell interface diffusion coefficients"

Not sure I can follow this. Have "cell interface diffusion coefficients" been introduced before?

It is beyond the bounds of this submission to explain computational fluid dynamics (the cited text devotes over 3 pages on this specific issue). So I'm not sure what to do here beyond the already present citation of an appropriate text on the subject. Perhaps, the context wasn't clear. To hopefully make this easier to follow, I've partially rewritten this as follows:

This is taken as the square root of the product of adjacent sediment cover fractions in partial accord with a self-consistent treatment for setting diffusion coefficients in a discretized linear diffusion process (the square root operation was chosen to provide an intermediate between an arithmetic mean and the appropriate harmonic mean, cf Patankar, 1980).

l202. "input sediment fraction"

Do we know how the sediment fraction enters the model?

cf relevant response above.

l209. "their data input requirements are unlikely to be met"

Not clear to me what this is referring to.

have now inserted "(metre scale bed topography)" after "input requirements"

l218. "A last motivation for this design choice"

While I agree with the arguments for paleo contexts, I wouldn't call that not-implementation a 'design choice'. Maybe just "choice".

sentence was changed to more appropriate:

There is a need for the development of robust basal drag parameterizations that can be applied to all paleo ice sheets, be it for regions that are presently subglacial, marine, or subaerial.

l221. "subgrid pinning points under the ice shelf that aren't presently active"

or not resolved.

If they are active, then they should be resolved from surface and remote observations otherwise their activity is not significant.

l241. "grid cell resolution $(\Delta xy)$"

How is that defined for a lat-lon grid?

depending on compile flag choice, can either use a constant delta latitude value or $\cos(\text{latitude}) * R_{earth} * \delta_{longitude} * \pi/180$ for the longitudinal direction.

This is too minor a detail for the main text but will be included at some point in the user manual.

l250. insert "is" before "focused on surge cycling"

I disagree on the grammar/semantics

l263. "GSM ice and permafrost resolving bed thermodynamics"

Reads like this section is on bed thermodynamics. Maybe "ice thermodynamics and permafrost resolving bed thermodynamics"

Could remove "GSM", in line with other headings.

changed to, albeit awkward:

"Ice thermodynamics and permafrost resolving bed thermodynamics"

l267. "Heat source terms include full SSA and SIA contributions to deformation work (Qd) and the boundary heat flux"

Also describe all the other components of the equation.

I've added what $\rho_i, c_i, and, k_i$ are, but otherwise assume that T, V, and z are self explanatory and expect anyone reading this part to recognize advective, diffusive, and heat capacity terms of the energy conservation equation, an assumption made in eg Wilkens et al. (2015).

l279. "Unlike many ice sheet models"

Not sure this is still the case. Energy conservation has been addressed in many ice sheet models these days.

Was previously the case, but I'm not up to date on all the current models, so have changed this to

"Unlike in at least most older generation ice sheet models"

l307. "orbital changes in short-wave forcing"

Could refer to https://doi.org/10.5194/tc-8-1419-2014 for another approach to this concern.

Done.

l311. "Observationally, fitted PDD melt coefficients vary over a wide range"

I was confused by "fitted". PDD factors can be experimentally determined by measuring ice and snow melt and temperature at the same place. Where does the fitting come in?

How do you get from a set of ice/snow melt and temperature measurements to a PDD melt coefficient? This would typically require regression fitting. Suggestion for alternate wording would be considered, but "experimentally determined" is vague and misleading.

Also, wide range during the day, between different locations, or generally?

generally, now clarified:

Observationally, fitted PDD melt coefficients vary over a wide range (both spatially and seasonally).

added (both spatially and seasonally, *e.g.,* Braithwaite, 1995; Hock, 2003).

l316. "only contributes to surface melt if the surface temperature is at $0^o$ C"

Do you account for sublimation separately?

Nope, this is part of the climate forcing that I forgot to describe. I've now added the following:

Evaporation, sublimation, and deposition are three further climate forcing components that affect surface mass-balance. As these processes are highly dependent on near surface vapour pressure gradients which in turn depend on boundary layer turbulent mixing, they are unlikely to have any simple large scale orographic dependencies as imposed on precipitation in the GSM. Lacking a better alternative, the GSM simply applies the same glacial index geometric interpolation between present-day and LGM fields as used for precipitation in eq. 45 to the net of deposition less evaporation and sublimation. The resultant field is then added to precipitation as the final step in determining monthly mean precipitation within the GSM (*i.e.* after all other adjustments described in this and the next section).

l324. "PDDs are computed for paleo modelling contexts based on a probabilistic distribution around mean monthly temperatures"

Calls for some references :

I've added a couple citations, but this is further discussed farther on.

(e.g., Tarasov and Peltier, 1997a; Wake and Marshall, 2015)

Figure 2 caption. "set to it's nominally regressed value" $->$ "set to its nominally regressed value"

done

l345. "it has been common for paleo ice sheet models to determine PDD" $->$ PDDs

Calls for some references.

added: (e.g., Tarasov and Peltier, 1997a, Albrecht et al., 2020b)

l359. "in the above supice equation"

Refer to equation number (20) instead.

done

l361. "RCMs" is not defined

fixed

l361. "This parameterization deviates from previous"

Add "studies", give references.

this part was rewritten to more accurate reflect progeny/motivation:

This parametrization was chosen based on the near best fits after retuning of the Huybrechts and de Wolde (1999) approach in Reijmer et al. (2012) with the added pore trapping condition based on the results shown in this refreezing model comparison. The slight retuning of dFRZ from the value in Reijmer et al. (2012) was necessitated by the use of the more physical N DY factor as opposed to the mean annual temperature used by Huybrechts and de Wolde (1999).

l364. not clear what "ibid" points to.

Ibid, as per its standard usage, always refers to the previous citation

l418. "submarine face melt"

Maybe "submarine calving face melt"?

bit awkward but done

l477. insert "the" before "grounding line"

done

l488. "bergy bits"

Do you mean ice mélange or "sikkussaq"/"sikkusak"?

changed to ice melange (sikkussaq seems to be restricted to sea ice)

l500. "self-consistent DEM"

Is the DEM updated based on GIA and blocking of ice? If so, mention/explain here.

yup, was confusing, cleaned up to :

For present-day ice free surface topography, the solver uses a modified version of the USGS EROS HYDRO1k hydrologically self-consistent DEM (USGS, 2004). The drainage preserving upscaling of the DEM includes some by hand corrections to capture the controlling sill elevation for the southern drainage of the central LIS (*e.g.,* pro-glacial lake Agassiz). This topography is then dynamically evolved for ice cover and GIA.

l520. "The change is ice thickness" replace "is" by "in"

done

l521. "surface runoff discharge calculation"

Is mass conservation imposed on the global level/per ice sheet domain? How does that work?

The surface drainage solver conserves the meltwater and calving discharge (ie mass) by design. It works as per the description (with "mass-conserving" and "diagnostically" now added:

"The mass-conserving solver simply diagnostically routes water downslope, filling depressions (lakes), until an ocean depth of 200 m or until no water is left."

The solver is described in the cited paper.

l522. "This ice remains subject to all the other mass-balance processes in the GSM."

Not clear to me what that means.

inserted "lake and sea" before ice in the above

l526. and elsewhere. What is the logic of capitalizing or not section titles? Make consistent.

fixed

l530. "August-February differences range up to 25%"

Is this a temperature difference? 25%/100% of what?

relative to LGM glacial index. But to simplify, I've changed this now to the actual difference in glacial index values (as well as corrected the EBM range):

'For example August − February differences range up to 0.30 over the last two glacial cycles for the EBM derived monthly glacial index (nominal 0:1 range for 0 ka to LGM). For a more advanced coupled ice-climate model over the last two glacial cycles, the glacial index differences can exceed 1.0 (Geng et al, manuscript in preparation).'

l533. "Ie is the mean monthly EBM temperature anomaly"

Could be useful to have a table of the different Is for a better overview, e.g. [I, purpose, equation, range]

I've modified the intro of the section to clarify the range:

"The GSM has various time and/or state evolving indices for driving components of the climate forcing. The indices cover a range of 0 to 1 with 0 representing the 0 ka (nominally 2000 CE) state and 1 the LGM state."

The suggested table would be awkward given the stated optional forms. I have made a point of clearly labelling and repeating the label of each index as you have suggested. I have also added a plotted timeseries comparison of glacial indices.

l584. "NCAR CSM general circulation climate model"

Was called "CCM" at that time. See referenced article.

fixed

l643. Consider introducing a subheading for temperature and precipitation (I think bold without numbering is the level 4 heading)

The problem is that precipitation is interwoven with the temperature description in a few places (EOFs, Keewatin dome index) so this would necessitate some repetition to make a full split. So I don't see such a split working.

l650. "as followings"

"as follows"

fixed

Eq 45. Resolve double subscript on T2m0.

done

l656. "Computed precipitation is then subject to the factor"

I presume this means moisture is not conserved? Could be good to mention.

water mass is conserved (ie precip $->$ ice $->$ runoff), but it's meaningless to talk about conserved precipitation given that it is a forcing, unless the GSM is coupled to a climate model that resolved precipitation for which the GSM configuration will be mass conserving except when the coupling uses ocean model acceleration.

l659. "climate index (Ic)", "dome elevation index Id"

I think using the longer form, like here, stating what the indices mean should be the standard throughout the paper.

done except for $I_N$ which doesn't have an obvious name except for maybe AMOC atmospheric heat flux parametrization index which would be awkward and not add anything given the already clear contexts where it's used.

l664. 2.10.3 would make more sense for me to be a part of the section on precipitation above (at level 5), rather then a level 3.

precip is in the previous 2.10.2 surface climate forcing subsection.

I judge the orographic downscaling is a distinct and somewhat novel enough process to warrant its own subsection. And I'd rather not have further subsubsection level hierarchy.

l665. "Paleo ice sheet modellers have traditionally relied"

done:

(e.g., Tarasov and Peltier, 1997a, Albrecht et al., 2020b)

l705. "the Eemian high-stand was inadequate"

add "sea level" to clarify what high-stand this is.

What magnitude/range are you assuming and trying to match? Reference.

The text has been revised as:

After an initial set of history matching waves (Tarasov and Goldstein, 2023; Lecavalier and Tarasov, 2025), it was found that the simulated Antarctic contribution to the Eemian sea level high-stand (generally less than 2 m eustatic

equivalent) was inadequate to cover the inferred possible range (*e.g.,* Kopp et al., 2009) even after accounting for potential contributions from Greenland (*e.g.,* Tarasov and Peltier, 2003).

l715. Does this feature really need a separate subsection?

This near unique feature required a whole paper to describe and test. I don't see where it can be subsumed since it relates to both SMB and ice flow.

l719. "is that module" $->$ "is that the module"

fixed to $->$ "is that this module"

l730. Clarify in how far this means mass is not conserved.

As in the response above, mass in conserved in the model, but precipitation is a forcing (unless coupled with a precipitation resolving climate model), so it is meaningless to talk about mass conservation of the precipitation forcing.

l755. "activated basal hydrology component"

Add reference to sec 2.13.

done, but seems a bit of overkill since it is the previous section on the same page..

l770. "as is typical for paleo ice sheet modelling"

Calls for some references.

done:

(as is typical for paleo ice sheet modelling, e.g., Tarasov and Peltier, 2002; Lecavalier et al., 2014)

l787. - l793. "To improve generalizability, ..."

This text gets very specific and is difficult to understand without further instructions. Suggest to keep the description more general or make an example and get much more detailed.

I think making it more general would make it even harder to understand. And without further guidance, I'm not clear what is hard to understand. If it's the reference to history matching, then I would argue that anyone interested in trying to constrain glacial cycle ice sheet evolution better understand what history matching is (cf the cited ref..) and a detailed explanation thereof is well beyond the bounds of the current submission.

l794. "For ice sheets with extensive present-day ice cover"

What ice sheets do you have in mind? Are there more than two (GrIS and AIS)

I could have just stated GRIS and AIS, but the point is that they have present-day ice cover, thus the choice of wording. Adding GRIS and AIS seems extraneous.

l794. "the sensitivity of the correction to discrepancies in simulated 0 ka ice thickness (compared to that observed) are too strong for such a correction approach"

What instead then? Is it not needed to correct those?

I've added the following text:

For ice sheets with extensive present-day ice cover, the sensitivity of the correction to differences in simulated 0 ka ice thickness are too strong for the use of common correction fields ($h_{\mathrm{boc}}^{n}$) for a given earth rheology. For this case, the correction fields have to be extracted for each GSM parameter vector

(as is also done in van Calcar et al., 2023). For the case of the GRIS, initial exploratory experiments with a 10 member NROY parameter vector set indicate one correction iteration can be marginally adequate, with *e.g.,* 0 ka RMSE differences between gridded observed and modelled (after a 240 kyr transient simuation) subglacial bed topography of less than 16 m. Two iterations brought this down to < 3 m for the majority of simulations. However some simulations diverged when 3 or more iterations were applied. Damping of the corrections fields (up to 25%) was tested but did no improve convergence.

l796. "Geoidal deflection within the GSM ice sheet grid is computed using a linear approximation."

Add a short introduction for the uninitiated. What is the purpose of this calculation?

I've reworded the first sentence to make it more self-explanatory:

"The GSM computes spatial variations in the Geoid in response to changing (ice/water/earth) mass distribution with a linear approximation. The model modifies the mean volumetric (eustatic) sea level change with a spatially varying Geoidal deflection computed as linear contributions from each of the 4 major ice sheets."

l811. " a future"

"the future" or "a future version".

done

l815. "inter-model comparison experiments"

Typically "model inter-comparison experiments"

done

l832. "This approximation will become more inaccurate"

While this statement gives some importance to this choice, it could be mentioned here that the choice of an initial temperature profile is arbitrary and mostly a question of convenience, to shorten the required equilibration time. In other words, this choice shouldn't really have a big influence on the final spun up temperature if the relaxation is done appropriately. Is this not illustrated in Fig 6?

While I would cautiously agree that a fully converged initialization will likely have little sensitivity (beyond time required for convergence) to some moderate range of initial temperature profiles, given that the coupled system is non-linear (especially if basal hydrology is a control on basal drag *e.g.,* Hank and Tarasov, 2024), I can't rule out that the ice sheet dynamical system has multiple attractors when subject to larger variations in initial temperature. Fig 6 only provides example validation of the chosen initialization. It does not document sensitivity to the choice of initial temperature profile. The choice of temperature profile is clearly motivated in the paper, so I'll leave it at that.

l841. "For a set of not-ruled-out-yet"

This concept should be introduced early in the manuscript. E.g. in the introduction.

I disagree. The term is only used twice in the text, it is introduced on first use with a reference that fully explains the term (and even a summary description of history matching is beyond the bounds of this submission). I don't see how

it would fit in the introduction. However, to aid the reader, I've prepended "history-matched" as that term is becoming better known in the community.

l846. "is the difference" − > "in the difference"

done

Table 7. "key GSM fields"

Maybe "Key prognostic GSM variables"?

done

l875. "drainage solver verification again present-day drainage" − > "drainage solver verification against present-day drainage"

done

l898. "process noise injection for internal discrepancy assessment is also to date unique"

See comment l41.

sentence changed to

"The GSM's design focus on addressing structural uncertainties for glacial cycle modelling context is also reflected in the inclusion of process noise injection for internal discrepancy assessment."

l961. I find the verbatim font without any structuring elements is difficult to read. Could this be presented e.g. as a bulleted list with different symbols in addition to the indentation?

This has been converted to a flow chart.

Figure B1. Difficult to make out differences. Consider zooming in or producing an inset, e.g. around the LGM, Eemian and/or present day.

The lack of significant differences (except arguably for the orange line THETANEW) is the point, ie that these numerical flags (chosen for run speed optimization) have little impact on at least the ice volume trajectory. Therefore, I find it important to show the whole time series.

l1035. Missing some dois throughout the reference list. Here e.g. https://doi.org/10.1007/s40641-017-0071-0

I see no requirement in the GMD author guidance for inclusion of DOIs (but correct me if wrong). With eg google scholar, their absence is little hassle for the reader and I hardly use them given proxy server limitations for University access to Journals.

Supplement

All figures in the supplement.

- increase axes label sizes and y-tick label sizes

- parameter number (x-ticks) can be removed.

- increase x-axis range to fully include rightmost parameter

- move parameter names up so they are not overlapping with the x-axis or outside of the figure to also avoid overlap of symbols and parameter names

- replace y-axis label "metric" by actual metric name and units.

all of above has been done.

Captions. Cryptical what the reference parameter vector is (e.g. an1600). Explain?

on first use, for clarity, this has now been expanded to : "Reference parameter vector identification code is an1600". If one has ten's of thousands of simulations archived for each ice sheet, a numerical ID is critical for traceability..

Figure 1. "for the parameter range in Tables2 and 3." ... "in the main manuscript".

done

**4    General comments: Referee 2**

[**Referee**] In the manuscript "The glacial systems model (GSM) version 24G", L. Tarasov and co-authors describe the glacial systems model, a software that simulates the evolution of ice sheets on paleoclimate timescales and that has been use for over two decades. The manuscript is very detailed, explaining both the physical processes included in the model, as well as their numerical implementation, and listed the different options available, including default configurations and parameter values.

The manuscript is usually well written and clear and the tables are useful, however I found the figures not very informative: it is unclear what these specific figures have been included as they are not very representative of the overall model, and additional the paper is missing some overall figures describing all the processes captured as well as typical runs including all the ice sheets and processes described (for example showing the ice extent for the different ice sheets at glacial maximum and interglacial times).

[**Lead author response**] The figures are primarily there to show example GSM sensitivities to various processes and numerical choices, which is what I expect to see in a model description paper. I have added an example plot in the supplement showing the 0 ka ice thickness and grounding line error between a 205 kyr GSM simulation and Bed Machine V2. But this has limited meaning without a detailed explanation of how the parameter vector was determined. However a full explanation of the choice of parameter vector and detailed configuration is well beyond this manuscript as I would then need to fully explain and document the history matching, constraint data, description of error model that relates constraint data to model output, fit to constraint data, and analysis that went into the choice of each parameter vector. And what does a "typical run" mean when you have order 40 ensemble parameters? Meaningful assessment of GSM results requires ensemble analysis. Each paleo ice sheet history matching is a major endeavour, requiring at least one paper per ice sheet. These are the appropriate places to assess model performance for a given paleo ice sheet. Three papers related to the AIS history matching are already published or under open review (Lecavalier et al., 2023; Lecavalier and Tarasov, 2025, 2024) and there is now explicit referencing of the second paper in the model tests subsection:

For partial verification of the whole ice sheet system, Lecavalier and Tarasov (2025) provides a recent application of the GSM to history matching of the last glacial cycle Antarctic ice sheet that includes comparisons to present-day

observations. For convenience, a comparison of 0 ka ice thickness from a sample 205 kyr transient simulation against observations (as inferred BedMachine V2 Morlighem et al., 2019) is presented in section 2 of the supplement.

The GRIS history matching paper will be submitted this fall. Given all of the components, dimension of the ensemble parameter space, and all the possible ice sheets that can be modelled, I don't know what kind of sensivity tests would satisfy being "representative of the overall model".

I have added a GSM component/process diagram to the main text and a flow chart diagram to the appendix (too cluttered for main text). This along with the process/component list of Table 1 should provide appropriate encapsulation of the GSM components.

My main concerns are related to the need to be more clear about all the processes included, since people reading the manuscript are not necessarily familiar with all the physics, processes and numerical implementation described. For example, some terms are not common and should be better described (e.g., discretization of basal ice grid cell, etc.). I also noticed that many sections have very few references if any, and previous both performed both using this model as well as other models should be better referenced. So many of my comments below refer to the need to better and justify choices made in the code.

Table 1 lists the key process components and this is now complemented by the Figure 1 (GSM key components and process linkages indicating key variables passed). For those not interested in using the GSM or appropriating aspects of it, I suspect most of the material outside of the introduction and conclusions are irrelevant. The text beyond these parts is targeted to ice sheet modellers. However, in response to all the detailed comments provided by the reviewer, the text has been significantly revised to address sources of confusion and improve motivation while trying to keep this submission from ballooning. I would also remind the review of the context and the statement from end of section 2.3:

As such, the approach has been to combine physical reasoning, parametric forms from the literature, and broaden degrees of freedom across various components to albeit incompletely convert process uncertainties into ensemble parameter uncertainties.

A model of nearly all key interactions between ice sheets and the surrounding physical system would be a challenge to fully motivate in even two or three papers. Furthermore, I'm aware of no paleo ice sheet model description paper that motivates or justifies the choices made to ignore permafrost, orographic forcing of precipitation, the large uncertainties in climate (which cannot be captured by the at best four ensemble parameters that most paleo modellers choose to use), impact of pro-glacial lakes, present-day isostatic disequilibrium,...

The revised draft has 150 references spanning 9 and a half pages. Citations have been added where requested by reviewers as appropriate.

The discussion and conclusions are a little overlapping, and several aspects mentioned in the conclusions could be moved to the discussion. I also think that additional discussion about how this model compares to other existing models would be beneficial to highlight the similarities and differences with other ice models.

There is no "discussion" section, so I'm confused here. And in response to the other reviewer comments, the conclusion section has been renamed "Summary and conclusions" to better reflect intent and content.

I'm also unclear what the reviewer would want moved out of this section. It briefly summarizes the unique features of the GSM, recommends other models for different contexts, summarizes results of presented sensitivity tests for 3 key features that are " typically ignored for glacial cycle contexts", outstanding key sources of uncertainty, and future GSM upgrade priorities. If I was reading a modelling paper, this is what I would want to see in the concluding section.

Finally, for the code and data availability, the configurations shown in this paper should be accessible and easily reproduced, on top of providing access to the GSM code.

The revised tarball asset will have a user's manual with instructions and parameter files for replicating configurations shown in this submission.

**5   Specific comments:**

l.3: "evolved from" − > "evolved over"

"evolved from" is more accurate

l.3: "constrained last glacial" − > "constrained the last glacial"

the text is correct as was stated:

"effort to constrain last glacial"

l.4: "each major ice sheet": these ice sheets should be listed

Done:

(North American, Greenlandic, Icelandic, Eurasian, Patagonian, and Antarctic, and soon Tibetan)

l.12: "traditional glacial indexed interpolation": what does this mean?

Scherrenberg et al. (2023) uses the term in their abstract without explanation. However, to make this more accessible, I've rewritten to:

"variants of traditional input time series weighted interpolation (aka "glacial indexing") of fields from General Circulation Model (GCM) simulations"

I've also added the explicit equation for glacial indexed interpolation for the near surface air temperature forcing in the climate forcing subsection.

l.23: "eg Drew . . . " − > e.g., Drew . . .  (same in all the manuscript)

fixed

l.28: add examples

Without more precise guidance on examples of exactly what, I can't respond. The identified statement should be quite obvious. Otherwise, I would expect at least one published paper by now of large (say at least 2000 member) glacial cycle ensembles of Bicycles or ElmerIce run at high grid resolution (say refined to less than 1 km near ice margins, ice streams, and grounding lines) for Greenland or Antarctica (such simulations have been done for present-day to near future).

l.34: "nor subglacial" − > "and subglacial"

done

l.70: "The resolving of pro-glacial" − > "Resolving pro-glacial"

I disagree on this phrase structure

l.74: "the capture of marine ice" $->$ "to capture marine ice"

disagree

l.85: It is not clear how this is addressed: more uncertain parameters make it very difficult to constrain as observations are sparse/lacking for many aspects

Actually, each major last glacial cycle ice sheet has more than enough available paleo observations that even with all the parameters listed, an appropriate fit to available data constraints can't be achieved with even a 2000 member Latin Hypercube (cf past and upcoming papers from my group and me). But even if this were not the case, you either fully address system uncertainties, or choose to ignore them. As uncertainties are effectively the specification of the relation of your model to the physical system, if you choose to ignore them, then you are choosing to make your model unrelatable to the physical system of interest. For more on the issues, cf the cited Tarasov and Goldstein (2023) history matching discussion paper as well as an upcoming submission on the detailed approximate history matching methodology I use for paleo ice sheet modelling/inference.

l.91-94: What is the objective here? Would it be more clear to use physical values?

As is typical in any adequately complex model of the earth system, there are many model parameters that have no true physical value nor a physically interpretable one. Eg, what is the "physical" meaning of the EOF weightings, or the bed roughness parameters and what would be their physical scaling? More critically, when dealing with order 40 to 50 ensemble parameters, having those without clear physical scaling on a 0 to 1 range is just plain easier. Anyway, any user is free to rescale ensemble parameters to whatever choice they desire...

Table 2 caption: "depending on the whole parameter vector value" $->$ "combination"

More precise/accurate as stated.

Table 2: why use a range of 0 to 2 for the basal drag subgrid parameters? I thought these parameters represented a range from fully floating to fully grounded (so 0 to 1)?

"Floating to grounded" makes no sense as floating/grounded is determined dynamically. Some of the parameters had ranges evolve for various reasons. Detailing such history will only clutter the description and is in good part irrelevant as any user is free to choose their own scalings. For this case, originally the range was from 0 to 1, but the relevant equation (current eq 10) has no clear scale and reconsideration over the years lead me to expand the range. The other option would have to to rescale eq 10, but that breaks continuity with past work.

Table 3: it would be better to include units in the table for as many parameters as possible

I've rechecked Tables 1 and 2, and have added the few missing units. The rest are non-dimensional.

Table 4: Why use a different number of layers for the dynamics and thermodynamics, and how does it work in the code? How are the values transferred

from one set of layers to the others and what is the impact?

High horizontal velocity gradients are concentrated near the bed with monotonic dependence on depth, and as such the vertical grid for velocities needs most of its layers near the bed. In contrast the vertical temperature profile is non-monotonic and high variable, with significant changes in the vertical temperature gradient possible at any relative depth (*e.g.,* Cuffey and Paterson, 2010). Values are transferred by linear interpolation. As for impact, I'm not clear for what context: interpolation method or just having different number of layers. The choices were made based on past numerical experiments, as is the case for the myriad of numerical choices any ice sheet model developer will have to make and for which the large majority will be undocumented (otherwise it is unlikely that any ice sheet model description paper would ever get published).

The relevant grid description in the submission has been expanded to capture the key points of the above:

As significant changes in the vertical temperature gradient are possible at any relative depth (*e.g.,* Cuffey and Paterson, 2010), the vertical temperature grid is a standard sigma grid for ice temperature. It has default 65 layers (GSM parameter NCZ) with an effectively vertically split basal cell to more accurately compute basal melt. As the vertical profile is monotonic with changes in the vertical gradient concentrated near the bed, the GSM uses an irregularly spaced sigma grid for velocities with high resolution near the bed (with default NLEV= 12 layers). Velocities and temperatures are transferred to each other's grid by linear interpolation. As is fairly standard, the ice sheet model uses an Arakawa C-grid, with fluxes and velocities computed on grid cell interfaces.

l.96: what are implicit none?

I've appended a parenthetical explanation:

"The GSM is mostly coded following Fortran 90 conventions and formatting, including the use of modules and implicit none (the latter requires each variable to be explicit declared)"

l.102: "vertical layers": are the layers fully vertical or terrain following?

as stated (sigma grids are terrain following):

"irregularly spaced sigma grid"

Section 2.4: lots of references missing in this part (SIA, SSA, etc.)

if you are referring to expansion of acronyms, SIA/SSA are expanded on first use (line 56), if you mean that a reference is needed to explain what SIA and SSA are and how they are derived, I disagree. This is not done for *e.g.,* Robinson et al. (2020) and would be akin to requiring every GCM description paper to reference some derivation of the primitive equations.

l.116: What is the appropriate boundary condition here? Explain it clearly

that line is changed to:

the appropriate stress balance boundary condition for a floating ice margin (cf. eg. Cuffey and Paterson, 2010; Winkelmann et al., 2011).

More explanation has also been added to the A1 appendix:

The SSA stress balance boundary condition at a floating ice margin is implemented effectively as in Winkelmann et al. (2011) with appropriate densities for ocean or lake. For tidewater ice margins, the subgrid ice margin is assumed to

be at flotation, and the grounding line flux condition (Schoof, 2007; Tsai et al., 2015) provides the effective boundary condition.

l.122: Does it mean that it fails even when using the CFL condition? In this case what is the reason for that and what are the solution?

CFL is not a sufficient condition for the numerical stability of a non-linear set of PDEs. Furthermore, CFL can only be diagnosed based on the last instance of calculated ice velocities and therefore the computed CFL condition result might not match that for the current numerical iteration. As I would class this as basic understanding for anyone doing numerical flow modelling, I do not see it warranting any additional discussion in the text.

As per the solution, that is already given in the text (repeat of last time step(s) with $dt/2$).

Eq.2: What is QL

previous sentence expanded to :

for preconditioner matrix that can be split into left and right components $Q = Q_L Q_R$

l.144: "floating ice grounding onto ice free land": this sentence is very confusing, try to explain that more clearly

reworded:

correction to handle a floating ice margin that subsequently becomes grounded on what was previously ice free terrestrial land (an atypical situation, that was found to occur in Northern Baffin Bay for some GSM glacial cycle simulations).

l.153-158: how were all these values chosen and calibrated? What does the uncertainty look like?

rewritten to clarify how values were set (skipping the basic algebra):

Invoking Occam's razor to give a functional form of $E_{shelf} = A/E_{SIA} + B$, along with the requirement that the SSA enhancement $E_{shelf}(E_{SIA} = 1) = 1$ and $E_{shelf}(E_{SIA} = 5.6) = 0.6$ from Ma et al. (2010), the SSA enhancement factor for ice shelves is therefore

$$E_{shelf} = 0.48696/E_f + 0.51304 \tag{1}$$

As explicitly stated, this only "partially accounts for anisotropic effects from fabric development in polar ice" however the reviewer raises a fair point about associated uncertainties. As such, the following is now appended:

The above ignores uncertainties in the relation between SIA and SSA flow enhancements, and therefore warrants future investigation. However for now we judge that such uncertainties are swamped by other relevant sources, especially those due to climate forcing controlling surface and basal (for ice shelves) mass-balance.

l.162: add references for the sliding laws

done :

For the Weertman case, the effective basal sliding law (for both hard or soft beds) is given by (*e.g.,* Weertman, 1957; Cuffey and Paterson, 2010):

l.165: "for basal drag" $->$ "with basal drag"

done

l.165: you should precise the form of this temperature ramp since this is a description paper

this is done in section 2.5.2, this subsection is now referenced here.

l.173: "scaling coefficients": how are they chosen?

As stated, $C_{Neff}$ is an ensemble parameter scaling coefficient. So no single value is chosen.

l.175: Greenland is mentioned here, is that valid for other regions?

The following has been appended:

Given the high statistical confidence in the results of the above inversion for the 4 (of 8 total) catchments with mostly strong (hard) beds, we tentatively assume this value is appropriate for all other paleo ice sheets. If there was evidence or judgment to the contrary, turning this exponent into an ensemble parameter would be trivial.

Eq.8: What is $\phi$?

It is the friction till angle, now changed to $\phi_t$, and the $\theta_t$ in eq 7 was corrected to $\phi_t$

Eq.9: How do you calculate $\frac{\partial \tau_b}{\partial U_b}$? And why is it needed?

Just take the partial derivative of equation 7. Equation 9 is the standard approach for linearizing a nonlinear term (cf the cited Patankar text for motivation).

l.190: Which two coulomb options? Again more references are needed here

The GSM also has a plain Coulomb plastic basal drag option, but since it's not recommended, I'm not describing it, and have removed an reference to it, so the relevant phrase to:

The Coulomb plastic option

l.196: What does "numerically self-consistent" mean here, and how is that assessed?

This is well described in the cited Patankar computational fluid dynamics intro textbook, and stated in the parenthetical comment:

"(the square root operation was chosen to provide an intermediate between an arithmetic mean and the numerically appropriate regularized harmonic mean for a linear diffusion process, cf Patankar, 1980)"

but at the cost of word repetition, I've now changed "appropriate" to "self-consistent" in case this was the source of confusion

l.212 and l.214: How are the regions with hard and soft beds determined? Can they evolve over time and if so under what conditions?

The criterion is described in the beginning of the subsection. Evolution over time will happen when full two-way coupling of the subglacial sediment processes model (sect 2.14) is turned on. This is now made clear in that subsection.

Eq.12: How is $T_{exp}$ chosen and what are typical values?

The following has been added to address this:

On the basis of numerical experiments, resolution dependence is minimized for values of $T_{\exp}$ between 5 and 10 (Hank et al., 2023), with the GSM having a default value of 10.

But this does effectively duplicate the existing statement:

" A detailed resolution scaling analysis of this issue has recently been published (Hank et al., 2023), and its recommended activation function is under compile flag choice"

l.242: Which "physical grounds" are referred to here? And what does that represent?

As stated in the subsequent parentheses:

" (since the range of subgrid basal temperatures for a grid cell, when not fully warm-based, will generally be larger for a larger grid cell)."

l.266: What is the horizontal diffusion not included? Also many coefficients and variables are not listed and explained in this equation.

Try a simple scale analysis; horizontal diffusion. The smallest grid resolution the GSM will be run at for a continental ice sheet is $\approx 10$ km. EAIS has the lowest velocities (to minimize the horizontal advection to horizontal diffusion ratio) with a scale velocity of 10 m/yr ice velocity (given mean accumulation of 3 cm/yr and EAIS dimensions):

$k\frac{d^2T}{dx^2} \approx 2Wm^{-1}K^{-1}\frac{1K}{(1\times10^4)^2m^2} = 2\times10^{-8}W/m^3$

advection $= \rho C V \nabla T$

$\approx 900\,kg/m^3 \cdot\, 2000J/kg \cdot 10\,m/yr \cdot 1yr/(3\times10^7 s) \cdot \frac{1K}{1\times10^4\,m} = 6\times10^{-5}W/m^3$

so horizontal advection has 3000 times the energy flux for this case. For smaller ice sheets, a smaller minimum grid size will be offset by higher scale velocities.

$\vec{r}$, z, $k_i$, $\rho_i$, and $c_i$ are standard symbols with the latter three listed in Table 4 for which a reference has now been added. If someone is reading this section, I will presuppose basic understanding of the energy conservation equation for a fluid otherwise this is like asking an author to explain the Navier Stokes equation in a GCM model description paper. I now make clear that $\mathbf{V}(\vec{r})$ is the 3D ice velocity.

l.267: Explain what the boundary heat flux is.

The heat generated by basal sliding occurs at the basal ice cell edge and is therefore not a source term but a heat flux at the bottom boundary of the ice sheet. I've added: $(\tau_\mathbf{b} \cdot \mathbf{u_b})$.

l.272: "discretization of the basal ice grid cell": what does that mean and how is it discretized?

The paragraph has been rewritten to a more accurate and less confusing form :

The discretization of the energy conservation equation for the basal ice grid cell is non-standard to enable a solution of the temperature at the basal interface as needed for accurately determining thermal activation of basal sliding. To do so, horizontal advection and the time derivative in eq. 16 use a basal grid-cell centre temperature that is determined via linear interpolation between the basal interface temperature and cell centre temperature of the vertically adjacent ice grid cell. On the other hand, solution of the basal interface temperature means that no interpolation is required for vertical diffusion and advection.

l.287: Why do that and not use all the terms? It should not be more complicated or expensive to run

Bit of a can of worms, but opening it up: I've changed "As is standard given

space-time scales to the more accurate.." to:

Given the grid cell dimensions for large ice sheet glacial cycle contexts (and generally much lower horizontal temperature gradients relative to vertical temperature gradients), the default bed thermodynamics configuration assumes vertical diffusive heat transport only (as supported by a straight-forward scale analysis). A near unique feature is that the bed thermal model accounts for permafrost via a standard heat capacity approximation (Osterkamp, 1987; Williams and Smith, 1989; Mottaghy and Rath, 2006). It also applies temperature forcing corrections at the top of subaerial frozen ground to partly account for the effects of seasonal snow cover and surface vegetation (Smith and Riseborough, 2002). The GSM thermal bed has a default depth (GSM parameter BEDTdepth) of 4 km for which the lower flux boundary condition is specified by an input map (section 2.17).

The GSM has the option (-DthreeDbedTdiffusion) of added explicit time-stepping horizontal bed thermal diffusion. This would be more appropriate for grid resolutions of 10 km or finer or for regions where there are large horizontal gradients in the input deep geothermal heat flux field. However, the available reconstructions for this boundary condition are somewhat vague as to the exact depth they represent, often self-described as being near the bed surface. These reconstructions will already embody horizontal heat diffusion up to their representative depth. If this depth is above the chosen (default 4 km) depth of the bed thermal model, activation of GSM horizontal heat diffusion would effectively result in erroneous doubling of horizontal heat diffusion over the depth of overlap.

The activation of horizontal diffusion is computationally inexpensive (about a 2% increase in run time). For a coarse 40 km grid resolution Antarctic two glacial cycle simulation, its addition can alter root-mean-square-error discrepancies with present day input ice topography and observed marginal ice velocities by approximately 10 m and 45 m/yr respectively.

l.293: Add references here

reference is given (Tarasov and Peltier, 2007), I can't provide a reference for the lack of the 4km deep geothermal heat flux inversion, because there isn't such an inversion. I've removed "confident" if that was the source of confusion. Anyway, this whole subsection has been rewritten to the above..

l.329: "mid to high latitude": why use such a wide range? Could that be narrowed down?

Not if you want a relation that will cover paleo ice sheet geographic locations.

Figure 2 caption: "Comparison GCM" − > "Comparison of GCM"

done

l.348: "non-Gaussian distribution": describe what are the properties of the distribution

I've added:

This distribution has skewness and kurtosis with linear dependence on mean monthly temperature, and quadratic dependence for the standard deviation

l.358: "for Greenland" − > can this change in other regions with different temperatures and precipitation?

likely, but unless you can point me to a similar study (to that of the RACMO one) for other ice sheets, I can only work with what is available. But good to remind the reader of this limitation, so I've added:

Given the tuning against Greenland RCM modelling, the applicability of this scheme and its current parameters to other ice sheets is unclear. It is likely reasonably applicable for other similar maritime proximal ice sheet ablation zones on the basis of climatic similarity. As such, RCM modelling of continental ablation zones would be a priority for testing/refinement of this scheme.

l.374-379: What about increased melt in warm conditions? This paragraph is confusing and would deserve additional information.

Warm conditions will give more PDDs (which are computed monthly as explicitly stated), so I don't see the relevance of this query here. However, the first part of the paragraph has been rewritten to hopefully provide more clarity:

Though precipitation, surface melt, PDDs, positive temperature insolation, and NDYs are computed monthly, surface meltwater refreezing is computed yearly. Furthermore, all net snow accumulation (ie after melt loss) in one year transitions to ice the next year.

l.384: "requisite" − > "require"

I find requisite more appropriate (= "needed for a particular purpose", www.merriam-webster.com/dictionary).

l.402: "horizontal advection due to sub ice shelf ocean circulation": Are you talking about the ice or ocean advection?

rephrased to:

A related limitation is the plume model is purely buoyancy driven and therefore ignores horizontal advection due to sub ice shelf ocean circulation.

l.414: Why is needed to further reduce the friction beyond the ratio of grounded ice? What are the physical and/or numerical reasons for that? And how was a value of 0.5 estimated?

You've lost me. The paragraph is discussing submarine melt at the grounding line, not friction. The motivation for the 0.5 value is as stated:

Sensitivity tests have found grounding retreat and advance to be more stable for RfactGLssm= 0.5 compared to the simulations with subshelf melt only for fully floating grid cells.

To better understand the motivation, I've also added the following:

This is in accordance with resolution convergence tests comparing application of submarine melt to just floating grid cells as well as to fractional inclusion of grid cells crossing the grounding line grid cells in proportion to the subgrid grounded ice fraction (Seroussi and Morlighem, 2018).

l.417: in the Rignot melt approximation, q is the freshwater discharge and its unit is in $m^3$/day. How is the value you are using related to the Rignot approximation? And what does it mean in terms of size of water channels?

All of the Rignot et al $q_*$ are in m/d for the ice front (cf their table 1 and equation 1). As stated in the original submission (with a slight expansion in the current version): "The above equation is rescaled for m/yr quantities. q is approximated by scaling the sum of twice the subglacial melt rate of the grid cell (to allow for some upstream contribution) and surface runoff by the grid

cell area to marine face area ratio." There is nothing in this that will specify or make an assumption about the size of the subglacial meltwater channels, so I don't understand your latter question.

l.420: the correct units should be described for a and $\beta$, with no units Eq. 24 is not consistent

a and $\beta$ are exponents, so have to be dimensionless. I had followed the source article convention of skipping the appropriate dimensions for A and B, but I have now rectified this in the revised version.

l.424: What is "$C_{face}$"? What does it represent?

As is indicated by eq 24, $C_{face}$ is an "ensemble parameter scaling" of the B coefficient and therefore of the component of marine ice face melt independent of subglacial meltwater flux.

To clearly motivate this, I've added:

To allow for uncertainties in the application of the above formula to the coarser grid scale resolution typical of paleo ice sheet modelling, an ensemble scaling parameter ($C_{\text{face}}$) is added. Until recently this has only been applied to the $B$ coefficient (as $C_{\text{face}_\text{B}}$ in eq. 24). However, given the potentially large uncertainties due to submarine circulation (driven in large part by buoyancy forcing from meltwater), the GSM has the option of switching the ensemble parameter to the $A$ coefficient (and thus $C_{\text{face}_\text{A}}$ in eq. 24, using the -DCfaceMltFW compile flag).

l.425: I thought ice shelves were necessarily marine. If "marine" is required, you want to explain why.

why can't you have an ice shelf on a proglacial lake?

l.426: "For marine floating ice calving" $->$ "For ice shelf calving"

cf previous. Furthermore, the boundary between ice shelf and ice tongue is not clear, and this is applied to both.

Eq.25: What are r and $r_c$?

I've added:

As indicated in eq. 26, calving is activated when the relative total crevasse depth (r) reaches the critical relative depth $r_c$, with latter set to the value in Pollard et al. (2015).

Eq.26: What are $d_a, d_t$, and $d_w$? Mention what they represent just after the equation since the details are half a page later.

The text already states:

where each d? term represents a contribution to crack depth propagation

to which I've now appended:

as detailed below.

adding anything more seems unnecessary to me, especially given the length concerns.

Eq.28: What use such a form for this equation? What does it represent?

The text after the equation has been expanded to the following:

As detailed in Pollard et al. (2015), this derives from a steady flow solution to the time integral of the ice divergence along the flowlines. To improve GSM fits to present day (PD) observed Antarctic ice shelf extents, the 1600 m/yr

value for the denominator (representing the flowline velocity at the beginning of the ice shelf trajectory) in Pollard et al. (2015) was reduced by a factor of 2.

l.483: "heat to melt icebergs" − > Does it mean the icebergs or at least the amount of icebergs are tracked? How does it work?

I've appended:

As such, lacustrine calving is simply implemented as extra melt applied to the grid cell covering the calving margin.

This along with the details in the next paragraph (including "the potential iceberg melt is just set to the total computed net potential surface melt of adjacent lake filled grid cells times a GSM parameter (flac)") hopefully make this clear now.

l.497: What is. Marine depth?

change to:

ocean depth

l.527: remove "to date"

Why? The current statement is accurate unless you can point me to another model that does this.

Eq.41: What is "scalarSealevel"? And where does the form of this equation come from?

I've replace "scalarSealevel" with the more precise "globalMeanSealevel" and have added the following:

This implementation assumes a $-125\ m$ mean glacial maximum sea level so that dradSea is the corresponding radiative forcing. The exponent value was chosen to approximately account for the limited radiative impact of the major Northern hemispheric ice sheets until their southern extent reaches regions not typically covered by snow or sea ice for the majority of the year. The exponent value will at some point be refined by modelling experiments with the EBM.

l.609-610: Why use such a process for these components?

as stated:

to account for the significant inter-model differences in the PMIP simulations

EOFs are the most efficient way to do this (each EOF successively captures the largest fraction of residual variance not accounted for by the previous EOFs, and does so orthogonally). I would expect anyone doing any kind of climate modelling to know what an EOF is. However, to help address any reader knowledge shortfalls, I've changed the parenthetical (EOF) to:

(EOFs, a mathematical tool to capture orthogonal modes of maximum variance).

l.627: What do $C_{HTM}$ and $tTheta_{wrm}$ represent?

The equation is self-explanatory and motivated by the prior statement

includes an added Holocene latitudinal warming gradient (Tagy)

I expect anyone reading this level of detail of the model, will understand enough mathematics to interpretation a multiplicative coefficient and denominator controlling the latitudinal range of the gradient interval. However, to facilitate read through, I've changed the text to:

with explicit dependence on latitude ($\theta$) and ensemble parameters $C_{\mathrm{HTM}}$ and $\Theta_{\mathrm{wrm}}$ respectively controlling amplitude and latitudinal range of the Northward linear ramp-up.

Eq.43: I don't understand how $I_c$ is calculated from $I_c$? Does it mean there are non-linear iterations? Or is it updated at different time steps?

It's a simple variable update step as would be stated in any programming language. To make this clearer, I've replace "=" with $\leftarrow$.

l.647: What is the interpretation of the ensemble phase factor?

As stated:

"to parameterize some of the uncertainty associated with the transition from interglacial to glacial atmospheric states"

Again, the interpretation is clearly expressed in the equation itself.

l.666: What are the typical lower and higher resolution? How are the fields interpolated or downscaled?

I've added:

(as per the form of eq. 46)

and:

(typically about 1000 km for climate models of intermediate complexity to at best 100 km for global general circulation climate models run for paleo contexts in semi-equilibrium time slice mode)

l.676: How is the weighting decided? What does appropriate mean in this context?

I've made this more explicit:

with weighting (Uweight) as per the corresponding Gaussian distribution.

l.716: What is a "subgrid ice flow"?

As is pretty standard, "subgrid" refers to process not resolved by the discretization at the given grid cell resolution. So subgrid ice flow is ice flow not resolved by the chosen grid resolution.

l.723-724: I don't understand this sentence. Is there always some nudging? What about the "regular" ice margin without nudging and its evolution? How about grounding lines?

The first sentence of that paragraph states:

The GSM has an **option** of automatically adjusting

I'm guessing the problem might be lack of context, so I've added:

For North America and Eurasia (Hughes et al., 2016; Dalton et al., 2020, 2022), there exist geologically reconstructed ice margin chronologies that include maximum and minimum isochrones for each time slice. For such a context, the GSM has an option of automatically ...

The remaining questions are answered via equation 49.

l.731: What is fmgm?

As already stated in the text, it is the nudging ablation factor. Its role is also indicated in the equation (49).

l.747: How is the bed roughness scale found? What dataset or parameterization is used for that?

As stated in the text, the bed roughness scale is an ensemble parameter. So the question makes no sense. Ensemble parameter values are determined by

whatever inference methodology the modeller is using. If the problem is not easily tracking what are ensemble parameters, I'm open to any suggestions for identifying font or such.

l.750: How is the location of the moulins decided?

As stated in the next sentence:

" This assumes that ice is thin enough and crevassed enough for all regions with significant surface runoff to have such englacial hydrological connectivity to the base"

but for further clarification, I've now inserted (*i.e.* grid cells) after "regions" in the above.

l.757: Additional information regarding the sediment erosion, transport and deposition is needed, including processes that they represent, time steps of these computations, tracking (or not) the sediments deposited, and their integration with the rest of the model.

All this is provided in detail in the cited references. I've added a bit more depth of the summary description included herein, but I see no point in repeating what is provided elsewhere. Note the more recent cited reference is an egusphere discussion paper that the reviewers suggested be split. In this case we agreed, and the sediment model description and validation part will soon be submitted to GMD (if I can afford the APC ...).

l.770: Why would not account for ice load changes in Greenland? Also some references to explain this further are needed.

The given sentence is:

"Not accounting for global ice load changes in Greenland simulations (as is typical for paleo ice sheet modelling) can have significant impacts (cf Figs. 4 and 5)."

It is not my job to explain why other modellers choose to ignore relevant physical processes when I'm arguing that these should be included. I also do not understand what has to be explained further. But I have modified the text to make the statement more accurate (especially given the very limited publication of last glacial cycle GRIS only modelling for the last decade).

Not accounting for global ice load changes in Greenland simulations (as used to be typical for paleo ice sheet modelling, *e.g.,* Tarasov and Peltier, 2002; Lecavalier et al., 2014), can have significant impacts (cf Figs. 7 and 8).

l.775: Why store 30 kyr and not more or less? References needed here as well

the answer was as given:

"a choice justified by sensitivity tests."

These tests are unpublished and therefore no references are available. However, the reviewer's query has prompted me to revisit the sensitivity tests, especially since they were originally done when the GSM only had one available earth rheology and were just done for the North American ice complex. This has also lead me to change the time step storage strategy. The revised draft includes a new figure (9.) showing example sensitivity results and the following description and discussion:

The load history must be stored as spherical harmonic coefficients and thereby represents a major memory load. To limit the required memory, the GSM only retains a specified past interval of load history at 100 year resolution (default 30 kyr) and then a subsequent second interval at either 500 year or 1 kyr resolution (default 210 kyr). Ice load changes prior to this second long time-step interval are continually summed and imposed as a step load. This approach requires the load intervals to be stored in first in first out (FIFO) memory stacks as well as the tracking of the variable time interval between the current time-step and the load steps in the long time-step stack.

This load history treatment has been verified with simple two step instant GRIS unloading tests (with half of the ice removed after 100 years, and the rest removed after 2.5 kyr). After 60 kyr, the maximum bed elevation difference between the default (100 year step for 30 kyr, then 1 kyr steps) versus a continuous 100 year load step configuration is $< 2$ cm for a relatively soft earth rheology ($2 \times 10^{20}$ and $3 \times 10^{21}$ Pa s respective upper and lower mantle viscosities and $< 41$ cm for an extremely stiff earth rheology (respectively $5 \times 10^{21}$ and $30 \times 10^{21}$ Pa s). However, this is not the case for a transient fully coupled simulation of the GRIS. In this case, high sensitivity of the coupled system can be evident depending on the ensemble parameters. As shown for an example parameter vector (chosen for higher sensitivity) in Figure 9., the simulated LGM ice volume anomaly relative to 0 ka can vary up to $\approx 10\%$ depending on the choice of intervals for the 100 year and either 500 or 1000 year load storage steps. Furthermore, the response to the number of time-steps is not monotonic, with eg 10 kyr coverage of the most recent load changes at 100 year time-steps performing worse for this case (relative to the maximum 38 kyr coverage simulation) than the simulation with 1 kyr coverage (NTIM1=10 in Figure 9.). These results show higher sensitivity to the GIA load history time-step than that of a previous study that also implemented variable load history time-steps Han et al. (2022). However the latter only used 200 years as their shortest GIA load history time-step, kept their short time-step interval to a fixed 5kyr and based their whole analysis on a single North American ice sheet configuration.

Note, the revised submission now has a supplemental section detailing recent changes to the GSM, so that recent submissions can reference this paper with modified GSM version numbers to eg track a version that did not have the above option of two different GIA load history time steps.

l.817: Which ones are "poorly constrained"?

I've appended "(as listed in Table 5)".

The various processes and inputs subject to noise input are listed in Table 5. The choice of noise distribution and amplitude was based on informed author judgment but should be reconsidered by any user based on context and confidence in relevant inputs.

l.823: Does it mean you can choose on which variables you put the noise? Or is it simultaneously included for all variables?

This is more a user manual question, so will be answered in that. FYI, default is is simultaneous for all variables, which is what you would want for internal discrepancy assessment. But it is easy to track the noise insertion in

the code (via the compile flag) and just turn on one at a time.

l.825: How are they interpolated on the simulation grid?

If you are talking about the input data sets they are upscaled by subgrid cell averaging or downscaled by bilinear interpolation. This I would think is more relevant for the user manual, and so will be briefly described there.

l.837: I don't understand this sentence.

The sentence is " This is facilitated by temporarily reducing the bed heat capacity by a factor of 1000. "

I have no idea what is not understandable about this sentence.

l.845: "forcing as is evident is the difference" $->$ rephrase

"is" should have been "in" :

The approach is sensitive to the initial climate forcing as is evident in the difference between simulations

l.856: How about the circular symmetry?

how can you preserve circular symmetry on a rectangular grid?

Table 7: How about the vertical velocity, the ice front and grounding line positions, and the surface mass balance and basal melt applied?

The first 3 in your list are all quantities easily diagnosed from the key fields now listed and do not appear in any of the equations in the submission (though I've added the vertical velocity component since it doesn't require an extra line). I've now added basal melt from ice thermodynamics (which is not "applied" but dynamically computed as described in the relevant section), surface mass-balance, submarine melt and refreezing, calving rate. This list isn't meant to cover all output variable options for the GSM nor all fields derived from other fields (for which eg variable maskwater(x,y,t) indicates if ice grounded or not with latter including open marine)

Figure 6: What is the difference between the last two lines? It cannot be figured out from the legend and figure caption.

It is visible in the plot: one starts at 205ka, one at 122 ka. The revised version has the starting time added in the caption.

l.863: what does "none-steady" mean?

changed to:

"simulations do not equilibrate."

l.875: "again" $->$ "against"

done

Code availability: make sure that the configurations shown in this paper and the code to reproduce the results shown (exact configuration) are also accessible.

Upon acceptance (to avoid extraneous archive versions I don't want to create another DOI'd version yet, as the user manual is an ongoing work, as are efforts to ease user configuration of the GSM), I'll provide a new DOI'd code archive with scripts and parameter files for replicating most of the plots in the main text (some plots need other code such as for RSL post-processing, which are not part of the GSM). In the meantime, I'll stick the revised archive on my website on Sept 24/25 (with the interim needed to finish the initial user manual for the archive).

**References**

[revised manuscript text omitted]

---

## Author Response (AR2)

**Response to reviewer comments**

I thank both reviewers for their careful copy-edit of the previous draft. I have also carefully re-read the whole manuscript (after not looking at it for a month) and cleaned up a few passages that were hard to follow.

**# technical staff request ###############**
**# First author responses prefixed with '#'**

Please number the sections of the supplement according to the guidelines (see "Supplements" at https://www.geoscientific-model-development.net/submission.html#assets), i.e. "S1 GSM ensemble parameter sensitivities", "S1.1 Example GSM ensemble parameter sensitivity results for Antarctica", etc.
**done**

**# referee #2 ###############**
**# All suggestions have been implemented (didn't bother with repeated "#done") unless otherwise indicated.**

Minor suggestions (line numbers refer to the manuscript version including track changes):
l.2: ". It is" -> ", which is"
l.4: "last glacial ice sheet": rephrase
l.7: which "context" are you talking about?
**-> "The above context" as explicitly stated**
l.47: remove "glaciological"
l.55: "most" was better than "all" if it's not 100% sure
**I know of no other published ice sheet modelling study using more than 6 ensemble parameters**
**so will stick to "all".**
l.59: add a comma after i.e.
l.79: add "sections" before "2.4 and A1"
l.79: spell out PSU3d
l.82: "ice shelf instabilities" -> "ice sheet instabilities"
l.84: add "section" before "2.5"
Table 1 title: "section for description" -> "sections describing these components"
l.138-141: it would be good to discuss this in more generic term, and talk more generally about the grid used and if it is the same between the different components of the model

l.142: "irregularly spaced sigma grid for velocities": is that just for the velocity or applied to all the fields? You should start by saying how many grids there are and what is common (or not) for all the components of the model. Then you can talk more about the details (in the current form we kind of have to guess as the text goes)
**changed to "for the ice dynamics solver"**
l.146-149: this is already said in line 138 and following lines
**deleted**
Eq.5: Why write u_b as a function of \tau_b and not the other way

around. Writing the basal stress as a function of velocity here would
be more consistent with the rest of the manuscript, like Eq.7. That's
also the way it naturally appears in the boundary condition of the
model.
**this form was chosen for the C_b coefficient, but I've now inverted it as requested**
l.345: "consistent 3 pt upwinding": rephrase to explain this more clearly
**rephrased to the mathematically precise:**
"discretized using a second-order consistent 3 point upwinding scheme."
l.383-385: why does it increase discrepancy?
**that is not what is stated, instead "alter" is used and the RMSE discrepancies**
**are just used as a metric of the impact on activating horizontal heat diffusion in the bed**
l.400: remove "crudely" to avoid judgment
l.426: I am confused about the "number of PDDs per day", does it mean
the number will be a fraction of a day?
**I've added a definition of PDDs to clarify:**
"PDDs for any day are herein defined as the hourly resolved time
average of the maximum of 0 and the near surface air temperature (in
degrees Celsius)."

l.453: "uses a" -> "uses an"
l.488: missing comma after "i.e."
l.554: "a ensemble" -> "an ensemble"
l.648: Why a depth a 200 m chosen for the threshold?
**to route runoff into the continental shelf for coarser grid climate model coupling.**
**Too fine a detail to worth mentioning here**
Fig.5 title: "example GSM" -> "example of GSM"
l.689: "high variance" -> "higher variance"
l.720: missing word after "Year and monthly"
**added "dependence"**
l.747: spell out "CCM"
l.810: "is obtain" -> "is obtained"
l.835: "subject to the factor": how is that applied?
**as a multiplicative factor, inserted "multiplicative" to make this clearer**
Figure 7 caption: "Example process" -> "Example of process"
l.987: I am surprised the default is "210 kyr" compared to the 500 or
1000 years mentioned above
**210kyr is the total length of the interval at 500 or 1kyr time resolution, now clarified**
**by revised placement of "(default 210 kyr)"**
Figure 8 caption: spell out RSL
l.1002: "Han et al. (2022)" -> "(Han et al. 2022)"
l.1027: missing parenthesis after "Pa s"
l.1055: for the sea level, does it include the other terms outside of
the ice sheets like the ocean thermal expansion?
**No, and ocean density changes are usually not accounted for in GIA**
**models, especially for paleo contexts**
**A key challenge would be the need for a reasonably well constrained**
**3D temperature field of the ocean through a glacial cycle.**

l.1080: "approximations of" -> "approximations for"

**I disagree given the context and given a comparative search on the internet**

Table 7 title: It is unclear if it's key or most or what? Also "with
year": does that mean every year or annually?
**removed confusing "(mostly) primary"**

l.1136: "as inferred BedMachine" -> "as inferred from BedMachine" (or based on BedMachine)
l.1145: add comma after contexts
l.1150: extra comma to remove
l.1192: add ":" after "following"
**Nope, since the subsequent list is composed of separate sentences**

**ref #1 ######################**

l55 "Verjans et al., 2022)."
Add opening parenthesis
**opening was farther up the sentence. Added 'c.f. ' to make phrase sequencing clearer**

l117 "Tables 2"
-> "Table 2"

Table2.
Maybe "North American and Eurasian specific" in line with Table 3.
**done**

l138 "2.3 A caveat on parametrizations in the GSM"
It seems 2.3 should follow right after 2.1.
**switched 2.1 with 2.2**

l274 "The GSM has not been setup"
--> "The GSM has not been set up"

l352 "A near unique feature is"
--> "A near unique feature of the GSM is"

l414 Figure 3.
The figure would be easier to read if the x and y axes had the same
range. Even better if the axes aspect ratio would be 1, which anyhow
works better in a 2 column paper.
**done**

l463 Figure 4.
Consider aspect ratio of 1 for the axes.
**done**

l498 "A related limitation is the plume model is purely buoyancy driven"
-> "A related limitation is *that* the plume model is purely buoyancy driven"

l521 "B= 0.15 oC−β mday−1 as per Rignot et al. (2016) assuming an average of 180 melt days"

For the justification of B I find "B expresses that qm is nonzero in the absence of subglacial water flux" in Rignot et al. (2016). Where does the assumption of 180 melt days come from? Wouldn't the amount of melt days change in e.g. a glacial or much warmer climate?
**This was burrying a bunch of uncertainties, which I now address explicitly,**
**by deleting the above "assuming an average of 180 melt days" and inserting:**
"An extra factor of 0.5 is also inserted in this rescaling to partially account for the impact of using annual mean inputs, given that intra-annual variations are largely driven by changes in q (annual averaging will increase the effective value of $q^a$ given that a = 0.39 < 1). The application of eq. 25 in the GSM introduces uncertainties arising from the coarser grid scale resolution typical of paleo ice sheet modelling as well as limitations in the required inputs and their averaging to annual means. As such, an ensemble scaling parameter ($C_{face}$ ) is added. "

l527 "the GSM has the option of switching the ensemble parameter to the A coefficient"
Not clear what that means.
**changed ->**
"the GSM has the option of making the A coefficient the ensemble parameter (..)."

l608 "The mass-conserving solver"
--> "The mass-conserving surface drainage solver"

l643 "August−February differences range up to 0.30 over the last two glacial cycles for the EBM derived monthly glacial index"
What is the reference for this result? Or refer to the EBM section below?
**the EBM derived index is described in the next paragraph, so just appended:**
 "described below"

l646 "with the much high variance of winter temperatures"
--> "with the much *higher* variance of winter temperatures"

l675 "Year (ty) and monthly (tm) "
Maybe "Year (ty) and month (tm)"?
**-> Yearly (ty) and monthly (tm)**

l711 "have also shown that strong grid resolution dependence"
--> "have also shown a strong grid resolution dependence"

l718 "under predict"
--> "underpredict"

l736 "CPEOF(i) and CTEOF(i))"
Add opening parenthesis

l740 "with the former have no Keewatin ice dome"
--> "with the former having no Keewatin ice dome"

l755 "A third temperature forcing component option for Antarctica"
Maybe "developed for Antarctica" or "specific for Antarctica"

l762 "is obtain with"
--> "is obtained with"

l778 "(ΘP) parameterize"
--> "(ΘP) parameterizes"

l817 "wind velocities ((UGCM)"
add extra closing bracket
**removed extra opening bracket**

l832 "CESM Earth system model"
The model was called "National Center for Atmospheric Research
Community Climate System Model version 3" or "NCAR CCSM3"

l840 "rToceanBiasCor + rw (1.−rToceanBiasCor)"
Better formatted as an equation.
**I've just created a separate equation for Tocean(t,x,y) to make all this clearer**

l849 "this inadequacy was assigned to this uncertainty, especially since this required"
A lot of "this" in this sentence.
**rewritten to**
"As the largest component of relevant climate
forcing uncertainty is the subshelf ocean temperature, we ascribe this inadequacy
to ocean temperature uncertainties. The latter are especially large
given the required spatial extrapolation of the TRACE ocean fields which do
not cover the Antarctic ice shelf sectors."

l866 "(Hughes et al., 2016; Dalton et al., 2020, 2022)"
Move to end of sentence or after "margin chronologies".

l925 "The load history must be stored as spherical harmonic
coefficients and thereby represents a major memory load" Similar to
l915 "This convolution necessitates storage of the discretized load
change history."
Combine?
**done**

l934 "(2 ×1020 and 3 ×1021 Pa s respective upper and lower mantle viscosities"
Add closing parenthesis

l942 "load history time-steps Han et al. (2022)"
--> load history time-steps (Han et al., 2022)"

l960 "in that improvements"

Something missing?
**just deleted the above**

l962 "(upper mantle viscosities were <1 ×1021 Pa s, lower ranged from
1 ×1021 to 30 ×1021 Pa s"
Not sure where to close the parenthesis. Something missing?
**closing now added to end of above quote**

l1000 "sign preserving square"
--> sign-preserving

l1039 "(Pollard et al., 2015). It has already been"
--> "(Pollard et al., 2015), it has already been"

l1071 - l1074
This paragraph feels out of place here and may be better placed at the end of 2.3.
**actually better fit for the "GSM parameters" subsection (now 2.2), so moved there**

l1111 "GIA model step at the cost of repeat iteration of the the ice sheet dynamics"
-->  "GIA model steps at the cost of repeat iteration of the ice sheet dynamics."

l1132 "isochronal depth inversions (MacGregor et al., 2015)"
A more recent update exists: https://doi.org/10.5194/essd-17-2911-2025
**updated**

l1135 "Depending on community interest and involvement"
Since the authors in their reply express hope to establish a community
around the model, I recommend proactively publishing and maintaining
the code in a more accessible place like GitHub.

**My (LT) old school workflow right now doesn't use Git. A move**
**towards that approach would need some time investment on my part and**
**given my perpetual overload, I'm only willing to do that if there is**
**some concrete interest expressed. I have added a statement to my**
**website so that potential users who would like a gitHub setup can**
**send me a brief note. If there is some concrete interest from more**
**than two respondents I will work through the GitHub conversion.**

l1171 "Nested flow charts are as follows. Relevant GSM subroutine or
variable names are shown within parentheses and relevant source files
are enclosed with square brackets."
Description should be moved to the Figure caption.
**done**

l1176 "delt is halved"
delt and other parameters appearing in the text should probably be
formatted differently. This is particularly easy to confuse with
normal text.
**-> italics**

---

## Author Response (AR3)

The code availabilty was updated to the explicit statement of
the current version as requested:

Code and data availability. A code and input data archive for the 25G
version of the GSM is available on Zenodo
(https://doi.org/10.5281/zenodo.17364330, Tarasov et al, 2025).